# Networked Communication for Decentralised Cooperative Agents in Mean-Field Control

## Abstract

The mean-field framework has been used to find approximate solutions to problems involving very large populations of symmetric, anonymous agents, which may be intractable by other methods. The cooperative mean-field control (MFC) problem has received less attention than the non-cooperative mean-field game (MFG), despite the former potentially being more useful as a tool for engineering large-scale collective behaviours. Decentralised communication algorithms have recently been introduced to MFGs, giving benefits to learning speed and robustness. Inspired by this, we introduce networked communication to MFC - where populations arguably have broader incentive to communicate - and in particular to the setting where decentralised agents learn online from a single, non-episodic run of the empirical system. We adapt recent MFG algorithms to this new setting, as well as contributing a novel sub-routine allowing networked agents to estimate the global average reward from their local neighbourhood. Previous theoretical analysis of decentralised communication in MFGs does not extend trivially to MFC. We therefore contribute new theory proving that in MFC, under certain simplifying assumptions, the networked policy communication scheme allows agents to increase social welfare faster than under *both* of the two typical alternative architectures, namely independent and centralised learning. We also prove bounds on the error of local estimates of the global average rewards. We provide experiments demonstrating the benefits of our algorithms across different classes of cooperative game even without enforcing our simplifying assumptions, with networked populations significantly outperforming both the centralised and independent baselines (Welch unpaired two-sample tests on Area-Under-Learning-Curve at $p < 0.05$) in the majority of comparisons. We also give numerous ablation studies and additional experiments concerning numbers of communication round and robustness to communication failures.

## 1 Introduction

The mean-field framework (Lasry & Lions, 2007; Huang et al., 2006) models a representative agent as interacting not with the rest of the population on a per-agent basis, but instead with a distribution over the other agents, known as the *mean field*. The framework analyses the limiting case when the population consists of an infinite number of symmetric and anonymous agents, that is, they have identical reward and transition functions which depend on the mean-field distribution rather than on the actions of specific other players. The mean-field *control* (MFC) problem is a cooperative scenario where the population seeks to maximise a social welfare criterion such as the average return received by the agents. Alternatively we can consider a non-cooperative scenario called a mean-field *game* (MFG), where each agent seeks to maximise its individual return, to which the solution is a MFG-Nash equilibrium (MFG-NE).

The MFC social optimum and the MFG-NE can respectively be used as approximate solutions to the associated finite-agent problem/game, with the error in the solution reducing as the number of agents $N$ tends to infinity (Saldi et al., 2018; Gu et al., 2021; Mondal et al., 2022; Cui et al., 2023a;b; Anahtarci et al., 2023; Yardim et al., 2024; Toumi et al., 2024; Hu & Zhang, 2024; Chen et al., 2024; Bayraktar & Kara, 2024). MFC and MFGs have therefore been used to address the difficulty faced by multi-agent reinforcement learning (MARL), which can struggle to scale computationally as $N$ increases (Yardim &

He, 2024; Zeng et al., 2024). While MFGs have been well-studied and applied to a wide variety of real-world problems (Laurière et al., 2022a), MFC has received less attention, despite possibly being more useful for engineering collective behaviours to achieve global objectives, such as in consensus, synchronisation, rendezvous, exploration, coverage or task allocation problems (Cui et al., 2023b). This paper seek to redress some of this imbalance by adapting recent developments in MFGs for MFC.

Since MFC problems can be interpreted as optimisation problems from the perspective of a social planner, classical approaches involve centralised methods (they also do so for reasons of simplicity, as in MFGs). In this context 'centralised' does not necessarily imply global observability of the whole population's actions - which could make computation infeasible given the complexity of the problem - but rather that learning is conducted from the samples of a single representative agent, whose policy updates are assumed to be automatically pushed to the rest of the population by the central node (Fornasier & Solombrino, 2014; Carmona et al., 2019; Ruthotto et al., 2020; Laurière et al., 2022a; Angiuli et al., 2022; 2023; Cui et al., 2023a; Lee et al., 2024; Denkert et al., 2024). For this reason, whilst 'centralised learning' is the term used in prior works, we generally refer to 'central-agent learning' to reduce confusion. Often the empirical mean field of the actual population is not even used to compute rewards or transitions, with the central learner instead updating an estimate of the mean field based only on its own policy, which is in turn used as input to its reward and transition functions (Carmona et al., 2019; Angiuli et al., 2022; 2023).

Recent works on MFGs, as in other areas of multi-agent research, have recognised that the existence of a central learner is a strong assumption in complex, real-world settings, as well as representing a bottleneck for computation and communication, and a vulnerable single point of failure of the system (Zhang et al., 2018; 2021a;b; Chen et al., 2021; Yardim et al., 2023; Benjamin & Abate, 2023; 2024; Jiang et al., 2024; Xu et al., 2025; Agyeman et al., 2025; Horyna et al., 2025). They advocate instead for the individual agents in the empirical population to learn policies for themselves without relying on a central node. Such works also argue that other strong classical assumptions should similarly be loosened in order to make MFGs applicable to real-world, embodied problems such as swarm robotics. They therefore contend that, aside from decentralised learning, desirable qualities for mean-field algorithms include: learning from the population's empirical mean field (i.e. the mean-field distribution is generated only by the agents' policies, rather than being manipulated by the algorithm itself or by an external oracle/simulator); learning online from a single, non-episodic system run (i.e. similar to above, the population is not arbitrarily reset by an external controller); learning without reliance on a model of the system; and using function approximation to allow scalability to high-dimensional observations (including the option to include the mean field in the input to policies).

Until now, no work on MFC has met all these criteria. Some recent works have considered decentralisation in MFC, but Bayraktar & Kara (2024) requires that decentralised agents optimise for learnt models of the system dynamics (and learning is only fully independent when the population is large but finite rather than infinite), while Cui et al. (2023b) presents a model-free deep learning algorithm that gives decentralised execution but requires centralised, episodic training. This latter work stipulates that decentralised training can be achieved if all agents can directly observe the mean-field distribution and use the same seed to correlate their actions, though they only provide empirical results for the centralised scenario, while Bayraktar & Kara (2024) provides no empirical results at all. However, assuming decentralised agents have access to this global information is unrealistic, and in the non-cooperative MFG setting Benjamin & Abate (2024) have shown that networked communication between decentralised agents allows agents to estimate the global mean field from a local neighbourhood. They also show that proliferating high-performing policies through the population via decentralised communication (in a manner reminiscent of distributed embodied evolutionary algorithms (Hart et al., 2015; Fernández Pérez et al., 2018; Fernández Pérez & Sanchez, 2019; Cazenille et al., 2025; Sissodia et al., 2025)) improves training time and avoidance of local optima, particularly over the case of agents learning entirely independently, but often also over populations with a single central learner.

Inspired by this non-cooperative MFG work, we introduce networked communication to MFC for the first time, where populations arguably have even more incentive to communicate. This allows us to present a model-free deep learning algorithm that fulfils all of the proposed desiderata, including learning online from a single non-episodic run of the empirical system, and decentralised training without needing to observe global information: we contribute a novel sub-routine for estimating the global average reward from local communication, in addition to the existing sub-routine for estimating the global mean field from Benjamin

& Abate (2024). Previous theoretical analysis of networked communication in the non-cooperative MFG setting does not extend trivially to MFC, so we contribute new theoretical proofs showing that, under certain simplifying assumptions, the decentralised policy exchange sub-routine of our algorithm allows networked populations to learn faster than both the independent *and* the central-agent alternatives in the MFC setting, across different classes of cooperative game (coordination and anti-coordination). We additionally prove error bounds on our sub-routine for local estimation of the global average reward. We demonstrate our algorithms experimentally in numerous games - where without enforcing our simplifying assumptions we find our networked algorithms learn significantly faster than the alternative architectures in the majority of comparisons, as confirmed by Welch unpaired two-sample tests on the Area-Under-Learning-Curve - as well as contributing an empirical study of the algorithms' robustness to communication failures, along with several ablation studies. The latter suggest that communication of policies has more influence on learning speed than communication-based estimation of average rewards or of the mean field. In summary, our contributions include:

- We provide the first algorithms in MFC for model-free training without any central coordination or provision of information, as well as the first MFC algorithms for online learning from a single, non-episodic run of the empirical system.

  - We contribute a novel sub-routine allowing decentralised agents to estimate the global average reward via networked communication, and incorporate an existing sub-routine used in MFGs for estimating the global mean field aided by local communication.

- We prove theoretically that in this context, decentralised networked communication of policies can improve learning speed over the independent *and* central-agent architectures, under certain simplifying assumptions.

- We also give theoretical error bounds on our sub-routine for local estimation of the global average reward.

- We provide extensive experiments demonstrating that our networked communication algorithms permit significantly faster learning than both the central-agent and independent architectures in the majority of our comparisons (Welch unpaired two-sample tests on Area-Under-Learning-Curve at $p < 0.05$), without enforcing our simplifying theoretical assumptions.

- We give ablation studies of various parts of our algorithms (which suggest that communication of policies is the most influential sub-routine), as well as a study of robustness to communication failures.

We provide further comparison with related work in Sec. 2, give preliminaries in Sec. 3, and our algorithms in Sec. 4. We present theoretical results in Sec. 5 and experiments in Sec. 6, before suggesting future work in Sec. 7.

## 2 Related work

We discuss here the research most closely related to our present work, focusing on decentralisation and networked communication, and clarifying the differences with prior methods and settings. We refer the reader to Laurière et al. (2022a) for a broader survey of MFC.

Numerous works claiming to study decentralisation in MFC take this to mean only that agents do not have access to the specific states of all other agents, and have policies depending on their local state and possibly the mean field, all of which we take as a given in our work. They nevertheless rely on a central learner or coordinator that provides global information to all agents, a dependence that we remove in our work. This applies, for example, to Grammatico et al. (2016), where a 'central population coordinator' broadcasts a common signal to all agents, and to Tajeddini et al. (2017), which presents a leader-follower setting where a virtual 'central population coordinator' estimates the mean-field trajectory of the whole population in place

of an empirical population. Farzaneh et al. (2020) similarly requires a central coordinator, and also presents a non-cooperative scenario so does not actually fall under MFC despite being referred to as such.

In Cui et al. (2023b), decentralisation applies only during execution, and they offer a centralised-training decentralised-execution method (as also in Cui et al. (2023a)). They say that decentralised training could be achieved if the global mean field is observable and all agents use the same seed to correlate their actions, whilst we do not require either assumption for our decentralised training algorithm. They also train episodically whereas we learn online from a single run of the system. Finally, their experiments focus only on coordination games, whereas we additionally explore empirical effects resulting from decentralised training in anti-coordination games, where populations can gain higher rewards by diversifying their behaviour.

Bayraktar & Kara (2024) considers independent, 'online' learning for MFC in a setting that is different from ours. Crucially, their method involves agents first estimating a model (reward and transition functions) of the system by conducting 'online' updates using samples collected while following exploration policies. Only once having done so do they compute execution policies that are optimal with respect to the estimated model. We argue that having a dedicated exploration phase is infeasible for many real-world applications, and instead present a fully model-free online learning algorithm. Moreover, their setting only permits independent learning if $N$ is large but finite. For infinite populations, a central coordinator is required to supply common noise to aid exploration during the initial phase, and if the optimal policy for the estimated model is not unique, centralised coordination is required to allow the agents to agree on which policy to execute. Our networked algorithm requires no such special considerations. Finally, their work is purely theoretical, whereas we provide extensive empirical results.

Angiuli et al. (2022) and Angiuli et al. (2023) provide algorithms for MFC learning from a single run, but there it is a single run only of a 'representative' player that is used to simulate the mean field, rather than a single run of the empirical population as in our work. Their algorithms are thus inherently centralised, as well as involving two timescales for updating the mean-field approximation, which we argue is unlikely to be a practical paradigm for training in complex real-world systems such as robotic swarms.

Our work is also closely related to Benjamin & Abate (2023) and Benjamin & Abate (2024), which introduce networked communication to the non-cooperative MFG setting. By adapting their communication scheme and learning algorithm, we introduce networked communication to the cooperative MFC setting, where it is arguably more applicable due to broader incentives for agents to communicate policies. Their works focus on coordination games to justify the sharing of policies (though Benjamin & Abate (2024) does demonstrate empirically that networked agents outperform independent agents in a non-cooperative anti-coordination game, indicating that self-interested agents do nevertheless have incentive to communicate), whilst we provide extensive theoretical and empirical results on the benefits of policy sharing in MFC for both coordination and anti-coordination games. We leverage Alg. 4 from Benjamin & Abate (2024) for estimating the global mean field from a local neighbourhood, but additionally contribute the novel Alg. 1 for estimating the global average reward from a local neighbourhood for the MFC setting.

## 3 Preliminaries

### 3.1 Mean-field control

We use the following notation. $N$ is the number of agents in a population, with $\mathcal{S}$ and $\mathcal{A}$ representing the finite state and common action spaces. The set of probability measures on a finite set $\mathcal{X}$ is denoted $\Delta_{\mathcal{X}}$, and $\mathbf{e}_x \in \Delta_{\mathcal{X}}$ for $x \in \mathcal{X}$ is a one-hot vector with only the entry corresponding to $x$ set to 1, and all others set to 0. For time $t \geq 0$, $\hat{\mu}_t = \frac{1}{N} \sum_{i=1}^{N} \sum_{s \in \mathcal{S}} \mathbb{1}_{s_t^i = s} \mathbf{e}_s \in \Delta_{\mathcal{S}}$ is a vector of length $|\mathcal{S}|$ denoting the empirical categorical state distribution of the $N$ agents at time $t$. For agent $i \in \{1 \dots N\}$, $i$'s policy $\pi^i \in \Pi$ depends on its observation $o_t^i$. We give different forms that this observation can take, and relatedly a more formal definition of the policy, after the following.

**Definition 3.1** (*N*-player stochastic cooperative control problem with symmetric, anonymous agents)**.** This is given by the tuple $\langle N, \mathcal{S}, \mathcal{A}, P, R, \gamma \rangle$, where $\mathcal{A}$ is the action space, identical for each agent, $\mathcal{S}$ is the identical state space of each agent, such that their initial states are $\{s_0^i\}_{i=1}^{N} \in \mathcal{S}^N$ sampled from some initial

distribution $\mu_0 \in \Delta_{\mathcal{S}}$, and their policies are $\{\pi^i\}_{i=1}^N \in \Pi^N$. $P : \mathcal{S} \times \mathcal{A} \times \Delta_{\mathcal{S}} \to \Delta_{\mathcal{S}}$ is the transition function and $R : \mathcal{S} \times \mathcal{A} \times \Delta_{\mathcal{S}} \to [0,1]$ is the reward function, both identical to all agents, and which map each agent's local state and action and the population's empirical distribution to transition probabilities and bounded rewards, respectively, i.e. $\forall i \in \{1, \dots, N\}$: $s_{t+1}^i \sim P(\cdot | s_t^i, a_t^i, \hat{\mu}_t)$ and $r_t^i = R(s_t^i, a_t^i, \hat{\mu}_t)$.

For the joint policy $\boldsymbol{\pi} := (\pi^1, \dots, \pi^N) \in \Pi^N$, an individual agent's discounted return is given by:

**Definition 3.2** (Individual expected discounted return). For all $i, j \in \{1, \dots, N\}$, $i$'s return is

$$V^i(\boldsymbol{\pi}, \mu_{\bar{t}}) = \mathbb{E}\left[\sum_{t=\bar{t}}^{\infty} \gamma^t R(s_t^i, a_t^i, \hat{\mu}_t) \Bigg|_{\substack{s_{\bar{t}}^j \sim \mu_{\bar{t}} \\ a_t^j \sim \pi^j(o_t^j) \\ s_{t+1}^j \sim P(\cdot|s_t^j, a_t^j, \hat{\mu}_t)}}\right].$$

However, the maximisation objective for this *cooperative* problem is:

**Definition 3.3** (Individual expected discounted returns averaged across the $N$-agent population). For all $i, j \in \{1, \dots, N\}$ the average return is

$$V^{pop}(\boldsymbol{\pi}, \mu_{\bar{t}}) = \frac{1}{N}\sum_i^N V^i(\boldsymbol{\pi}, \mu_{\bar{t}}) = \mathbb{E}\left[\frac{1}{N}\sum_{t=\bar{t}}^{\infty}\sum_i^N \gamma^t R(s_t^i, a_t^i, \hat{\mu}_t) \Bigg|_{\substack{s_{\bar{t}}^j \sim \mu_{\bar{t}} \\ a_t^j \sim \pi^j(o_t^j) \\ s_{t+1}^j \sim P(\cdot|s_t^j, a_t^j, \hat{\mu}_t)}}\right].$$

That is, the solution to the control problem is $\boldsymbol{\pi}^* = \arg\max_{\boldsymbol{\pi} \in \Pi^N} V^{pop}(\boldsymbol{\pi}, \mu_{\bar{t}})$.

At the limit as $N \to \infty$, the infinite population of agents can be characterised as a limit distribution $\mu \in \Delta_{\mathcal{S}}$; the infinite-agent setting is termed a MFC problem. The *mean-field flow* $\boldsymbol{\mu}$ is given by the infinite sequence of mean-field distributions s.t. $\boldsymbol{\mu} = (\mu_t)_{t \geq 0}$.

**Definition 3.4** (Induced mean-field flow). We denote by $I(\pi)$ the mean-field flow $\boldsymbol{\mu}$ induced when all the agents follow $\pi$, where this is generated from $\pi$ by

$$\mu_{t+1}(s') = \sum_{s,a} \mu_t(s)\pi(a|o_t)P(s'|s, a, \mu_t).$$

The snapshot of this induced flow at $t$ is given by $I(\pi)_t$.

**Definition 3.5** (Social welfare). When all agents follow policy $\pi$ giving mean-field flow $\boldsymbol{\mu} = I(\pi)$, $\pi$'s social welfare is

$$W(\pi; I(\pi)) = \mathbb{E}\left[\sum_{t=\bar{t}}^{\infty} \gamma^t (R(s_t, a_t, I(\pi)_t)) \Bigg|_{\substack{s_{\bar{t}} \sim \mu_{\bar{t}} \\ a_t \sim \pi(\cdot|o_t) \\ s_{t+1} \sim P(\cdot|s_t, a_t, I(\pi)_t)}}\right].$$

**Definition 3.6** (Social optimum). The solution to the MFC problem is a social optimum policy $\pi^* \in \Pi$ that maximises the social welfare function in Def. 3.5, i.e. $\pi^* = \arg\max_{\pi \in \Pi} W(\pi; I(\pi))$.

**Remark 3.7.** Previous works showed that the MFC social optimum $\pi^*$ gives a good approximate solution to the harder-to-solve finite-agent problem (i.e. if $\boldsymbol{\pi} = (\pi^*, \dots, \pi^*)$), with the error characterised by $\mathcal{O}(\frac{1}{\sqrt{N}})$ (Gu et al., 2021; Mondal et al., 2022; Cui et al., 2023a;b; Bayraktar & Kara, 2024).

When the distribution is the same for all $t$, i.e. $\mu_t = \mu_{t+1}$ $\forall t \geq 0$, we say the mean-field flow is *stationary*, giving a stationary MFC problem. *Non-stationary* problems may require the policy to depend on the mean field such that $o_t^i = (s_t^i, \hat{\mu}_t)$, whereas the observation in the stationary case can be simplified to $o_t^i = s_t^i$. However, since classical approaches to the MFC problem often conceive of a central planner trying to guide the population to a distribution that maximises the expected return, they sometimes have policies that depend on the mean field even in the stationary case (Laurière et al., 2022a; Carmona et al., 2023; Cui et al., 2023b). Therefore *we permit mean field-dependent policies for the sake of generality, but show through our ablation studies that in practice our algorithms require only $\pi^i(a|o_t^i) = \pi^i(a|s_t^i)$ in our experimental tasks*, which have *stationary* solutions.

Furthermore, it is unrealistic to assume that decentralised agents with a possibly limited communication radius would have perfect observability of the global mean field $\hat{\mu}_t$. Therefore we allow agents to form a

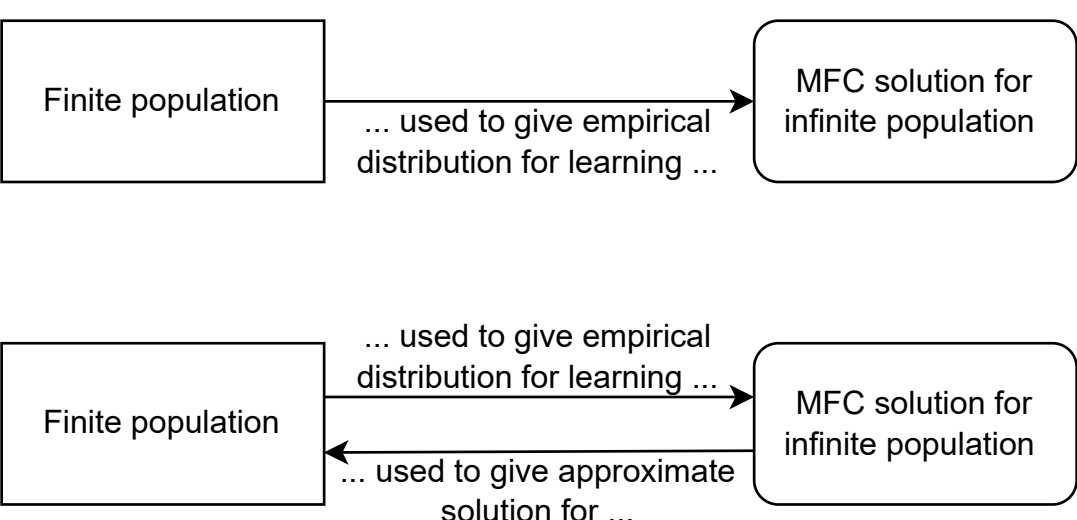

Figure 1: Two possible ways to conceive of our work regarding the relationship between the infinite- and finite-population control problems, described in Rem. 3.8. Note that using the finite empirical population to learn the single-policy MFC social optimum $\boldsymbol{\pi} = (\pi^*, \ldots, \pi^*)$ for the infinite population (Def. 3.6) is *not* the same as directly finding $\boldsymbol{\pi}^* = \arg\max_{\boldsymbol{\pi} \in \Pi^N} V^{pop}(\boldsymbol{\pi}, \mu_{\bar{t}}) = (\pi^1, \ldots, \pi^N)$, i.e. the tuple of *individual* policies that maximises the individual expected discounted returns averaged across the $N$-agent population in Def. 3.3, a problem known to be hard (Cui et al., 2023b; Bernstein et al., 2002).

local estimate $\tilde{\hat{\mu}}_t^i$ which can be improved by communication with neighbours, using Alg. 4 (from Alg. 3 in Benjamin & Abate (2024) for the MFG setting). We thus have $o_t^i = (s_t^i, \tilde{\hat{\mu}}_t^i)$. Formally we can now say that when $o_t^i = (s_t^i, \hat{\mu}_t)$ or $(s_t^i, \tilde{\hat{\mu}}_t^i)$, we have the set of policies defined as $\Pi = \{\pi : \mathcal{S} \times \Delta_{\mathcal{S}} \to \Delta_{\mathcal{A}}\}$, and the set of Q-functions denoted $\mathcal{Q} = \{q : \mathcal{S} \times \Delta_{\mathcal{S}} \times \mathcal{A} \to \mathbb{R}\}$. (N.b. when $o_t^i = s_t^i$, we instead have $\Pi = \{\pi : \mathcal{S} \to \Delta_{\mathcal{A}}\}$ and $\mathcal{Q} = \{q : \mathcal{S} \times \mathcal{A} \to \mathbb{R}\}$.)

**Remark 3.8.** We have just seen that an optimal solution to the theoretical MFC problem is a single policy that, when followed by all agents in the infinite population, maximises the population's expected return. We give two ways to conceive of our work, illustrated in Fig. 1, which mirror and make more explicit the motivations underpinning other MFC works (Cui et al., 2023b; Dayanikli et al., 2024; Zaman et al., 2024; Bayraktar & Kara, 2024; Yang et al., 2025).

1. Firstly, while previous works might make unrealistic assumptions about access to an oracle for the infinite population (Carmona et al., 2019; Laurière et al., 2022a; Angiuli et al., 2022; 2023), we contribute algorithms that allow the solution to a MFC problem to be learnt using the empirical distribution of a decentralised finite population. Note that it is unnecessary (and may be impractical) to assume the decentralised agents always follow a single identical policy throughout training.

2. Alternatively, we may have originally been interested in solving a cooperative problem for a large, finite population, but, due to the scalability issues of learning approaches like MARL, were forced to turn to the MFC framework to find a policy that gives an approximate solution to the finite-population problem. We contribute algorithms that allow the deployed finite population to find the MFC solution that in turn approximately solves the original problem, without unrealistic assumptions about centralised training. Under this framing, it may matter less whether all agents follow a single policy in practice (Yardim et al. (2023) and Benjamin & Abate (2023; 2024) follow a similar logic in MFGs).

## 3.2 Munchausen Online Mirror Descent

Recent works have solved MFGs from non-episodic runs of the finite-population empirical system using a form of policy iteration called Online Mirror Descent (OMD) (Benjamin & Abate, 2024); we adapt this to learn a social optimum in the MFC setting. OMD begins with an initial policy $\pi_0$, and then at each iteration $k$, evaluates the current policy $\pi_k$ with respect to its induced mean-field flow $\boldsymbol{\mu} = I(\pi_k)$, to compute the Q-function of $\pi_k$, which is denoted $Q_{k+1}$. To stabilise the learning process, we then use a weighted sum over this and past Q-functions, and set $\pi_{k+1}$ to be the softmax over this weighted sum, such that $\pi_{k+1}(\cdot|o) = softmax\left(\frac{1}{\tau_q}\sum_{\kappa=0}^{k+1}Q_\kappa(o,\cdot)\right)$. $\tau_q$ is a temperature parameter that scales the entropy in Munchausen RL (see Sec. 4.2) (Vieillard et al., 2020); this is a different temperature to the one agents use when communicating policies, denoted $\tau_k^{comm}$ and discussed in Sec. 4.3.

If the Q-function is approximated non-linearly, it is difficult to compute this weighted sum. The *Munchausen trick* addresses this by computing a single Q-function that mimics the weighted sum using implicit regularisation based on the Kullback-Leibler (KL) divergence between $\pi_k$ and $\pi_{k+1}$ (Vieillard et al., 2020). Using this reparametrisation gives Munchausen OMD (MOMD), detailed in Sec. 4.2 (Laurière et al., 2022b; Wu et al., 2024). MOMD does not bias policies, and has the same convergence guarantees as OMD (Hadikhanloo, 2017; Perolat et al., 2021; Wu et al., 2024).

## 3.3 Networks

Our decentralised population exhibits two time-varying graphs. The first is a communication network, by which agents can exchange information:

**Definition 3.9** (Time-varying communication network)**.** The time-varying graph $(\mathcal{G}_t^{comm})_{t\geq 0}$ is given by $\mathcal{G}_t^{comm} = (\mathcal{N}, \mathcal{E}_t^{comm})$, where $\mathcal{N}$ is the set of vertices each representing an agent $i \in \{1, \ldots, N\}$, and the edge set $\mathcal{E}_t \subseteq \{(i,j) : i,j \in \mathcal{N}, i \neq j\}$ is the set of undirected links present at time $t$. A network's *diameter* $d_{\mathcal{G}_t^{comm}}$ is the maximum of the shortest path lengths between any pair of nodes.

In principle, agents can use this same communication network to receive information about others' state in order to estimate the mean field. However, we also define an alternative observation graph that is useful in a specific subclass of environments, which can most intuitively be thought of as those where agents' states are positions in physical space, which include those in our experiments. When this is the case, we usually think of agents' ability to observe each other as depending more abstractly on whether states are visible to each other. This visibility graph is:

**Definition 3.10** (Time-varying state-visibility graph)**.** The time-varying state visibility graph $(\mathcal{G}_t^{vis})_{t\geq 0}$ is given by $\mathcal{G}_t^{vis} = (\mathcal{S}', \mathcal{E}_t^{vis})$, where $\mathcal{S}'$ is the set of vertices representing the environment states $\mathcal{S}$, and the edge set $\mathcal{E}_t^{vis} \subseteq \{(s,s') : s, s' \in \mathcal{S}'\}$ is the set of undirected links present at time $t$, representing all pairs of states (i.e., vertices in $\mathcal{S}'$) that are visible to each other.

In Sec. 4.4 we present Alg. 4, which forms an initial estimate of the global empirical mean field (to serve as an observation input for agents' Q-/policy-networks) via the visibility graph $\mathcal{G}_t^{vis}$, before refining this estimate via the communication graph $\mathcal{G}_t^{comm}$. Benjamin & Abate (2024) discusses an algorithm for more general settings where the visibility graph $\mathcal{G}_t^{vis}$ does not apply.

## 4 Learning and estimation algorithms

We adapt recent algorithms for the MFG setting, where networked communication is used 1) to form local estimates of the global empirical mean field, and 2) to allow agents to adopt better-performing policies from neighbours to accelerate learning (Benjamin & Abate, 2024). We adapt these algorithms for cooperative MFC, where decentralised agents must optimise the individual returns averaged across the $N$-agent population instead of solely their individual return as in the MFG case (the decentralised agents may not always follow a common policy while training unless we make strong assumptions on the communication network as in Sec. 5, so we do not directly optimise social welfare from Def. 3.5). In order to optimise for this cooperative objective, learners must update their Q-functions with respect to samples containing the individual

---
**Algorithm 1** Average reward estimation and communication

---
**Require:** Time-dependent communication graph $\mathcal{G}_t^{comm}$, rewards $\{r_t^i\}_{i=1}^N$, number of communication rounds $C_r$

1: $\forall i$ : Initialise reward sets $\hat{\mathcal{R}}_{t,1}^i \leftarrow \{(ID^i, r_t^i)\}$
2: **for** $c_r$ in $1, \ldots, C_r$ **do**
3:      $\forall i$ : Broadcast $\hat{\mathcal{R}}_{t,c_r}^i$
4:      $\forall i$ : $J_t^i \leftarrow \{j \in \mathcal{N} : (i,j) \in \mathcal{E}_t^{comm}\}$
5:      $\forall i$ : $\hat{\mathcal{R}}_{t,(c_r+1)}^i \leftarrow \hat{\mathcal{R}}_{t,c_r}^i \cup \bigcup_{j \in J_t^i} \hat{\mathcal{R}}_{t,c_r}^j$
6: **end for**
7: $\forall i$ : $\tilde{\hat{r}}_t^i \leftarrow \frac{1}{|\hat{\mathcal{R}}_{t,C_r}^i|} \sum_{(ID,r) \in \hat{\mathcal{R}}_{t,C_r}^i} r$
8: **return** Estimates of average reward $\{\tilde{\hat{r}}_t^i\}_{i=1}^N$

---

rewards averaged across the population, rather than the simply their own individual reward. In the latter case they would be selfishly optimising only their own returns rather than the collective good, reverting to the prior algorithm that solves the non-cooperative MFG.

It is unrealistic to assume that decentralised agents have direct access to the global average reward, so we find a third use of the communication network in 3) allowing agents to estimate the global average reward $\hat{r}_t$ from a local neighbourhood. We contribute a novel algorithm Alg. 1 for this purpose (Sec. 4.1), and we describe our main learning method Alg. 2 in Sec. 4.2. Our policy communication algorithm Alg. 3, based on that in Benjamin & Abate (2024) for the MFG setting, is described in Sec. 4.3. Meanwhile Alg. 4 for estimating the mean field, which is taken from Alg. 3 in Benjamin & Abate (2024) for the MFG setting, is described in Sec. 4.4.

### 4.1 Sub-routine for networked estimation of global average reward

Our novel Alg. 1 involves agents using the communication network $\mathcal{G}_t^{comm}$ to locally estimate the global population-average reward received after a given step in the environment. Maximising the population-average reward ensures agents are solving the cooperative MFC problem instead of the non-cooperative MFG. Prior work on networked communication in MFGs did not need this algorithm to estimate the average reward, because they were only directly optimising for agents' individual rewards. Meanwhile other works on MFC involved centralised methods and assumed access to global information, so were not interested in estimating the global average reward in a decentralised manner.

Agents broadcast their received reward with a unique ID to ensure each reward is only counted once (Line 1). They add those received from their neighbours (who constitute the set $J_t^i$) to their collection, and repeat the process of broadcasting and expanding their collections for a further $C_r - 1$ rounds, so as to receive rewards from agents more than one hop away on the network (Lines 2-6). They finally set their estimate of the global average to the average of the rewards they have collected (Line 7).

### 4.2 Main learning algorithm for updating Q-networks and policies

Our novel Alg. 2, adapted from non-cooperative Alg. 1 in Benjamin & Abate (2024), contains the core method for online MFC learning using the empirical mean field in a non-episodic system run. Our MOMD-based method (Sec. 3.2) works as follows. Each agent $i$ approximates its Q-function $\check{Q}_{\theta_k^i}(o, \cdot)$ with its own neural network parametrised by $\theta_k^i$. Agent $i$'s policy is determined by

$$\pi_{\theta_k^i}(a|o) = \text{softmax}\left(\frac{1}{\tau_q}\check{Q}_{\theta_k^i}(o, \cdot)\right)(a).$$

We denote this as $\pi_k^i(a|o)$ for simplicity when appropriate. Each agent maintains a buffer (with size $M$) of collected transitions of the form $(o_t^i, a_t^i, \tilde{\hat{r}}_t^i, o_{t+1}^i)$, where $\tilde{\hat{r}}_t^i$ is $i$'s local estimate of the global average reward obtained by running Alg. 1 (Line 7). At each iteration $k$, agents empty their buffer (Line 3) before collecting

---

**Algorithm 2** Decentralised MFC learning from non-episodic system run

---

**Require:** loop parameters $K, M, L, E, C_e, C_r, C_p$, learning parameters $\gamma, \tau_q, |B|, cl, \nu, \{\tau_k^{comm}\}_{k \in \{0,\dots,K-1\}}$
**Require:** initial states $\{s_0^i\}_{i=1}^N$; $t \leftarrow 0$

1: $\forall i$ : Randomly initialise parameters $\theta_0^i$ of Q-networks $\check{Q}_{\theta_0^i}(o, \cdot)$, and set $\pi_0^i(a|o) = \text{softmax}\left(\frac{1}{\tau_q}\check{Q}_{\theta_0^i}(o, \cdot)\right)(a)$ and $\check{Q}_{\theta_{0,0}^{i,\prime}} \leftarrow \check{Q}_{\theta_0^i}(o, \cdot)$

2: **for** $k \in 0, \dots, K-1$ **do**
3:    $\forall i$: Empty $i$'s buffer
4:    **for** $m \in 0, \dots, M-1$ **do**
5:       $\{o_t^i\}_{i=1}^N \leftarrow$ **EstimateMeanFieldAlg. 4**$\left(\mathcal{G}_t^{vis}, \mathcal{G}_t^{comm}, \{s_t^i\}_{i=1}^N\right)$
6:       Take step $\forall i : a_t^i \sim \pi_k^i(\cdot|o_t^i), r_t^i = R(s_t^i, a_t^i, \hat{\mu}_t), s_{t+1}^i \sim P(\cdot|s_t^i, a_t^i, \hat{\mu}_t); t \leftarrow t+1$
7:       $\{\tilde{r}_t^i\}_{i=1}^N \leftarrow$ **EstimateAverageRewardAlg. 1**$\left(\mathcal{G}_t^{comm}, \{r_t^i\}_{i=1}^N\right)$
8:       $\forall i$: Add $\left(o_t^i, a_t^i, \tilde{r}_t^i, o_{t+1}^i\right)$ to $i$'s buffer
9:    **end for**
10:    **for** $l \in 0, \dots, L-1$ **do**
11:       $\forall i$ : Sample batch $B_{k,l}^i$ from $i$'s buffer
12:       Update $\theta$ to minimise $\hat{\mathcal{L}}(\theta, \theta')$ as in Def. 4.1
13:       If $l \mod \nu = 0$, set $\theta' \leftarrow \theta$
14:    **end for**
15:    $\check{Q}_{\theta_{k+1}^i}(o, \cdot) \leftarrow \check{Q}_{\theta_{k,L}^i}(o, \cdot)$
16:    $\forall i$ : $\pi_{k+1}^i(a|o) \leftarrow \text{softmax}\left(\frac{1}{\tau_q}\check{Q}_{\theta_{k+1}^i}(o, \cdot)\right)(a)$
17:    $\left(\{\pi_{k+1}^i\}_i, \{s_t^i\}_i, t\right) \leftarrow$ **CommunicatePolicyAlg. 3**$\left(\mathcal{G}_t^{comm}, \{\pi_{k+1}^i\}_i, \{s_t^i\}_i, t\right)$
18: **end for**
19: **return** policies $\{\pi_K^i\}_{i=1}^N$

---

$M$ new transitions in the environment (Lines 4-9). Each decentralised agent then trains its Q-network $\check{Q}_{\theta_k^i}$ via $L$ updates (Lines 10-14) as follows.

For stability, $i$ also maintains a target network $\check{Q}_{\theta_{k,l}^{i,\prime}}$ with the same architecture but parameters $\theta_{k,l}^{i,\prime}$ copied from $\theta_{k,l}^i$ less regularly than $\theta_{k,l}^i$ themselves are updated, i.e. only every $\nu$ learning iterations (Line 13). At each iteration $l$, the agent samples a random batch $B_{k,l}^i$ of $|B|$ transitions from its buffer (Line 11). It then trains its Q-network using stochastic gradient descent to minimise the loss in Def 4.1 (Line 12). The trained Q-network determines $i$'s updated policy (Line 16).

**Definition 4.1** (Q-network empirical loss). The training loss to be minimised is given by

$$\hat{\mathcal{L}}(\theta, \theta') = \frac{1}{|B|} \sum_{transition \in B_{k,l}^i} \left|\check{Q}_{\theta_{k,l}^i}(o_t, a_t) - T\right|^2,$$

where $\quad T = \tilde{r}_t + \left[\tau_q \ln \pi_{\theta_{k,l}^{i,\prime}}(a_t|o_t)\right]_{cl}^0 + \gamma \sum_{a \in \mathcal{A}} \pi_{\theta_{k,l}^{i,\prime}}(a|o_{t+1})\left(\check{Q}_{\theta_{k,l}^{i,\prime}}(o_{t+1}, a) - \tau_q \ln \pi_{\theta_{k,l}^{i,\prime}}(a|o_{t+1})\right).$

For $cl < 0$, $[\cdot]_{cl}^0$ is a clipping function used in Munchausen RL to prevent numerical issues if the policy is too close to deterministic, as the log-policy term is otherwise unbounded (Vieillard et al., 2020; Wu et al., 2024).

### 4.3 Sub-routine for communicating and refining policies

Alg. 3 (based on Alg. 1 in Benjamin & Abate (2024) for MFGs) uses the communication network $\mathcal{G}_t^{comm}$ to spread policy updates that are estimated to be better performing through the population, allowing faster learning than in the independent and central-agent cases.

---

**Algorithm 3** Policy communication and selection

---

**Require:** Time-dependent communication graph $\mathcal{G}_t^{comm}$, loop parameters $E, C_p$, learning parameters $\gamma$, $\{\tau_k^{comm}\}_{k\in\{0,\dots,K-1\}}$

**Require:** policies $\{\pi_{k+1}^i\}_{i=1}^N$; states $\{s_t^i\}_{i=1}^N$; $t$

1: $\forall i: \sigma_{k+1}^i \leftarrow 0$
2: **for** $e \in 0, \dots, E-1$ evaluation steps **do**
3:      $\{o_t^i\}_{i=1}^N \leftarrow$ **EstimateMeanFieldAlg. 4** $(\mathcal{G}_t^{vis}, \mathcal{G}_t^{comm}, \{s_t^i\}_{i=1}^N)$
4:      Take step $\forall i: a_t^i \sim \pi_k^i(\cdot|o_t^i), r_t^i = R(s_t^i, a_t^i, \hat{\mu}_t), s_{t+1}^i \sim P(\cdot|s_t^i, a_t^i, \hat{\mu}_t)$
5:      $\forall i: \sigma_{k+1}^i \leftarrow \sigma_{k+1}^i + \gamma^e \cdot r_t^i$
6:      $t \leftarrow t + 1$
7: **end for**
8: **for** $C_p$ rounds **do**
9:      $\forall i:$ Broadcast $\sigma_{k+1}^i, \pi_{k+1}^i$
10:      $\forall i: J_t^i \leftarrow i \cup \{j \in \mathcal{N} : (i,j) \in \mathcal{E}_t^{comm}\}$
11:      $\forall i:$ Select $\text{adopted}^i \sim \Pr(\text{adopted}^i = j) = \frac{\exp(\sigma_{k+1}^j/\tau_k^{comm})}{\sum_{x\in J_t^i}\exp(\sigma_{k+1}^x/\tau_k^{comm})} \;\forall j \in J_t^i$
12:      $\forall i: \sigma_{k+1}^i \leftarrow \sigma_{k+1}^{\text{adopted}^i}, \pi_{k+1}^i \leftarrow \pi_{k+1}^{\text{adopted}^i}$
13:      $\{o_t^i\}_{i=1}^N \leftarrow$ **EstimateMeanFieldAlg. 4** $(\mathcal{G}_t^{vis}, \mathcal{G}_t^{comm}, \{s_t^i\}_{i=1}^N)$
14:      Take step $\forall i: a_t^i \sim \pi_k^i(\cdot|o_t^i), r_t^i = R(s_t^i, a_t^i, \hat{\mu}_t), s_{t+1}^i \sim P(\cdot|s_t^i, a_t^i, \hat{\mu}_t); t \leftarrow t+1$
15: **end for**
16: **return** (policies $\{\pi_{k+1}^i\}_{i=1}^N$, states $\{s_t^i\}_{i=1}^N$, $t$)

---

Alg. 3 is run after agents have independently updated their policies according to their newly trained Q-networks at each iteration $k$ of the main learning algorithm (Line 17, Alg. 2). In Alg. 3, agents obtain an approximation of their *individual* expected discounted return $\{V^i(\boldsymbol{\pi}, \mu_t)\}_{i=1}^N$ (Def. 3.2), i.e. *not* the individual expected discounted returns averaged across the $N$-agent population, which would not give differentiation between the different updated policies. They do so by collecting individual rewards for $E$ steps (not added to the training buffer), and calculating the discounted sum of rewards over these finite steps, setting this value to $\sigma_{k+1}^i$ (Lines 1-7). We can characterise this approximation of the infinite-step return as $\{\sigma_{k+1}^i\}_{i=1}^N = \{\hat{V}^i(\boldsymbol{\pi}_{k+1}, \mu_t; E)\}_{i=1}^N$.

They then broadcast their Q-network parameters along with $\sigma_{k+1}^i$ (Line 9). Receiving these from their neighbours $J_t^i$ on the network, agents select which set of parameters to adopt by taking a softmax over their own and the received estimate values $\sigma_{k+1}^j \;\forall j \in J_t^i$, defined as follows (Lines 10-12):

$$\text{adopted}^i \sim \Pr\left(\text{adopted}^i = j\right) = \frac{\exp(\sigma_{k+1}^j/\tau_k^{comm})}{\sum_{x\in J_t^i}\exp(\sigma_{k+1}^x/\tau_k^{comm})}.$$

They repeat this broadcast and adoption process for $C_p$ rounds (distinct from the $C_r/C_e$ communication rounds for the other sub-routines).

### 4.4 Sub-routine for networked estimation of global empirical mean field

Networked agents use Alg. 4 (this is Alg. 3 from Benjamin & Abate (2024) for the MFG setting) to locally estimate the global empirical mean field, to serve as an observation input for their Q-/policy-networks. Recall that we include this added observation and sub-routine for generality, especially for non-stationary problems. However it is often not necessary, particularly in stationary problems like those in our experiments, where agents can find the social optimum while only observing $o_t^i = s_t^i$, and therefore would not need to estimate the mean field.

Alg. 4 involves agents using the visibility graph $\mathcal{G}_t^{vis}$ to count the number of agents in locations that fall within the visibility radius (Line 2). For $C_e$ communication rounds, agents can supplement this local count with those received from neighbours over the communication network $\mathcal{G}_t^{comm}$, in order to count agents that

---

**Algorithm 4** Mean-field estimation and communication for environments with $\mathcal{G}_t^{vis}$

---

**Require:** Time-dependent visibility graph $\mathcal{G}_t^{vis}$, time-dependent communication graph $\mathcal{G}_t^{comm}$, states $\{s_t^i\}_{i=1}^N$, number of communication rounds $C_e$
1: $\forall i, s$ : Initialise count vector $\hat{v}_{t,1}^i[s]$ with $\emptyset$
2: $\forall i, \forall s' \in \mathcal{S}' : (s_t^i, s') \in \mathcal{E}_t^{vis} : \hat{v}_{t,1}^i[s'] \leftarrow \sum_{j \in \{1,...,N\}:s_t^j = s'} 1$
3: **for** $c_e \in 1, \ldots, C_e$ **do**
4: $\quad \forall i$ : Broadcast $\hat{v}_{t,c_e}^i$
5: $\quad \forall i : J_t^i \leftarrow i \cup \{j \in \mathcal{N} : (i,j) \in \mathcal{E}_t^{comm}\}$
6: $\quad \forall i, s$ : Initialise new count vector $\hat{v}_{t,(c_e+1)}^i[s]$ with $\emptyset$
7: $\quad \forall i, s$ and $\forall j \in J_t^i : \hat{v}_{t,(c_e+1)}^i[s] \leftarrow \hat{v}_{t,c_e}^j[s]$ if $\hat{v}_{t,c_e}^j[s] \neq \emptyset$
8: **end for**
9: $\forall i : counted\_agents_t^i \leftarrow \sum_{s \in \mathcal{S}:\hat{v}_t^i[s] \neq \emptyset} \hat{v}_t^i[s]$
10: $\forall i : uncounted\_agents_t^i \leftarrow N - counted\_agents_t^i$
11: $\forall i : unseen\_states_t^i \leftarrow \sum_{s \in \mathcal{S}:\hat{v}_t^i[s] = \emptyset} 1$
12: $\forall i, s$ where $\hat{v}_t^i[s]$ is not $\emptyset$ : $\tilde{\hat{\mu}}_t^i[s] \leftarrow \frac{\hat{v}_t^i[s]}{N}$
13: $\forall i, s$ where $\hat{v}_t^i[s]$ is $\emptyset$ : $\tilde{\hat{\mu}}_t^i[s] \leftarrow \frac{uncounted\_agents_t^i}{N \times unseen\_states_t^i}$
14: **return** $\{(\text{states } s_t^i, \text{mean-field estimates } \tilde{\hat{\mu}}_t^i)\}_{i=1}^N$

---

do not fall within the visibility radius (Lines 3-8). We assume agents know the population's total size $N$, and therefore can distribute the uncounted agents uniformly over the states that remain unaccounted for after the communication rounds (Lines 9-11). Agents now have a vector containing a true or estimated count for every state; this is converted to an estimated empirical mean field by dividing all counts by $N$ (Lines 12-13).

## 5 Theoretical results

### 5.1 Introduction

We follow the definitions of the central-agent and independent-learning baseline architectures from closely related works that solve MFGs online from a non-episodic run of the empirical system (Yardim et al., 2023; Benjamin & Abate, 2023; 2024); both alternative architectures can each be seen as special cases of our networked algorithm:

- In the **central-agent** case, only arbitrary central agent $i = 1$ updates a Q-network and automatically pushes this to all other agents in place of the decentralised policy communication in Line 17 of Alg. 2. Additionally, the true global mean-field distribution and average reward are always used in place of the local estimates, i.e. $\tilde{\hat{\mu}}_t^i = \hat{\mu}_t$ and $\tilde{\hat{r}}_t^i = \hat{r}_t$.

- In the **independent** case, there are never any links in $\mathcal{G}_t^{comm}$ or $\mathcal{G}_t^{vis}$, i.e. $\mathcal{E}_t^{comm} = \mathcal{E}_t^{vis} = \emptyset$.

In Sec. 5.3 we prove theoretical bounds on the errors of agents' local estimates of the global average rewards via Alg. 1. In Sec. 5.2 we focus on the policy communication and adoption scheme, and prove theoretically that it allows networked agents to increase their returns faster than these alternatives (with the central-agent paradigm being potentially unrealistic and vulnerable in any case). Rem. 5.1 suggests informal reasons for our formal results to aid intuitive understanding.

**Remark 5.1.** Like many cooperative learning paradigms, the central-agent alternative to our networked architecture may suffer from the credit-assignment problem, in that it is not clear how learners' local state $s_t^i$ and local action $a_t^i$ contributed to the (locally estimated) *average* reward $\tilde{\hat{r}}_t^i$ (Li & Li, 2024; Cazenille et al., 2025). Agents may receive low individual reward $r_t^i$ by taking action $a_t^i$ given $o_t^i$, but would nevertheless learn that doing so was 'good' if the rest of the population took highly rewarded actions at the same step giving high average reward $\tilde{\hat{r}}_t^i$. By drawing spurious relations, an agent's updated policy $\pi_{k+1}^i(a|o)$ may negatively

impact (or simply not advance) the goal of maximising social welfare, which is problematic if such a policy is automatically pushed from the central learner to the rest of the population. (A similar argument could also be applied to independent agents, were it not for the fact that realistically they only have access to their own rewards in any case, though we given an ablation of this in Fig. 9 which shows that our logic still applies.) Including the (estimated) empirical mean field in the observation $o_t^i = (s_t^i, \tilde{\hat{\mu}}_t^i)$ might mitigate this slightly by indicating which mean fields gave high average rewards. However, this does not solve the issue of allowing learners to distinguish between helpful or unhelpful local actions $a_t^i$, whether those learners are centralised or not, since actions can affect rewards in ways other than simply by helping to reach a certain mean field. By updating policies with respect to average return but then spreading updates through the population which are estimated to give a higher *individual* return, despite this being a cooperative problem, we reduce the credit-assignment problem by replicating updated policies that should contribute positively to the individual returns averaged across the $N$-agent population, and filtering out those that do not.

Moreover, even if we assumed credit assignment were not a problem, there is randomness in the Q-network update: agents have stochastic policies and thus may collect a wide variety of transitions to add to their individual buffers, from which they sample randomly when training their Q-networks. There may therefore be considerable variance in the quality of their estimated Q-functions, leading in turn to variance in the quality of policy updates. At each iteration of the central-agent algorithm, in *expectation* the central learner will by definition have an average-quality update, and its updated policy will be pushed to the entire population whether or not it performs well. Our decentralised networked approach permits beneficial parallelisation in place of this single-learner method, by generating a whole population of possible updates, from which the one(s) estimated to be best-performing can be selected via a process akin to the comparison of fitness functions in evolutionary algorithms. These are then spread around the population, biasing networked populations towards better performing updates.

We give the theoretical analysis separately for two important subclasses of cooperative game usually found in MFC, which have different reward structures and therefore can incentivise different population behaviour:

1. *coordination games*, where the social welfare is increased by agents aligning their strategies, such as in consensus/synchronisation/rendezvous tasks;

2. *anti-coordination games*, where the social welfare is increased by the population exhibiting diverse strategies, such as in exploration, coverage or task allocation games.

These subclasses cover a large proportion of cooperative objectives in symmetric, anonymous settings with large populations. We emphasise that the fact that agents would in principle benefit from having diverse policies in anti-coordination games does not contradict the classical MFC framework that simplifies the infinite population problem by finding the single policy to be shared by all agents. In the symmetric (i.e. identical reward and transition functions) MFC limit, an optimal solution can be realised by having the infinite agents all follow the single socially optimal policy, even for reward functions that favour diversity. A very large number of works on both MFC and MFGs conduct experiments on anti-coordination games, particularly dispersal and exploration tasks, despite assuming that the population follows a shared single policy learnt by a central node (Ruthotto et al., 2020; Laurière et al., 2022a; Lee et al., 2024). We make the distinction between coordination and anti-coordination games to aid theoretical analysis of our decentralised policy adoption scheme compared with entirely independent learning: while it is intuitive that adopting independently-updated policies from neighbours via the communication scheme could be beneficial in coordination games, we also show theoretically and empirically that the adoption scheme provides a benefit in anti-coordination games, though this requires separate analysis.

To define the two types of game, we first introduce the following functions. $\mathbb{I}[\cdot]$ is the indicator function, which equals 1 if the condition inside is true and 0 otherwise. $b : \Pi \to \mathbb{R}_{\geq 0}$ is a *base return function* that quantifies a policy's inherent ability to receive rewards regardless of how many other agents follow the same strategy. For example, if agents are rewarded for agreeing on one of a number of targets at which to meet, then policies that visit none of the designated targets will have lower returns than those that do, whether agents are aligned or not. Finally, $f_c : \mathbb{N} \to \mathbb{R}_{>0}$ (resp. $f_d : \mathbb{N} \to \mathbb{R}_{>0}$) is a *coordination (resp. anti-*

*coordination) scaling function.* It has minimum $f_c(1) > 0$ (resp. $f_d(0) > 0$), and increases monotonically with the number of agents whose policies match (resp. are different from) $i$'s.

**Definition 5.2** (Coordination game). The agents' return can be decomposed as follows, $\forall i, j \in \{1, \dots, N\}$: $V^i(\boldsymbol{\pi}, \mu_{\bar{t}}) = h\left(b(\pi^i), f_c\left(\sum_{j\in\{1,\dots,N\}} \mathbb{I}\left[\pi^i = \pi^j\right]\right)\right)$, where $h : \mathbb{R}_{\geq 0} \times \mathbb{R}_{>0} \to \mathbb{R}_{\geq 0}$ is a function that composes $b(\cdot)$ and $f_c(\cdot)$ and is monotonic in both arguments, i.e. an increase in either the policy's intrinsic ability to attain rewards, or the extent to which it is aligned with other agents' policies, gives a higher return.

**Definition 5.3** (Anti-coordination game). The agents' return can be decomposed as follows, $\forall i, j \in \{1, \dots, N\}$: $V^i(\boldsymbol{\pi}, \mu_{\bar{t}}) = h\left(b(\pi^i), f_d\left(N - \sum_{j\in\{1,\dots,N\}} \mathbb{I}\left[\pi^i = \pi^j\right]\right)\right)$, where $h : \mathbb{R}_{\geq 0} \times \mathbb{R}_{>0} \to \mathbb{R}_{\geq 0}$ is a function that composes $b(\cdot)$ and $f_d(\cdot)$ and is monotonic in both arguments, i.e. an increase in either the policy's intrinsic ability to attain rewards, or the extent to which it is different from other agents' policies, gives a higher return.

Note that in our setting, where policy parameters are directly communicated and adopted among the population, we focus on exact equality of policies for simplicity of the theory. However, these definitions could be made more general and inclusive by instead considering similarity kernels or label mappings of strategically relevant parts of policies.

## 5.2 Analysis of policy communication

Our sub-routines involve time-varying networks sharing different types of information at different points in the algorithm, meaning that theoretical analysis can potentially grow complicated. We seek to simplify this analysis to make it more intuitive and useful by focusing on the benefit of the decentralised policy exchange scheme in Alg. 3. This is because our ablation studies of Algs. 1 (average reward estimation) and 4 (mean field estimation) in Sec. 6.3.3 indicate that the policy exchange scheme is the dominant factor in driving the benefit of the networked paradigm in our experimental settings. Moreover, recall that Alg. 4 is only necessary when we allow population-dependent policies such that $o_t^i = (s_t^i, \tilde{\mu}_t^i)$, whereas for stationary problems, including all those in our experiments and many others, using a mean field observation or estimation is not actually required for finding the optimal policy. For separate theoretical analysis of Alg. 1 on average reward estimation, please see Sec. 5.3.

For simplicity of the theory, we make several assumptions. We explore the conditions under which these assumptions apply in practice, and discuss how even when loosening the assumptions, they still provide useful heuristic insight as to how our networked communication scheme affords benefits over the central-agent and independent-learning architectures. We do not enforce the assumptions in our experiments, and our empirical results nevertheless follow our theoretical theorems in all but some specific instances that we discuss.

The first assumption formalises this section's focus on the decentralised policy communication scheme, by presuming that it is only the policy communication sub-routine that creates a difference in learning between the networked and central-agent cases: it assumes that the estimated mean fields and average rewards are equivalent to the true ones used in the central-agent case. Note that populations with fully connected networks will in any case always be able to accurately estimate $\hat{r}_t$ and $\hat{\mu}_t$ by Algs. 1 and 4, even for $C_r = 1$ and $C_e = 0$. This is may apply reasonably commonly in practice depending on the scenario; for example, if the network is defined by a broadcast radius (as in our experiments), then the network will be fully connected whenever that radius is at least large enough to cover the area that all the agents fall within. We leave analysis of the theoretical impact of worsening mean-field and average-reward estimates for future work.

**Assumption 5.4.** Assume that Algs. 1 and 4 allow networked agents to obtain accurate estimates of the true population-average rewards and global empirical mean field respectively, i.e. $\forall i \; \tilde{\hat{\mu}}_t^i = \hat{\mu}_t$ and $\tilde{\hat{r}}_t^i = \hat{r}_t$.

Now recall that at each iteration $k$ of Alg. 2, after individually updating their policies in Line 16, the population has the policies $\{\pi_{k+1}^i\}_{i=1}^N$. There is randomness in these individual policy updates, stemming from the random sampling of each agent's individually collected buffer. In Lines 1-7 of Alg. 3, agents estimate the individual infinite-step discounted returns $\{V^i(\boldsymbol{\pi}, \mu_0)\}_{i=1}^N$ (Def. 3.2) of their updated policies

by computing $\{\sigma_{k+1}^i\}_{i=1}^N$: the $E$-step discounted return with respect to the empirical mean field generated when agents follow policies $\{\pi_{k+1}^i\}_{i=1}^N$.

We next assume that the populations' policies are all pair-wise distinct after the updates in Line 16 and before the policy communication. This ensures that policies that are estimated to receive higher returns (and are thus adopted) are being evaluated as higher-performing due to receiving higher base returns, rather than simply because of how aligned or distinct they already happen to be with regard to other policies. This avoids scenarios where, for example, suboptimal policies that are shared across multiple agents after the update (in the case of a coordination game) end up spreading through the population by communication at the expense of a more promising but less common policy, decelerating rather than accelerating improvement. In practice, this assumption is highly likely to apply in most situations in any case. Even if agents start a given iteration with identical policies, their different random seeds are likely to mean that they collect different sample transitions to add to their reinitialised buffers. Even if their buffers happen to end up containing identical transitions, their different random seeds are likely to mean that they sample differently from their buffers, leading to slightly different updates to their policy networks.

**Assumption 5.5.** Assume that directly after the policy updates in Line 16 (Alg. 2), before any policy transfer as in the networked or central-agent algorithms, all policies are pair-wise distinct due to the randomness in these updates, i.e. $\forall i, j \in \{1, \ldots, N\}$ $\pi_{k+1}^i \neq \pi_{k+1}^j$. This means the function $f_c$ attains its minimum $f_c(1)$, and $f_d$ attains its maximum $f_d(N-1)$.

We now assume that the finite-step estimates of the returns give sufficiently accurate comparisons between policies, so that better policies are indeed the ones that get adopted in expectation.

**Assumption 5.6.** Assume that $\{\sigma_{k+1}^i\}_{i=1}^N$ are sufficiently good estimates so as to respect the ordering of the true infinite discounted individual returns $\{V^i(\boldsymbol{\pi}_{k+1}, \mu_0)\}_{i=1}^N$, i.e.

$$V^i(\boldsymbol{\pi}_{k+1}, \mu_0) > V^j(\boldsymbol{\pi}_{k+1}, \mu_0) \iff \sigma_{k+1}^i > \sigma_{k+1}^j \qquad \forall i, j \in \{1, \ldots, N\}.$$

In practice, even if Assumption 5.6 does not strictly hold, the softmax parameter $\tau_k^{comm}$ allows a smooth degradation as the ordering of the approximations worsens with respect to the ordering of the true values. That is, if instead of the exact correct policy ordering we have that better policies are simply *more likely* to be given higher estimated evaluations, then the softmax means that these policies remain *more likely* to spread, and a better policy may still be adopted even if it is not evaluated as being better.

The next assumption presumes that the networked population reaches consensus on a single policy within the policy communication rounds of each $k$ iteration. We use this assumption in only one of our three theorems (Thm. 5.9), and we do so to give general and intuitive comparison with the central-agent population which always shares a single policy. Incomplete consensus would give different levels of alignment/diversity, such that the relative performance of the central-agent and networked architectures might then depend on the specific reward function of the task, and whether base return or alignment/diversity were more important in that reward function.

**Assumption 5.7.** Assume that after the $C_p$ rounds in Lines 8-15 (Alg. 3), in which agents exchange and adopt policies from neighbours, the networked population is left with a single policy such that $\forall i, j \in \{1, \ldots, N\}$ $\pi_{k+1}^i = \pi_{k+1}^j$.

While this may sound like a strong assumption, we phrase it like this so as not to make overly strong restrictions on the communication network instead - we intentionally leave it so that Assumption 5.7 can be fulfilled in numerous ways. Most simply we can think of Assumption 5.7 holding if:

1. we set $\tau_k^{comm}$ close to 0 for all $k$, such that the softmax essentially becomes a max function; and

2. the communication network $\mathcal{G}_t^{comm}$ is static and connected during the $C_p$ communication rounds, where $C_p$ is at least as large as the network diameter $d_{\mathcal{G}_t^{comm}}$.

Under these conditions, previous results on max-consensus algorithms show that all agents in the network will converge on the highest value $\sigma_{k+1}^{max}$ (and hence the associated $\pi_{k+1}^{max}$) within a number of rounds equal

to the diameter $d_{\mathcal{G}_t^{comm}}$ (Nejad et al., 2009; Benjamin & Abate, 2023). If we assumed more strongly that the network was always *fully* connected, policy consensus would be achieved within a single communication round.

Policy consensus can be achieved even outside of these conditions, including if the network is dynamic and not connected at every step. The *union* of a collection of graphs $\{\mathcal{G}_t, \mathcal{G}_{t+1}, \cdots, \mathcal{G}_{t+\omega}\}$ ($\omega \in \mathbb{N}$) is the graph with vertices and edge set equalling the union of the vertices and edge sets of the graphs in the collection (Jadbabaie et al., 2003). A collection is *jointly connected* if its members' union is connected. Now, instead of assuming that the communication network is static and connected, we assume instead only that the sequence of networks contains one or more sequential jointly connected collections. Then max-consensus is reached within $C_p$ if $C_p$ is large enough that the number of sequential jointly connected collections occurring within $C_p$ is equal to the largest diameter of the union of any such collection.

Thus Assumption 5.7 may not hold if $C_p$ is not large enough or if parts of the population remain isolated. However, we do not enforce this assumption in our experiments, where we use $C_p = 1$ to show the benefit of even just one communication round, yet we still see networked populations significantly outperforming central-agent populations across anti-coordination games. In coordination games, while networked populations that are more connected (due to having larger communication radii) usually perform similarly to or better than central-agent populations, those that are less connected occasionally perform less well than the central-agent populations. This is probably due to Assumption 5.7 being empirically more likely to be violated in less connected populations, which in turn is more of an issue in coordination games (where consensus is more likely to be beneficial) than in anti-coordination games (where some lack of consensus does not prevent, or even helps, networked populations to outperform central-agent ones in practice).

The next assumption presumes that if a certain policy, when followed by all members of a finite population, is better than another policy when the latter is followed by all members of a finite population, then the same quality ordering will apply when members of infinite populations follow each policy. We require this in order to relate our analysis of learning in the empirical finite population back to the mean field limit when comparing with central-agent learning. Since the finite population can be arbitrarily large, and in many environments when all agents follow the same policy the individual expected discounted returns averaged across the $N$-agent population will converge smoothly to the infinite population social welfare, this assumption will naturally hold in many scenarios. For example, a policy that is better than another at getting a population of 500 or 5,000,000 agents to cluster in a particular location will also be better than the other policy at getting an infinite population to gather at the location. Nevertheless this order preservation is not a completely general phenomenon, and strict inequalities can vanish or reverse in the limit, especially in models with thresholds or discontinuities in the dependence of rewards or transitions on the mean field, so we state it as an explicit condition in the following.

**Assumption 5.8.** Say we have two different policies that could be shared by the whole population such that $\boldsymbol{\pi}^x = (\pi^x, \ldots, \pi^x)$ and $\boldsymbol{\pi}^y = (\pi^y, \ldots, \pi^y)$. We assume that:

$$V^{pop}(\boldsymbol{\pi}^x, \mu_0) > V^{pop}(\boldsymbol{\pi}^y, \mu_0) \iff W(\pi^x, I(\pi^x)) > W(\pi^y, I(\pi^y)).$$

We have now given all the assumptions for our first theorem. Assumption 5.7 assumes that after the $C_p$ policy exchange rounds in Lines 8-15 of Alg. 3, the networked population is left with a single policy. Call this consensus policy $\pi_{k+1}^{\mathrm{net}}$, and its associated finitely approximated return $\sigma_{k+1}^{\mathrm{net}}$. Recall that the central-agent case is where the Q-network update of arbitrary agent $i = 1$ is automatically pushed to all the others instead of the policy evaluation and exchange in Line 17 of Alg. 2; this is equivalent to a networked case where policy consensus is reached on a *random* one of the policies $\{\pi_{k+1}^i\}_{i=1}^N$. Call this policy *arbitrarily* given to the whole population $\pi_{k+1}^{\mathrm{cent}}$, and its associated finitely approximated return $\sigma_{k+1}^{\mathrm{cent}}$.

We can now give our first theorem under these conditions, namely that in expectation networked populations will increase their returns at least as fast as central-agent ones.

**Theorem 5.9.** *Let us set $\tau_k^{comm} \in \mathbb{R}_{>0}$. In coordination and anti-coordination games where Assumptions 5.4-5.8 apply, we have $\mathbb{E}[W(\pi_{k+1}^{\mathrm{net}}, I(\pi_{k+1}^{\mathrm{net}}))] \geq \mathbb{E}[W(\pi_{k+1}^{\mathrm{cent}}, I(\pi_{k+1}^{\mathrm{cent}}))]$.*

**Remark 5.10.** Assumption 5.5 presumes that all policies are pairwise distinct after the updates, but does not restrict their returns in the same way. If we additionally make the very weak assumption that at least

one of these distinct policies in each $k$ iteration has a base return that is distinct from the others (which is likely to hold in all but the most trivial environments), the inequality in the theorem above will be strict, i.e. *networked learning will always be faster in expectation.*

*Proof.* Recall that before the communication rounds in Line 8 (Alg. 3), the randomly updated policies $\{\pi_{k+1}^i\}_{i=1}^N$ have associated estimated returns $\{\sigma_{k+1}^i\}_{i=1}^N$. Denote the mean and maximum of this set $\sigma_{k+1}^{\text{mean}}$ and $\sigma_{k+1}^{\max}$ respectively. Since $\pi_{k+1}^{\text{cent}}$ is chosen arbitrarily from $\{\pi_{k+1}^i\}_{i=1}^N$, it will obey $\mathbb{E}[\sigma_{k+1}^{\text{cent}}] = \sigma_{k+1}^{\text{mean}}$ $\forall k$, though there will be high variance. Conversely, for the networked case the softmax adoption scheme (Line 11, Alg. 3), which for $\tau_k^{comm} \in \mathbb{R}_{>0}$ gives non-uniform adoption probabilities for distinct $\sigma$ values, means by definition that some communicated policies are more likely to be adopted than others if they have distinct finitely estimated returns (those with higher $\sigma_{k+1}^i$ are more likely to be adopted at each communication round). Thus the consensus $\pi_{k+1}^{\text{net}}$ that gets adopted by the whole networked population will obey $\mathbb{E}[\sigma_{k+1}^{\text{net}}] > \sigma_{k+1}^{\text{mean}}$ if at least one policy receives a distinct return from the others, or $\mathbb{E}[\sigma_{k+1}^{\text{net}}] \geq \sigma_{k+1}^{\text{mean}}$ in the rare circumstance that all policies receive the same return. If $\tau_{k+1}^{comm}$ is close to 0, the consensus policy will obey $\mathbb{E}[\sigma_{k+1}^{\text{net}}] = \sigma_{k+1}^{\max}$ $\forall k$. As such:

$$\mathbb{E}[\sigma_{k+1}^{\text{net}}] \geq \mathbb{E}[\sigma_{k+1}^{\text{cent}}]. \tag{1}$$

In Eq. 1 and the remaining equations of the proof, bear in mind that the equality will be strict if at least one policy receives a distinct return from the others.

Refer to the agent whose update originally gave rise to $\pi_{k+1}^{\text{net}}$ and $\sigma_{k+1}^{\text{net}}$ as agent $(i, \text{net})$; we equivalently also have the arbitrary agent $(j, \text{cent})$. Prior to consensus being attained in each case, the joint policy can be written as $\boldsymbol{\pi}^{(i,\text{net};j,\text{cent})} := (\pi^1, \ldots, \pi^{i-1}, \pi^{(i,\text{net})}, \pi^{i+1}, \ldots, \pi^{j-1}, \pi^{(j,\text{cent})}, \pi^{j+1}, \ldots, \pi^N)$.

Given Eq. 1, and by Assumption 5.6 on the quality of finite-step estimates, we know that directly after the policy update in Line 16 (Alg. 2), *prior to the consensus being reached*, we have:

$$\mathbb{E}\left[V^{(i,\text{net})}(\boldsymbol{\pi}_{k+1}^{(i,\text{net};j,\text{cent})}, \mu_t)\right] \geq \mathbb{E}\left[V^{(j,\text{cent})}(\boldsymbol{\pi}_{k+1}^{(i,\text{net};j,\text{cent})}, \mu_t)\right]. \tag{2}$$

We now need to show that this ordering is maintained in the case that each policy is given to the whole population.

By Assumption 5.5 we know that straight after the random policy updates there is no alignment among policies, i.e. in a coordination task we have $f_c^{(i,\text{net})} = f_c^{(j,\text{cent})} = \min f_c$, and in an anti-coordination task we have $f_d^{(i,\text{net})} = f_d^{(j,\text{cent})} = \max f_d$. Therefore if Eq. 2 pertains, by Def. 5.2 it must be because:

$$\mathbb{E}[b(\pi^{(i,\text{net})})] \geq \mathbb{E}[b(\pi^{(j,\text{cent})})], \tag{3}$$

i.e. because the base policy quality is higher for $\pi^{(i,\text{net})}$ than for $\pi^{(j,\text{cent})}$.

By Assumption 5.7 on policy consensus, we know that in the networked and central-agent cases the joint policies respectively become $\boldsymbol{\pi}^{\text{net}} := (\pi^{\text{net}}, \pi^{\text{net}}, \pi^{\text{net}}, \ldots)$ and $\boldsymbol{\pi}^{\text{cent}} := (\pi^{\text{cent}}, \pi^{\text{cent}}, \pi^{\text{cent}}, \ldots)$. We therefore end up with maximum alignment in both cases, such that $f_c^{\text{net}} = f_c^{\text{cent}} = \max f_c$ in a coordination game, and $f_d^{\text{net}} = f_d^{\text{cent}} = \min f_d$ in an anti-coordination game. Due to this, along with Eqs. 2 and 3, we know

$$\mathbb{E}\left[V^i(\boldsymbol{\pi}_{k+1}^{\text{net}}, \mu_t)\right] \geq \mathbb{E}\left[V^j(\boldsymbol{\pi}_{k+1}^{\text{cent}}, \mu_t)\right]. \tag{4}$$

In turn we have:

$$\mathbb{E}\left[V^{pop}(\boldsymbol{\pi}_{k+1}^{\text{net}}, \mu_t)\right] \geq \mathbb{E}\left[V^{pop}(\boldsymbol{\pi}_{k+1}^{\text{cent}}, \mu_t)\right], \tag{5}$$

which by Assumption 5.8 gives

$$\mathbb{E}[W(\pi_{k+1}^{\text{net}}, I(\pi_{k+1}^{\text{net}}))] \geq \mathbb{E}[W(\pi_{k+1}^{\text{cent}}, I(\pi_{k+1}^{\text{cent}}))],$$

namely the result. $\qquad\square$

We now give results showing that, under these conditions, learning is at least as fast in the networked case than in the independent case - empirically we find networked learning always to be strictly faster. We give separate theorems for coordination and anti-coordination games. Since we cannot necessarily expect the independent agents to share a single policy $\pi_{k+1}$ after the update in each iteration of learning, we give these results in terms of the individual expected discounted returns averaged across the $N$-agent population (Def. 3.3) instead of the single-policy social welfare (Def. 3.5) as before.

Again, we assume for simplicity of the theory that it is only the policy communication scheme that creates a difference in learning between the networked and independent cases, i.e. we assume that networked agents receive the same estimates of the mean field and average reward as independent agents. As mentioned above, our ablation studies suggest policy communication is the dominant factor in our experimental settings anyway, with the estimated mean field not required at all in the broad class of stationary problems. Nevertheless, in practice the networked estimates of the (mean field and) average reward are likely to be substantially better than the independent ones, giving an additional performance increase over the independent case. Thus loosening this assumption is likely to actually enhance the effects identified in the theorems.

**Assumption 5.11.** Assume that the estimated global mean field and average reward in the networked case are the same as the independent case, i.e. $\forall i, j \ \tilde{\hat{\mu}}_t^{(i,net)} = \tilde{\hat{\mu}}_t^{(j,ind)}$ and $\tilde{\hat{r}}_t^{(i,net)} = r_t^i$.

We refer to the joint policy in the networked case after communication round $c$ as $\boldsymbol{\pi}_{k+1,c}^{\text{net}} = \left( \pi_{k+1,c}^{(1,\text{net})}, \ldots, \pi_{k+1,c}^{(N,\text{net})} \right)$, and the joint policy in the independent case as $\boldsymbol{\pi}_{k+1}^{\text{ind}} = \left( \pi_{k+1}^{(1,\text{ind})}, \ldots, \pi_{k+1}^{(N,\text{ind})} \right)$.

We can now give our second theorem, namely that, under these conditions, in expectation networked populations will increase their returns at least as fast as independent ones in coordination games with only a single round of policy communication in each iteration.

**Theorem 5.12.** *Let us again set $\tau_k^{comm} \in \mathbb{R}_{>0}$. In a coordination game, given Assumptions 5.5, 5.6 and 5.11, for $c = 0$, $\mathbb{E}\left[ V^{pop}(\boldsymbol{\pi}_{k+1,c+1}^{\text{net}}, \mu_t) \right] \geq \mathbb{E}\left[ V^{pop}(\boldsymbol{\pi}_{k+1}^{\text{ind}}, \mu_t) \right]$.*

**Remark 5.13.** As mentioned in Rem. 5.10, Assumption 5.5 presumes that all policies are pairwise distinct after the updates, but does not restrict their returns in the same way. If we additionally make the very weak assumption that at least one of the distinct policies in each $k$ iteration has a distinct base return from the others (as is generally likely to be the case), the inequality in the theorem above will be strict, i.e. *networked learning will always be faster in expectation.*

*Proof.* Let us consider two scenarios. Firstly let us imagine that within the communication round, agents swap policies, but no policy drops out of the population, such that if agent $i$ adopts policy $\pi_{k+1}^j$, there exists an agent $i'$ that adopts policy $\pi_{k+1}^i$, and so on. That way we end up with the same policies in the population as before the change, but with each one possibly carried by different arbitrary agents. This is equivalent to if no communication had taken place, meaning that in this scenario $V^{pop}(\boldsymbol{\pi}_{k+1,c+1}^{\text{net}}, \mu_t) = V^{pop}(\boldsymbol{\pi}_{k+1}^{\text{ind}}, \mu_t)$. The mostly likely circumstance for this to occur is when no $\sigma_{k+1}^i$ value is distinct from the others.

Let us now consider an alternative scenario. The softmax adoption scheme (Line 11, Alg. 3), which for $\tau_k^{comm} \in \mathbb{R}_{>0}$ gives non-uniform adoption probabilities for distinct $\sigma$ values, means by definition that some communicated policies are more likely to be adopted than others if they have distinct finitely estimated returns. Thus in expectation the number of distinct policies in the population will decrease if at least one policy has a distinct return from the others (of course, there is a possibility of this still happening even if no policy has a distinct return from the others). Let us start by saying for simplicity that during the first communication round a single $\pi_{k+1,c}^{(j,\text{net})}$ is replaced by $\pi_{k+1,c}^{(i,\text{net})}$, such that for $c = 0$

$$\boldsymbol{\pi}_{k+1,c}^{\text{net}} = \left( \pi_{k+1,c}^{(1,\text{net})}, \ldots, \pi_{k+1,c}^{(\mathbf{i},\text{net})}, \ldots, \pi_{k+1,c}^{(\mathbf{j},\text{net})}, \ldots \pi_{k+1,c}^{(N,\text{net})} \right),$$

$$\text{and} \quad \boldsymbol{\pi}_{k+1,c+1}^{\text{net}} = \left( \pi_{k+1,c+1}^{(1,\text{net})}, \ldots, \pi_{k+1,c+1}^{(\mathbf{i},\text{net})}, \ldots, \pi_{k+1,c+1}^{(\mathbf{i},\text{net})}, \ldots \pi_{k+1,c+1}^{(N,\text{net})} \right).$$

For this to have occurred, we know that

$$\mathbb{E}[\sigma_{k+1,c}^{(i,\text{net})}] > \mathbb{E}[\sigma_{k+1,c}^{(j,\text{net})}],$$

and therefore by Assumption 5.6 that

$$\mathbb{E}\left[V^{(i,\mathrm{net})}(\boldsymbol{\pi}_{k+1,c}^{\mathrm{net}}, \mu_t)\right] > \mathbb{E}\left[V^{(j,\mathrm{net})}(\boldsymbol{\pi}_{k+1,c}^{\mathrm{net}}, \mu_t)\right]. \tag{6}$$

By Assumption 5.5 we know that straight after the random policy updates there is no alignment among policies, i.e. in a coordination game we have $f_c^{(i,\mathrm{net})} = f_c^{(j,\mathrm{net})} = \min f_c$. Therefore if Eq. 6 pertains, by Def. 5.2 it must be because:

$$\mathbb{E}[b(\pi^{(i,\mathrm{net})})] > \mathbb{E}[b(\pi^{(j,\mathrm{net})})], \tag{7}$$

i.e. because the base policy quality is higher for $\pi^{(i,\mathrm{net})}$ than for $\pi^{(j,\mathrm{net})}$. For this reason we have, for $c = 0$:

$$\mathbb{E}\left[V^{pop}(\boldsymbol{\pi}_{k+1,c+1}^{\mathrm{net}}, \mu_t)\right] > \mathbb{E}\left[V^{pop}(\boldsymbol{\pi}_{k+1,c}^{\mathrm{net}}, \mu_t)\right]. \tag{8}$$

Additionally, replacing $\pi_{k+1,c}^{(j,\mathrm{net})}$ with a second copy of $\pi_{k+1,c}^{(i,\mathrm{net})}$ will increase the alignment ($f_c$) of $\pi_{k+1,c}^{(i,\mathrm{net})}$ such that $\mathbb{E}\left[V^{(i,\mathrm{net})}(\boldsymbol{\pi}_{k+1,c+1}^{\mathrm{net}}, \mu_t)\right] > \mathbb{E}\left[V^{(i,\mathrm{net})}(\boldsymbol{\pi}_{k+1,c}^{\mathrm{net}}, \mu_t)\right]$, increasing the improvement even further. This effect is even greater if more than one policy is replaced.

Since the independent case is equivalent to the networked case when $C_p = 0$, we can say that $\boldsymbol{\pi}_{k+1}^{\mathrm{ind}} = \boldsymbol{\pi}_{k+1,0}^{\mathrm{net}}$. This gives the result, i.e.

$$\mathbb{E}\left[V^{pop}(\boldsymbol{\pi}_{k+1,c+1}^{\mathrm{net}}, \mu_t)\right] \geq \mathbb{E}\left[V^{pop}(\boldsymbol{\pi}_{k+1}^{\mathrm{ind}}, \mu_t)\right],$$

where this inequality will be strict if the first scenario does not apply in expectation. $\qquad\square$

To prove the benefit of the networked case over the independent case in anti-coordination games, we use a final additional assumption. This presumes that the base return is not yet fully maximised (as naturally applies for a certain amount of time during training), and that the benefit to an agent's overall return of increasing its base return by adopting a neighbour's better-performing policy outweighs the resulting decrease in diversity. This establishes the conditions under which our policy adoption scheme is able to advantage networked agents over those whose policies are always independent. The second part of the assumption applies in most non-trivial scenarios, namely where the goal of the task is not simply for agents to have distinct policies that are otherwise inconsequential, and thus where the benefit of diverse behaviour can only be fully felt once agents have a certain level of aptitude at accomplishing the given task. For example, in all of the anti-coordination games in our experiments, agents are always penalised for moving, and only start to receive higher rewards if they are stationary. Therefore in these anti-coordination games agents will receive higher returns by *aligning* on policies that prioritise stationarity, than by maintaining diverse policies that have high levels of movement. Of course once base return is maximised and the assumption no longer holds, one can consider terminating policy communication and adoption to avoid decreases in diversity (one may also be ready to stop training entirely at this point, as the population is likely to be reaching the optimal average return). Please see Sec. 6.3.1 for further discussion of the applicability of this assumption in practice.

**Assumption 5.14.** Assume that the agents have not yet maximised their base return function i.e. $b(\pi_{k+1}^i) < \sup_{\pi \in \Pi} b(\pi) \ \forall i \in \{1, \ldots, N\}$, and that an increase in the base return function outweighs a decrease in the policy diversity, namely $h(b + \Delta b, f_d - \Delta f_d) > h(b, f_d), \ \forall \Delta b > 0, \Delta f_d > 0$.

We now give our final theorem, namely that under these conditions in anti-coordination games, in expectation networked populations will increase their returns at least as fast as independent ones with only a single round of communication in each iteration.

**Theorem 5.15.** *Let us once again set $\tau_k^{comm} \in \mathbb{R}_{>0}$. In an anti-coordination game, given Assumptions 5.5, 5.6, 5.11 and 5.14, for $c = 0$, $\mathbb{E}\left[V^{pop}(\boldsymbol{\pi}_{k+1,c+1}^{\mathrm{net}}, \mu_t)\right] \geq \mathbb{E}\left[V^{pop}(\boldsymbol{\pi}_{k+1}^{\mathrm{ind}}, \mu_t)\right]$.*

**Remark 5.16.** As mentioned in Rems. 5.10 and 5.13, Assumption 5.5 presumes that all policies are pairwise distinct after the updates, but does not restrict their returns in the same way. If we additionally make the very weak assumption that at least one of the distinct policies in each $k$ iteration has a distinct base return from the others (as is generally likely to be the case), the inequality in the theorem above will be strict, i.e. *networked learning will always be faster in expectation.*

*Proof.* The proof begins similarly to that for a coordination game. Let us consider two scenarios. Firstly let us imagine that within the communication round, agents swap policies, but no policy drops out of the population, such that if agent $i$ adopts policy $\pi_{k+1}^j$, there exists an agent $i'$ that adopts policy $\pi_{k+1}^i$, and so on. That way we end up with the same policies in the population as before the change, but with each one possibly carried by different arbitrary agents. This is equivalent to if no communication had taken place, meaning that in this scenario $V^{pop}(\boldsymbol{\pi}_{k+1,c+1}^{\text{net}}, \mu_t) = V^{pop}(\boldsymbol{\pi}_{k+1}^{\text{ind}}, \mu_t)$. The mostly likely circumstance for this to occur is when no $\sigma_{k+1}^i$ value is distinct from the others.

Let us now consider an alternative scenario. The softmax adoption scheme (Line 11, Alg. 3), which for $\tau_k^{comm} \in \mathbb{R}_{>0}$ gives non-uniform adoption probabilities for distinct $\sigma$ values, means by definition that some communicated policies are more likely to be adopted than others if they have distinct finitely estimated returns. Thus in expectation the number of distinct policies in the population will decrease if at least one policy has a distinct return from the others. Say for simplicity that during the first communication round a $\pi_{k+1,c}^{(j,\text{net})}$ is replaced by $\pi_{k+1,c}^{(i,\text{net})}$, such that for $c = 0$

$$\boldsymbol{\pi}_{k+1,c}^{\text{net}} = \left( \pi_{k+1,c}^{(1,\text{net})}, \ldots, \pi_{k+1,c}^{(\mathbf{i},\text{net})}, \ldots, \pi_{k+1,c}^{(\mathbf{j},\text{net})}, \ldots \pi_{k+1,c}^{(N,\text{net})} \right),$$

$$\text{and} \quad \boldsymbol{\pi}_{k+1,c+1}^{\text{net}} = \left( \pi_{k+1,c+1}^{(1,\text{net})}, \ldots, \pi_{k+1,c+1}^{(\mathbf{i},\text{net})}, \ldots, \pi_{k+1,c+1}^{(\mathbf{i},\text{net})}, \ldots \pi_{k+1,c+1}^{(N,\text{net})} \right).$$

For this to have occurred, we know that

$$\mathbb{E}[\sigma_{k+1,c}^{(i,\text{net})}] > \mathbb{E}[\sigma_{k+1,c}^{(j,\text{net})}],$$

and therefore by Assumption 5.6 that

$$\mathbb{E}\left[ V^{(i,\text{net})}(\boldsymbol{\pi}_{k+1,c}^{\text{net}}, \mu_t) \right] > \mathbb{E}\left[ V^{(j,\text{net})}(\boldsymbol{\pi}_{k+1,c}^{\text{net}}, \mu_t) \right]. \tag{9}$$

By Assumption 5.5 we know that straight after the random policy updates there is no alignment among policies, i.e. in the anti-coordination game we have $f_d^{(i,\text{net})} = f_d^{(j,\text{net})} = \max f_d$, while by Assumption 5.14 we know that the agents have not yet maximised their base return function. Therefore if Eq. 9 pertains, by Def. 5.2 it must be because:

$$\mathbb{E}[b(\pi^{(i,\text{net})})] > \mathbb{E}[b(\pi^{(j,\text{net})})], \tag{10}$$

i.e. because the base policy quality is higher for $\pi^{(i,\text{net})}$ than for $\pi^{(j,\text{net})}$.

Assumption 5.14 assumes that any increase in the base quality of the policy will outweigh the decrease in diversity that will come from having more than one agent following $\pi_{k+1,c+1}^{(i,\text{net})}$. Therefore we have, for $c = 0$:

$$\mathbb{E}\left[ V^{pop}(\boldsymbol{\pi}_{k+1,c+1}^{\text{net}}, \mu_t) \right] > \mathbb{E}\left[ V^{pop}(\boldsymbol{\pi}_{k+1,c}^{\text{net}}, \mu_t) \right].$$

These steps apply similarly if more than one policy is replaced.

Since the independent case is equivalent to the networked case when $C_p = 0$, we can say that $\boldsymbol{\pi}_{k+1}^{\text{ind}} = \boldsymbol{\pi}_{k+1,0}^{\text{net}}$. This gives the result, i.e.

$$\mathbb{E}\left[ V^{pop}(\boldsymbol{\pi}_{k+1,c+1}^{\text{net}}, \mu_t) \right] \geq \mathbb{E}\left[ V^{pop}(\boldsymbol{\pi}_{k+1}^{\text{ind}}, \mu_t) \right],$$

where this inequality will be strict if the first scenario does not apply in expectation. $\qquad\square$

### 5.3 Analysis of Alg. 1 on local estimation of global average reward

In this section we provide bounds on the error of the estimates computed via Alg. 1 - i.e. the outputs in Line 8 - as well as on the average error of these estimates.

In Line 5 of Alg. 1, agents collect rewards from neighbours that fall within their broadcast radius. By broadcasting and expanding these collections for multiple communication rounds, they can collect rewards from agents $C$ hops away on the network. If the network is connected and $C$ is greater than or equal to the diameter of the network, then agents will be able to collect the rewards of all the agents in the population.

In Line 7, agents compute the exact average of the rewards they have collected. That is, the algorithm allows agents to gain an exact average for however many agents are connected by the number of hops permitted by the number of communication rounds. If this includes the whole population, the agents will accurately compute the true average reward.

We can therefore see that the error relates to how many agents are in disconnected sub-populations such that their reward information is not reached; the error additionally relates to the *relative* average rewards of these sub-populations. We now give several formal results regarding this error.

**Theorem 5.17.** *Say the rewards are normalised between 0 and 1. The error in an individual agent $i$'s estimate of the global average reward has the bounds*

$$0 \;\leq\; |\tilde{\hat{r}}_t^i - \hat{r}_t| \;\leq\; \frac{N-1}{N}.$$

*Proof.* If an agent's local estimate of the global average reward is the same as the true average reward, then the error is of course 0. The worst case error in the average reward estimate is given by the situation when a single agent is isolated from the rest of the population with a reward diametrically opposed to that of the rest of the population. We therefore consider the maximum and minimum rewards in a given environment; we have said for simplicity that the rewards are normalised between 0 and 1. Let us say arbitrarily that the single isolated agent receives a reward of 0, while all $N-1$ other agents receive a reward of 1 (though the maths resolves to the exact same result if these are reversed).

Then we know the true global average reward is

$$\frac{(1 \times 0) + ((N-1) \times 1))}{N} \;=\; \frac{N-1}{N}.$$

The isolated agent would estimate that global average reward is 0, while all the other agents would estimate that global average reward is 1. Thus the error of the isolated agent is

$$\frac{N-1}{N} - 0 \;=\; \frac{N-1}{N},$$

while the error of all the other agents is

$$1 - \frac{N-1}{N} \;=\; \frac{1}{N}.$$

If the isolated agent instead received a reward of 1 and the rest of the population received a reward of 0 the results would be the same. Apart from this, any other change in the reward received by any agent would reduce the error in the isolated agent's estimate. For the original reward assignment, if the isolated agent's reward were any greater than 0 then the average would increase by an amount smaller than the increase in its reward, making the error smaller than before. Equally if any other other agent received a reward less than 1, this would reduce the average reward and in turn also the error of the the isolated agent's estimate (the same logic also applies if the reward attributions are flipped). Equally, if any number of agents receiving reward of 1 formed a connected sub-population with the currently isolated agent, the latter's estimate of the average reward would increase, again reducing the error. Therefore the original scenario gives the largest possible error that an estimate is able to have, providing an error bound on the estimate as in the theorem. □

**Theorem 5.18.** *When the error of any one agent $i$ is at its upper bound as per Thm. 5.17, the average error in the population's estimates is given by*

$$\frac{1}{N} \sum_{i \in \{1,\dots,N\}} |\tilde{\hat{r}}_t^i - \hat{r}_t| \;=\; \frac{2(N-1)}{N^2}.$$

*Proof.* Thm. 5.17 tells us that $(N-1)/N$ is the largest possible error for any one agent. As per the proof of that theorem, in the scenario when this maximum error occurs for one agent, the *average* error in the

population is

$$\frac{\left(\frac{N-1}{N} \times 1\right) + \left(\frac{1}{N} \times (N-1)\right)}{N} =$$

$$= \frac{\left(\frac{1}{N} \times (N-1)\right) + \left(\frac{1}{N} \times (N-1)\right)}{N}$$

$$= \frac{2(N-1)}{N^2}.$$

□

Therefore, in the case when the error of any one agent is at its upper bound, the *average* error is actually small and decreases fast as $N$ increases, making the isolated agent negligible, as one would hope in a mean-field setting.

We finally give a result that imposes bounds on the error averaged across the whole population.

**Theorem 5.19.** *Again let us say that rewards are normalised between 0 and 1. The average error has the following bounds:*

$$0 \leq \frac{1}{N} \sum_{i \in \{1,\dots,N\}} |\tilde{\hat{r}}_t^i - \hat{r}_t| \leq 0.5.$$

*Proof.* The average error is maximised by definition when all estimates are as far away as possible from the true average reward. Rewards (and hence estimates) are normalised to lie in [0,1]. Any value between 0 and 1 is not as far as possible from the average reward, and can be made further from the average reward by pushing to one of the endpoints, i.e. 0 or 1. Thus the highest average error occurs when all estimates take one of the extreme values 0 or 1 and thus is as far as possible from the average reward. In this case the average reward is the fraction $p$ of agents with estimate 1, and the average absolute error is given by $2p(1-p)$, which is maximised when $p = 0.5$, giving a maximum error of 0.5. This means that the maximum average error in the estimate is given when half the population has estimates of 0, and the other half has estimates of 1. The two halves may or may not be connected by communication internally, but there is no communication connection between the halves, as then the estimates of these connected agents would improve, leading to lower errors and therefore also lower average error. □

While the bounds are the same in the independent-learning case, there the worse case scenarios are far more likely to occur. For the networked populations to reach the upper bounds we have specifically constructed situations where communication does not occur between sub-populations in the proofs, whereas this by definition the default in the independent case.

## 6 Experiments

### 6.1 Experimental setup

We present experiments from grid worlds, following the gold standard in similar works on MFGs and MFC (Laurière et al., 2022a). We compare our networked architecture with the central-agent and independent architectures, defined as at the beginning of Sec. 5. All trials are conducted non-episodically, meaning that agents' states are never reset to initial states during learning, and the full learning process is a single rollout/trajectory for each trial.

We give results from six cooperative tasks similar to those found in prior works, defined by the agents' reward/transition functions and relating to agents' positions relative to other agents. Two are coordination tasks and four are anti-coordination tasks, where in each case the reward function reflects a coordination/anti-coordination ($f_c/f_d$) element alongside other elements that may be crucial for receiving reward, reflected in the policies' base quality $b(\pi)$ (Sec. 5). In all cases, rewards are normalised in [0,1] after they are computed, and the cooperative objective is to maximise the social welfare / individual expected discounted returns averaged across the $N$-agent population.

The two coordination tasks are:

- **Cluster.** This game is also used in Benjamin & Abate (2023; 2024). Agents are encouraged to gather together by the reward function $R(s_t^i, a_t^i, \hat{\mu}_t) = \log(\hat{\mu}_t(s_t^i))$. That is, agent $i$ receives a reward that is logarithmically proportional to the fraction of the population that is co-located with it at time $t$. We give the population no indication where they should cluster, agreeing this themselves over time.

- **Target selection.** This game is also used in Benjamin & Abate (2023; 2024). Unlike in the above 'cluster' game, the agents are given options of locations at which to gather, and they must reach consensus among themselves. If the agents are co-located with one of a number of specified targets $\phi \in \Phi$ (in our experiments we place one target in each of the four corners of the grid), and other agents are also at that target, they get a reward proportional to the fraction of the population found there; otherwise they receive a penalty of -1. In other words, the agents must coordinate on which of a number of mutually beneficial points will be their single gathering place. Define the magnitude of the distances between $y, z$ at $t$ as $dist_t(y, z)$. The reward function is given by $R(s_t^i, a_t^i, \hat{\mu}_t) = r_{targ}(r_{coord}(\hat{\mu}_t(s_t^i)))$, where

$$r_{targ}(x) = \begin{cases} x & \text{if } \exists \phi \in \Phi \text{ s.t. } dist_t(s_t^i, \phi) = 0 \\ -1 & \text{otherwise,} \end{cases}$$

$$r_{coord}(x) = \begin{cases} x & \text{if } \hat{\mu}_t(s_t^i) > 1/N \\ -1 & \text{otherwise.} \end{cases}$$

The anti-coordination tasks are:

- **Disperse.** This game is also used in Benjamin & Abate (2024) and is similar to the 'exploration' tasks in Laurière et al. (2022b); Wu et al. (2024) and other MFG works. In our version agents are rewarded for being located in more sparsely populated areas but only if they are stationary, to avoid trivial random policies. The reward function is given by $R(s_t^i, a_t^i, \hat{\mu}_t) = r_{stationary}(-\log(\hat{\mu}_t(s_t^i)))$, where

$$r_{stationary}(x) = \begin{cases} x & \text{if } a_t^i \text{ is 'remain stationary'} \\ -1 & \text{otherwise.} \end{cases}$$

- **Target coverage.** The population is rewarded for spreading across a certain number of targets, as long as agents are stationary at the target. As in the 'target selection' game, we have targets $\phi \in \Phi$, where in our experiments we place one target in each of the four corners of the grid. Again define the magnitude of the distances between $y, z$ at $t$ as $dist_t(y, z)$. The reward function is given by

$$R(s_t^i, a_t^i, \hat{\mu}_t) = r_{stationary}\left(r_{targ}\left(-\log(\hat{\mu}_t(s_t^i))\right)\right),$$

where $r_{stationary}$ and $r_{targ}$ are as defined above.

- **Beach bar.** Such games are very common in MFG works (Perrin et al., 2020; Laurière et al., 2022a; Cui et al., 2023a; Wu et al., 2024). In our version agents are rewarded for being stationary in sparsely populated locations as close as possible to a target $\phi_b$, located in the centre of the grid. The maximum possible distance from the target is denoted $maxDist$. The reward is given by

$$R(s_t^i, a_t^i, \hat{\mu}_t) = r_{stationary}\left(maxDist - dist_t(s_t^i, \phi_b) - \log(\hat{\mu}_t(s_t^i))\right),$$

where $r_{stationary}$ is as defined above.

- **Shape formation.** The population is rewarded for spreading around a ring shape, accomplished by encouraging agents to be dispersed a distance of 3 (chosen arbitrarily to fit the grid) from a centre point $\phi_c$. The reward is given by

$$R(s_t^i, a_t^i, \hat{\mu}_t) = r_{stationary}\left(r_{ring}\left(-\log(\hat{\mu}_t(s_t^i))\right)\right),$$

where $r_{stationary}$ is as defined above, and

$$r_{ring}(x) = \begin{cases} x & \text{if } dist_t(s_t^i, \phi_c) = 3 \\ -1 & \text{otherwise.} \end{cases}$$

In these spatial environments, we choose to define both the communication network $\mathcal{G}_t^{comm}$ and the visibility graph $\mathcal{G}_t^{vis}$ by the physical distance from $i$, though this does not need to be the case. We show plots for various transmission radii, given as fractions of the maximum distance in the grid. Note that the networked population with the largest radius is always fully connected, and therefore these agents are always able to accurately estimate $\hat{r}_t$ and $\hat{\mu}_t$ even for $C_r = 1$ and $C_e = 0$. That is, when we set $C_r = C_e > 0$ their observations are equivalent to those that the central-agent population would receive, albeit that policies are updated and spread differently.

We evaluate our experiments according to a finite-step estimate of the individual expected discounted returns averaged across the $N$-agent population (Def. 3.3) over the $M$ steps within each outer $k$ loop (Line 4, Alg. 2), i.e. $\hat{V}^{pop}(\boldsymbol{\pi}_k, \mu_t; M)$. We run five trials with different random seeds for each experiment, and plot the mean and 95% Student's-$t$ confidence intervals (CI) across the seeds. Random seeds are set in our code in a fixed way dependent on the trial number to allow easy replication of experiments.

For each figure we additionally report Area-Under-Learning-Curve (AUC) statistics in the Appendices (Tables 1-9): a 95%-CI on each architecture's own mean AUC, and a Welch (unpaired) two-sample CI on its mean-AUC difference $\Delta$ from the central-agent baseline. AUC summarises each per-run learning curve as a single scalar, and the Welch comparison pools evidence across all $K$ iterations to test for systematic differences between architectures, complementing the per-iteration variation displayed in the plotted bands. Since the seed count is constant (5) across all populations in every comparison, narrower bands also correspond to lower across-seed sample variance.

Experiments were conducted on a Linux-based machine with 2 x Intel Xeon Gold 6248 CPUs (40 physical cores, 80 threads total, 55 MiB L3 cache). We use the JAX framework to parallelise elements of our code. Our code is included in the publicly available supplementary material for reproducibility. Tables 10-18 in the Appendices report per-run CPU time in seconds for each of the 9 experimental settings $\times$ 6 games $\times$ 7 architectures $\times$ 5 seeds. In the standard setting, per-run CPU time is $\approx 211s$ for the central-agent baseline and $\approx 276s$ for our networked architecture (so decentralised communication imposes a constant-factor overhead of $\approx 1.31\times$), and this overhead is essentially independent of broadcast radius. Increasing the number of communication rounds $C$ scales sub-linearly: $C = 10$ costs $\approx 1.8\times$ the $C = 1$ baseline, and $C = 50$ costs $\approx 8.8\times$ the same baseline, because a communication round accounts for only a fraction of per-step cost. The total across all full experiments (1,890 runs) is $\approx 229$ wall-clock hours, with CPU and wall-clock time agreeing to within 1%.

## 6.2 Hyperparameters

See Table 20 in the Appendices for our hyperparameter choices. We can group our hyperparameters into those controlling the size of the experiment, those controlling the size of the Q-network, those controlling the number of iterations of each loop in the algorithms and those affecting the learning/policy updates or policy adoption.

In our experiments we generally want to demonstrate that our communication-based algorithm learns faster than the central-agent and independent architectures, even when the Q-function / mean field / average reward are poorly estimated as is likely to be the case in complex real-world scenarios. There is a similar motivation in the related works on networked communication in the MFG setting by Benjamin & Abate (2023; 2024). Moreover we want to show that there is a large benefit even to a small amount of communication, so that communication rounds themselves do not excessively add to time complexity. As such, we generally select hyperparameters at the lowest end of those we tested during development, to show that our algorithms are particularly successful and robust given what might otherwise be considered 'undesirable' hyperparameter choices.

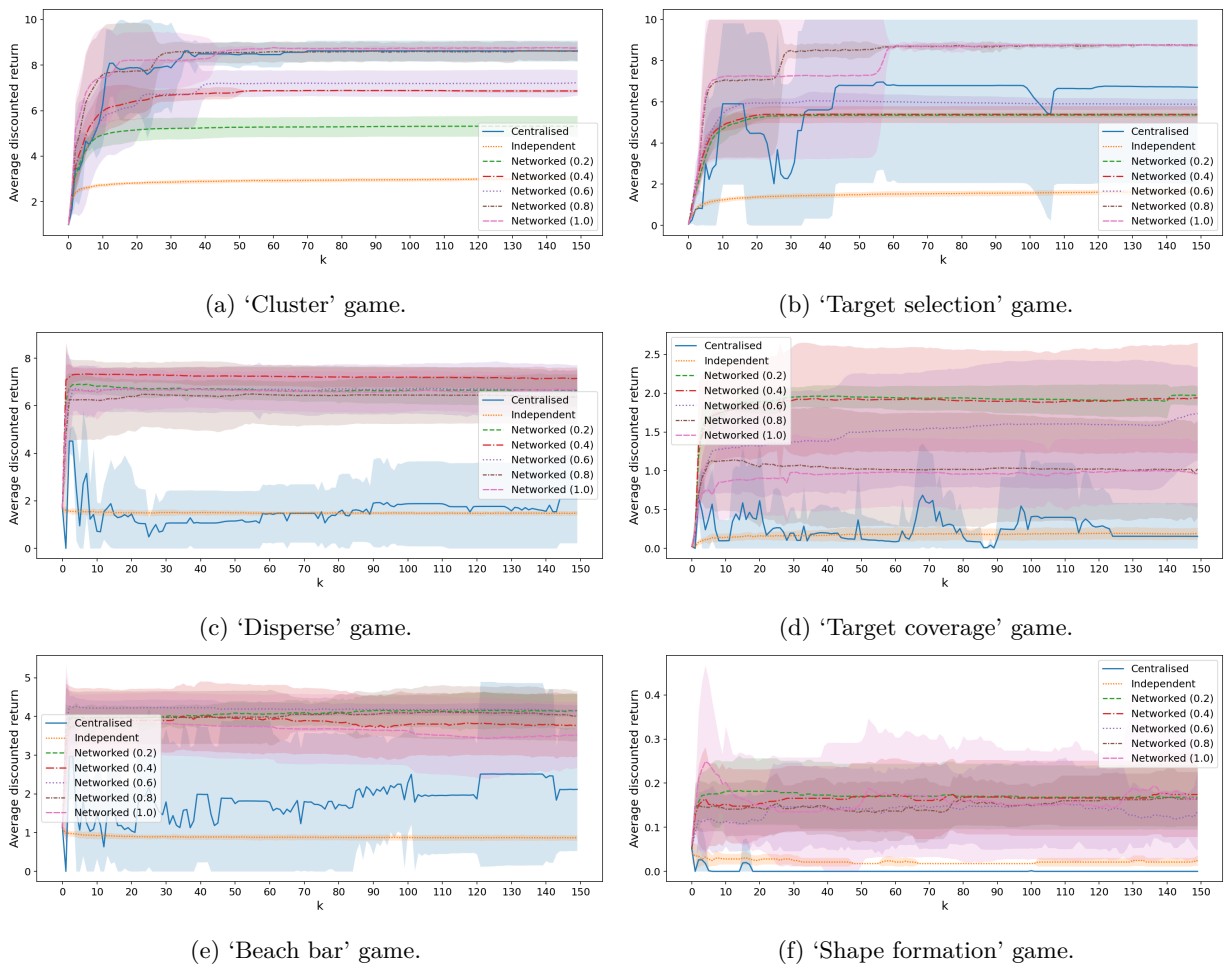

(a) 'Cluster' game.

(b) 'Target selection' game.

(c) 'Disperse' game.

(d) 'Target coverage' game.

(e) 'Beach bar' game.

(f) 'Shape formation' game.

Figure 2: Standard settings with $C_e = C_r = C_p = 1$. In all games networked agents of all broadcast radii significantly outperform the independent (orange) populations, and in most games they also outperform the central-agent (blue) populations, reflecting our theoretical results. The central-agent populations also have markedly higher variance than networked ones in several games, since the central learner pushes an arbitrary updated policy to the whole population regardless of its quality, leading to large fluctuations in performance, whereas our communication scheme biases networked populations towards better performing updates.

## 6.3 Results and discussion

### 6.3.1 Standard setting

Fig. 2 gives results for our standard experimental settings involving 500 agents, each with their own Q-network. When networked agents communicate, they have only a *single* communication round. We do not enforce the simplifying theoretical assumptions from Sec. 5.2 - of particular note, in these standard settings we allow Algs. 1 and 4 to run normally, without assuming that they give perfect estimates of the global average reward or mean field. In Sec. 6.3.3 we ablate these sub-routines, which suggests that the communication of *policies* is the factor with the most influence on learning speed.

Fig. 2 shows that in all of our tasks, networked populations of all broadcast radii significantly outperform independent (orange) agents, which hardly appear to increase their returns, if at all. Networked populations of all broadcast radii also appear to outperform the central-agent (blue) populations in all but the two coordination tasks, where only networked agents of the smaller radii (green, 0.2; red, 0.4; purple, 0.6) underperform them (due to these less connected populations being more likely to experience violations of

Assumption 5.7 on policy consensus, which is a disadvantage in scenarios where alignment is beneficial). In the anti-coordination tasks the central-agent populations perform similarly to purely independent ones in hardly appearing to increase their returns, performing even worse than independent agents in the 'shape formation' task.

These observations are corroborated by the Welch (unpaired) AUC differences in Table 1: larger broadcast radii (0.8 and 1.0) beat the central-agent baseline in mean AUC across all six games, with 95%-CIs excluding zero and $p < 0.05$ in all the anti-coordination games. As above, in the 'cluster' game, radii 0.2, 0.4 and 0.6 are all significantly worse than the central-agent baseline ($\Delta = -2.99 \pm 0.56, -1.53 \pm 0.50, -1.32 \pm 0.62$ respectively; all $p < 0.005$), reflecting the extent to which less-connected populations satisfy Assumption 5.7 on policy consensus, the violation of which is a disadvantage in coordination tasks where alignment is beneficial. The independent baseline is dominated by all other architectures in every game (Welch mean $\Delta$ of networked $-$ independent aggregated across all settings: $+3.03$; 251/270 pairwise comparisons significant at $p < 0.05$, none significantly negative).

In Fig. 2 the central-agent populations also have markedly higher variance than networked ones in several tasks ('target selection', 'disperse', 'beach bar'). While we do not empirically enforce the theoretical assumptions from Sec. 5, this empirical across-seed asymmetry is consistent with the variance asymmetry in our proof of Theorem 5.9: the central learner pushes an arbitrary updated policy to the whole population regardless of its quality (formally, the central-agent estimator $\sigma_{k+1}^{\text{cent}}$ is unbiased but high-variance), leading to large fluctuations in performance, whereas our communication scheme biases networked populations towards better-performing updates (formally, $\sigma_{k+1}^{\text{net}}$ is upward-biased via the softmax-weighted adoption of higher-$\sigma$ alternatives broadcast by neighbours). We do not claim a tight equivalence between these two asymmetries: the theorem bounds the per-step estimator, whereas the empirical bands aggregate across-seed variation in the full RL trajectory, and these can diverge through other sources of seed sensitivity (exploration, gradient paths). Our formal ordering claim regarding networked versus central-agent populations is therefore carried by the Welch AUC differences in Table 1, with our point here about the wide confidence bands being only that their shape is not evidence against that ordering, and is in fact consistent with it.

In the 'target coverage' task, and sometimes the other anti-coordination tasks to a lesser extent, networked agents of smaller broadcast radii appear to outperform those of larger radii, i.e. the ordering is reversed from that of the coordination tasks, albeit not necessarily significantly so. This reflects the point up to which our Assumption 5.14 (base return is not yet maximised, and increase in base return outweighs decrease in diversity in anti-coordination tasks) holds in practice, which we discuss in the following.

The second part of Assumption 5.14 strictly holds throughout the 'disperse', 'target coverage' and 'shape formation' anti-coordination tasks: agents get no reward for diversity unless they are stationary (and also unless they are in one of the correct locations in the latter two cases). This means that any increase in base return (likelihood of being stationary or in the right location) achieved by policy adoption does indeed outweigh the loss of diversity. The second part of Assumption 5.14 mostly holds in the 'beach bar' game, apart from in a small window for agents that are stationary close to the bar target, with the window defined by the size of the empirical population and hence the potential magnitude of the $\log(\hat{\mu}_t(s_t^i))$ term in the reward function. Inside this window, increasing base return by moving even closer to the target, at the cost of being in a more crowded area, would not necessarily be beneficial. Regardless, in all of these tasks the networked populations of all broadcast radii significantly outperform the independent agents, which do not appear to be able to learn at all without the helpful bias towards policies with better base returns enabled by the communication scheme.

However, among these networked populations, the base return quickly reaches its capacity, i.e. agents learn to be primarily stationary in one of the right locations, such that the first part of Assumption 5.14 no longer holds. This is not an issue when comparing with the independent populations, which have not maximised their base returns and therefore perform worse, but it does give rise to the reverse ordering of returns which we see among networked populations of different radii and hence connectivities. Once base return is maximised, policies that are estimated to receive higher returns in these anti-coordination tasks may be less aligned with other policies than those other policies are with each other (at least regarding the strategically relevant parts of policies which are rewarded for greater diversity, e.g. these policies visit the less congested locations),

or they simply visited the less congested locations by chance during the finite evaluation steps. Either way, more adoption of policies now becomes a disadvantage, since it reduces diversity without an additional positive impact on base return. Therefore architectures that give less communication now perform better by preserving diversity. Populations with lower broadcast radii usually have less connected networks, especially if sub-populations become isolated from each other, which is more likely in our 'target coverage' game than the others since the target locations are as far apart as possible from each other. Therefore these populations have less communication than those with larger broadcast radii and so may perform better, even while all networked populations outperform the independent agents that have not maximised their base returns.

This intuition also gives additional justification for why central-agent populations markedly underperform networked populations in these anti-coordination games, especially when policy consensus is not enforced for the networked populations. The ultimate choice of consensus level might depend on the considerations from Rem. 3.8: namely, whether one is using the empirical population as a practical way of learning the social optimum for a MFC problem (Def. 3.6), where a single policy $\pi^*$ is desired to be given to an infinite population, or whether one is solving the MFC problem to approximate the solution to a finite-agent control problem (maximising Def. 3.3) involving the same number of agents as the empirical population from which one is learning. In the latter case some policy diversity may be accepted/desired if it affords a better approximation to the $N$-agent solution.

### 6.3.2 Studies on communications rounds and failures

We conduct two sets of experiments regarding the amount of communication; please find the plots in the Appendices.

**Robustness to communication failures - Fig. 3.** All communication links suffer a 90% probability of failure, including in the central-agent case, where the link between the central learner and the rest of the population may fail.

The central-agent population, which in the standard setting matched networked performance only in the 'cluster' game, now learns slower even in this game, due to suffering from the single point of failure. Our networked scheme appears robust to the failures in all tasks, with only small differences compared to performance in the standard setting. In fact, several broadcast radii appear to perform better in the 'shape formation' game with these failures than without (though not significantly so), probably because the reduced communication permits greater diversity in policies while still having an advantage over purely independent learners (as discussed in Sec. 6.3.1). However, the smallest broadcast radius (green, 0.2) does drop in performance in this game, which might be expected given it now acts similarly to the independent case. Networked populations appear to have less variance in this setting than in the standard setting, at least in the first four games. This is possibly because the communication failures prevent both particularly high- and particularly low-performing policies from spreading fast in the population, preventing large performance fluctuations and smoothing learning progress. Meanwhile central-agent populations still have large variance even with communication failures, due to enforcing the adoption of an arbitrarily-chosen consensus policy - in some games variance is higher in this setting (though in some it may be marginally lower). This points to an additional benefit of our networked scheme over the central-agent case.

The AUC statistics in Table 2 reflect these findings. The networked architectures are essentially unaffected by the failures: averaged across all networked radii and games, networked AUC shifts by only $-0.03$ between standard and failure settings (paired $t$ across the 30 cell-means: $p = 0.67$, i.e. no detectable effect of failures on networked performance). Central-agent AUC, by contrast, falls by 0.97 in the 'cluster' game ($8.17 \rightarrow 7.20$) and by 1.32 in the 'beach bar' game. In the 'cluster' game, where the central-agent population in the standard setting had matched the networked populations at the larger radii (e.g. at radius 1.0: $\Delta = +0.25 \pm 0.51$, $p = 0.28$), this opens a significant gap at radius 1.0 ($\Delta = +1.30 \pm 1.07$, $p = 0.028$, Welch). In the 'beach bar' game, the networked-vs-central gap at radius 1.0 also widens significantly ($\Delta = +1.79 \rightarrow +3.36$; $\Delta\Delta = +1.57 \pm 1.49$, $p = 0.041$). This is consistent with our argument that networked communication provides redundancy that the central learner lacks.

**Increased communication rounds - Figs. 4 and 5.** We show experiments with $C_e = C_r = C_p = 10$ and $C_e = C_r = C_p = 50$.

With both 10 and 50 communication rounds, *all* networked populations receive higher returns than the independent agents in all games, and also than the central-agent population in all but the 'cluster' game.

As is intuitive when moving from 1 to 10 communication rounds, in the coordination games the networked agents with lower broadcast radii now receive returns almost as high as those with larger radii, albeit at the cost of greater variance (having more communication rounds leads to greater policy consensus in the population at each iteration of the outer loop, and there may be some noise in the quality of these consensus policies). In the 'target selection' game, unlike in the standard setting all networked populations now appear to outperform the central-agent (blue) population, though again with high variance. In the anti-coordination 'target coverage' game, the smaller broadcast radii (green, 0.2; red, 0.4; purple, 0.6) receive slightly lower returns than before, since the additional communication rounds now make policy alignment more likely, reducing $f_d$ as per Def 5.3. The same is true of the smallest radius population (green, 0.2) in the 'shape formation' game, which receives a lower return than before. This reflects the discussion in Sec. 6.3.1 regarding the detrimental effect of additional policy adoption once the maximum base return has been achieved in anti-coordination games.

Having 50 communication rounds does not appear to significantly change networked performance compared to 10 rounds, with most increases or decreases in average return appearing within the margin of error. Most notably, the largest broadcast radius (pink, 1.0) receives slightly lower return now than with 10 rounds in the 'disperse' game, while pink (1.0), brown (0.8) and green (0.2) receive lower returns and have higher variance now in the 'beach bar' game. These results suggest that in our experimental settings there is a very large benefit to a single communication round, with limited benefit to increasing the algorithms' time complexity with additional communication rounds. Per-run CPU time grows from $\sim 1.83\times$ the $C{=}1$ baseline at $C{=}10$, and to $\sim 8.77\times$ the $C{=}1$ baseline at $C{=}50$ (Tables 12, 13).

This is again reflected in the AUC, where across all radii the mean $\Delta$ vs. the central-agent population rises only modestly with additional communication rounds ($+1.34$ at $C{=}1$, $+1.71$ at $C{=}10$, $+1.74$ at $C{=}50$; Tables 3, 4). The gain concentrates on the coordination games, where it is most impactful that Assumption 5.7 on policy consensus is easier to satisfy as $C$ increases. For the networked population with radius 1.0 in the 'cluster' game, what was a non-significant Welch gap in the standard setting ($\Delta = +0.25 \pm 0.51$, $p = 0.28$) opens to $\Delta = +0.90 \pm 0.66$ ($p = 0.019$) at $C{=}10$. For this population in the 'target selection' game, the gap grows from $\Delta = +2.07 \pm 3.83$ ($p = 0.22$) at $C{=}1$, to $\Delta = +4.46 \pm 4.25$ ($p = 0.043$) at $C{=}50$.

### 6.3.3 Ablations

We provide ablations of various parts of our algorithms; please find the plots in the Appendices. In all of these scenarios there is only one communication round as relevant, i.e. $C_e = C_r = C_p = 1$ (though one of these loops may be being ablated in each case).

Of particular note, the ablations of Algs. 1 (estimating global average reward) and 4 (estimating global empirical mean field) suggest that:

- Since all of our games have stationary solutions, observing the mean field is not actually necessary. This sub-routine serves more for generality.

- In our experimental settings the policy communication scheme (Alg. 3) is the dominant factor in the better performance of networked populations over the other architectures.

We elaborate on these findings below.

**Ablation study with population-independent policies - Fig. 6.** No agents, including centralised and networked ones, observe or estimate the empirical mean field, and all receive a vector of zeros in its place (so as to keep the neural networks the same size as in the standard setting).

Networked populations do not appear to perform substantially differently to the standard population-dependent setting, though some radii (red, 0.4; pink, 1.0) appear to perform slightly better in the 'shape formation' game. This is likely because all of our games have stationary solutions, such that observing the mean field is not actually necessary, even if it could potentially be useful (see Sec. 3.1 for discussion of the conception of MFC as a central planner trying to guide the population to a distribution that maximises the expected return). Indeed, in the coordination games, and particularly the 'target selection' game, the central-agent population receives a lower return in this setting, whereas our networked populations are more robust to this change.

The Welch AUC differences in Table 5 corroborate this robustness. Averaged across all networked radii and games, networked AUC shifts by only $-0.04$ between the standard setting and this ablation, while central-agent AUC drops by 0.88, widening the mean $\Delta$ vs. central-agent from $+1.34$ to $+2.18$ across all radii, and from $+1.70$ to $+2.54$ at radii $\geq 0.8$ (across all radii: 21/30 significantly positive at $p < 0.05$, 1 significantly negative). In the 'target selection' game in particular, the standard-setting Welch gap at radius 1.0 ($\Delta = +2.07 \pm 3.83$, $p = 0.22$) opens to $\Delta = +6.67 \pm 3.29$ ($p = 0.005$) under the ablation, and the change-in-$\Delta$ itself is significant ($\Delta\Delta = +4.60 \pm 4.32$, $p = 0.039$).

**Ablation study of Alg. 4 for estimating the empirical mean field - Fig. 7.** All agents, including independent ones, directly receive the true global empirical mean field.

This does not appear to change performance in the networked populations (apart from greater variance here in the 'shape formation' game), nor does it help independent agents. This may be evidence that Alg. 4 enables networked agents to accurately estimate the global mean field from local observations. However, our ablation study on population-independent policies above (Fig. 6) suggests that not observing the mean field does not markedly disadvantage agents in our experimental settings in any case (apart from for the central-agent populations in the coordination games). This is likely because all of our games have stationary solutions, such that observing the mean field is not necessary. Therefore further evidence is perhaps needed in MFC settings that require population-dependent policies, in order to confirm the efficacy of Alg. 4 for estimating the mean field, though Benjamin & Abate (2024) has already showed this for non-stationary games in the non-cooperative MFG setting.

This is borne out in the AUC analysis: averaged across all networked radii and games, networked AUC shifts by only $-0.02$ between the standard setting and this ablation (Table 6), consistent with Alg. 4 already letting networked agents estimate the global mean field accurately enough that direct access to the true mean field does not improve performance. However, as noted above, this may also reflect that observing the mean field is not strictly required for our stationary games.

**Ablation study for observation of true/estimated average reward $\hat{r}_t / \tilde{\hat{r}}_t^i$ - Fig. 8.** All agents, including centralised ones, only have access to their individual rewards $r_t^i$, where in the central-agent case $i = 1$.

The greatest effect of this is on the central-agent (blue) populations, which perform much worse in the 'target selection' game, and with higher variance in the 'cluster' and 'beach bar' games, i.e. they suffer without access to the global average reward. The networked agents appear more robust to the loss of the (estimated) average reward, pointing to an additional benefit of the policy communication scheme, though do experience a slight performance decrease, mostly among populations with the largest broadcast radii (pink, 1.0; brown, 0.8), i.e. those most similar to the central-agent case in terms of $\tilde{\hat{r}}_t^i$, as might be expected. In particular, note the greater variance of pink (1.0) in the 'target selection' game; slower learning and higher variance of pink (1.0) and brown (0.8) in the 'beach bar' game; lower returns for pink (1.0) and brown (0.8) in the 'shape formation' game; and slower learning and convergence of the smallest radii (green, 0.2; red, 0.4) in the 'target coverage' game. This all demonstrates the usefulness and efficacy of our novel Alg. 1 for decentralised estimation of the global average reward.

Again, this advantage shows up in AUC as well, where removing the average-reward signal drops central-agent AUC by 0.69 on average across games but networked AUC, averaged across all radii and games, by only 0.13, widening the mean $\Delta$ vs. central-agent from $+1.34$ to $+1.91$ across all radii, and from $+1.70$ to

+2.19 at radii $\geq 0.8$ (Table 7; across all radii, 20/30 significantly positive at $p < 0.05$, only 1 significantly negative). In the 'target selection' game in particular, where the standard setting had not shown a significant Welch gap (radius 0.8: $\Delta = +2.28 \pm 3.87$, $p = 0.18$; radius 1.0: $\Delta = +2.07 \pm 3.83$, $p = 0.22$), removing the average-reward signal opens significant gaps at both radii ($\Delta = +5.89 \pm 3.11$, $p = 0.006$ and $\Delta = +5.28 \pm 3.12$, $p = 0.006$ respectively).

**Ablation study for Alg. 1 for estimating the true global average reward - Fig. 9.** All agents, including both networked and independent ones, directly receive the true global average reward such that $\tilde{\hat{r}}_t^i = \hat{r}_t$.

Access to the true average reward does not help networked agents to improve their returns, demonstrating that our novel Alg. 1 already affords networked populations robustness against the lack of access to this global information (having this global information would be an unrealistic assumption in practice). Access to the true average reward also does not help independent agents to improve their returns, suggesting that the *policy* communication scheme is the dominant factor in improving the performance of decentralised agents.

The AUC analysis supports this. Giving networked populations direct access to the true global average reward actually reduces their AUC, averaged across all radii and games, but only by $-0.08$. This shift is smaller in magnitude than the seed-level noise observed in the central-agent runs themselves (mean shift $-0.23$ across games), even though the central-agent baseline is by construction unaffected by this ablation. Thus the networked shift is indistinguishable from zero, and Alg. 1 appears already to be providing a sufficiently accurate decentralised estimate. The mean $\Delta$ vs. central-agent accordingly holds at $+1.50$ across all radii and $+1.95$ at radii $\geq 0.8$, while it is $+1.34$ and $+1.70$ in the standard setting (Table 8; across all radii, 20/30 significantly positive at $p < 0.05$ in both settings, 1 significantly negative under this ablation as opposed to 3 in the standard setting). Independent AUC also barely moves (mean shift $-0.02$ across games), indicating that access to (estimates of) the average reward is not what separates networked from independent populations.

**Ablation study of the choice of $\tau_k^{comm}$ - Fig. 10.** Here $\forall k$ $\tau_k^{comm} = $ 1e-18 (i.e. $\tau_k^{comm}$ is close to 0, turning the softmax into a max function), rather than linearly increasing from 0.001 to 1 across the $K$ iterations as in all other experiments (see Table 20).

In this setting, networked agents continue to outperform the central-agent (blue) and independent (orange) populations in all games except the 'cluster' game, but otherwise generally appear to receive lower average returns than before and with greater variance. This is because Assumption 5.6 on the quality of the finite-step approximations $\{\sigma_{k+1}^i\}_{i=1}^N = \{\hat{V}^i(\boldsymbol{\pi}_{k+1}, \mu_t; E)\}_{i=1}^N$ may not always apply in practice, especially as the difference between updated policies becomes less stark once they are closer to convergence. This means the policy estimated to perform the best may not actually be among the best updates, such that enforcing the adoption of this policy can lead to noisy, unstable learning. Using a higher temperature value smooths out this noise. On the other hand, having a lower temperature ensures faster learning at the beginning of training when the difference in the quality of nascent policies is likely to be more stark, hence our inverse annealing scheme. Moreover, using $\tau_k^{comm}$ close to 0 more effectively enforces consensus on a single policy in the networked case, which in anti-coordination games may also reduce the average return (see the body of Sec. 6.3.1). This all provides empirical support for our inverse annealing scheme for $\tau_k^{comm}$, but further optimising the choice might lead to additional performance increase.

AUC corroborates this again. The networked architectures still outperform the central-agent baseline (Table 9: mean $\Delta = +1.33$ across all radii, $+1.53$ at radii $\geq 0.8$; across all 30 cells, 18 significantly positive at $p < 0.05$, 2 significantly negative), but by less than under our chosen annealed $\tau_k^{comm}$ schedule: networked AUC, averaged across all radii and games, drops by 0.69 between standard and ablation, narrowing the mean $\Delta$ vs. central-agent from $+1.34$ to $+1.33$ across all radii and from $+1.70$ to $+1.53$ at radii $\geq 0.8$. This supports our choice of $\tau_k^{comm}$ schedule in the main algorithm.

# 7    Conclusion

We provided the first algorithms for decentralised training in MFC, as well as the first for online learning in MFC from a single non-episodic run of the empirical system. We did so by modifying existing algorithms for the MFG setting, and contributing a novel algorithm for estimating the global average reward via local communication. We proved theoretically that, under certain simplifying assumptions, networked communication of policies can accelerate learning over both the independent and central-agent architectures. We also proved bounds on the individual and average error of local estimates of the global average rewards. Without enforcing the simplifying theoretical assumptions, we demonstrated empirically that our algorithms allow networked agents to learn faster than both the independent and central-agent alternatives - significantly so (Welch unpaired two-sample tests on AUC at $p < 0.05$) in 172/270 comparisons across all radii against the central-agent baseline (69/108 at radii $\geq 0.8$, with no significant losses there), and in 251/270 comparisons across all radii against the independent baseline (100/108 at radii $\geq 0.8$) - and we discussed the empirical effects of different communication radii. Our ablations of the sub-routines of our communication algorithm suggest that, in our stationary experimental settings, estimation of the mean-field may be unnecessary, and communication of policies has more impact on learning speed than does communication-based estimation of global average rewards.

## 7.1    Future work

Our work follows the gold standard in MFGs by presenting experiments on grid world toy environments. Nevertheless future work includes experiments in other games, including non-stationary games, more realistic environments and ones where both the transition function and the reward function depend on the mean field. Please note, however, that Benjamin & Abate (2024) already demonstrated in the MFG setting that the communication scheme affords faster learning when both the transition and reward functions depend on the mean field.

Although our Welch AUC tests already establish statistical significance for our networked-vs-baseline comparisons, the per-iteration 95% confidence intervals on the learning-curve plots often remain visibly wide with the 5 trials we have run. Future work could run the experiments with more seeds to narrow these bands further, although since pointwise halfwidths scale by $1/\sqrt{n}$, we expect additional seeds to narrow the bands and push some currently non-significant AUC cells into significance, without altering the direction or significance of the cells that already support our claims.

In Sec. 5 we give theoretical results showing that our networked algorithm can outperform the central-agent and independent alternative. We leave more general theoretical results, such as proofs of convergence and sample complexity, for future work.

Our algorithms contain numerous inner loops and thus require synchronisation between communicating agents. Our ablation studies of the sub-routines and our experiment on robustness to communication failures (Fig. 3) indicate that synchronisation failure is not necessarily a problem in practice, but future work nevertheless lies in simplifying the nested loops of our algorithms.

In grid-world settings such as those in our experiments, passing the (estimated or true global) mean-field distribution as a flat vector to the Q-network ignores the geometric structure of the problem. Perrin et al. (2022) therefore proposes to create an embedding of the distribution by first passing the vector to a convolutional neural network, essentially treating the categorical distribution as an image. This technique is also followed in Wu et al. (2024) (for their additional experiments, but not in the main body of their paper). As future work, we can test whether such a method increases the usefulness of observing the mean field in population-dependent policies, and therefore increases the importance of being able to accurately estimate the global mean field via Alg. 4.

**Broader Impact Statement**

Our work develops decentralised learning and communication mechanisms for large populations of cooperative agents, motivated by applications such as robotic swarms and distributed control systems. While we do not identify immediate risks in our experimental setting, we acknowledge potential dual-use considerations.

In particular, improved decentralised coordination could be applied in safety-critical or adversarial contexts, including large-scale autonomous systems such as robotic or drone swarms (e.g. in defence or surveillance settings). In such systems, faster and more robust collective behaviour could be undesirable if learned policies are misaligned with safety objectives or deployed without sufficient oversight, potentially leading to unsafe system-level outcomes.

Our work is primarily theoretical and evaluated in simplified environments, and does not address issues such as safety constraints, robustness to adversarial agents, or formal verification. Future work could incorporate these considerations to mitigate such risks. Overall, we view this work as a step towards more scalable multi-agent learning, and encourage further research into ensuring such systems are deployed safely and responsibly.

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

# Appendices

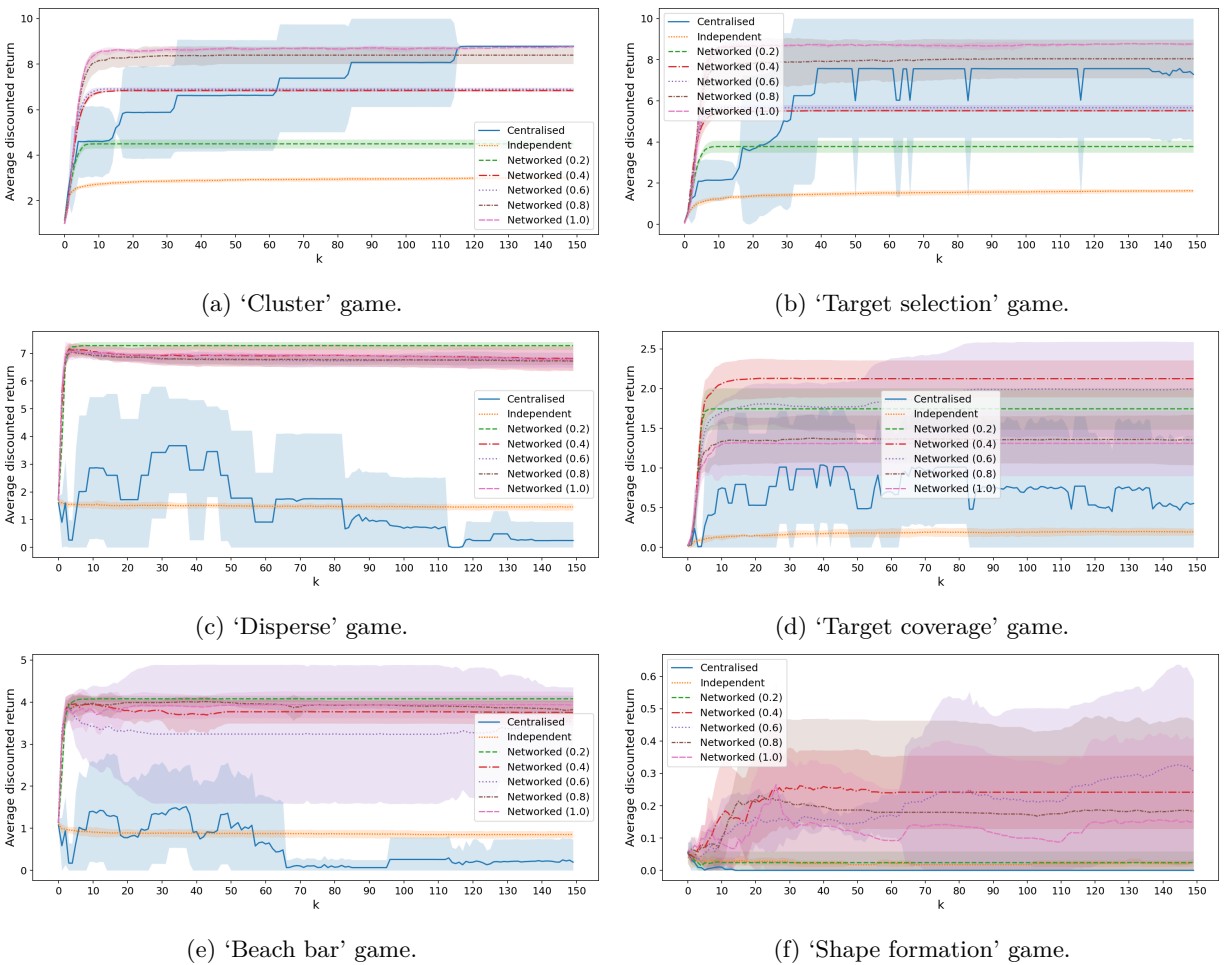

(a) 'Cluster' game.

(b) 'Target selection' game.

(c) 'Disperse' game.

(d) 'Target coverage' game.

(e) 'Beach bar' game.

(f) 'Shape formation' game.

Figure 3: All communication links suffer a 90% probability of failure, including in the central-agent case, where the link between the central learner and the rest of the population may fail. $C_e = C_r = C_p = 1$. Discussed in Sec. 6.3.2.

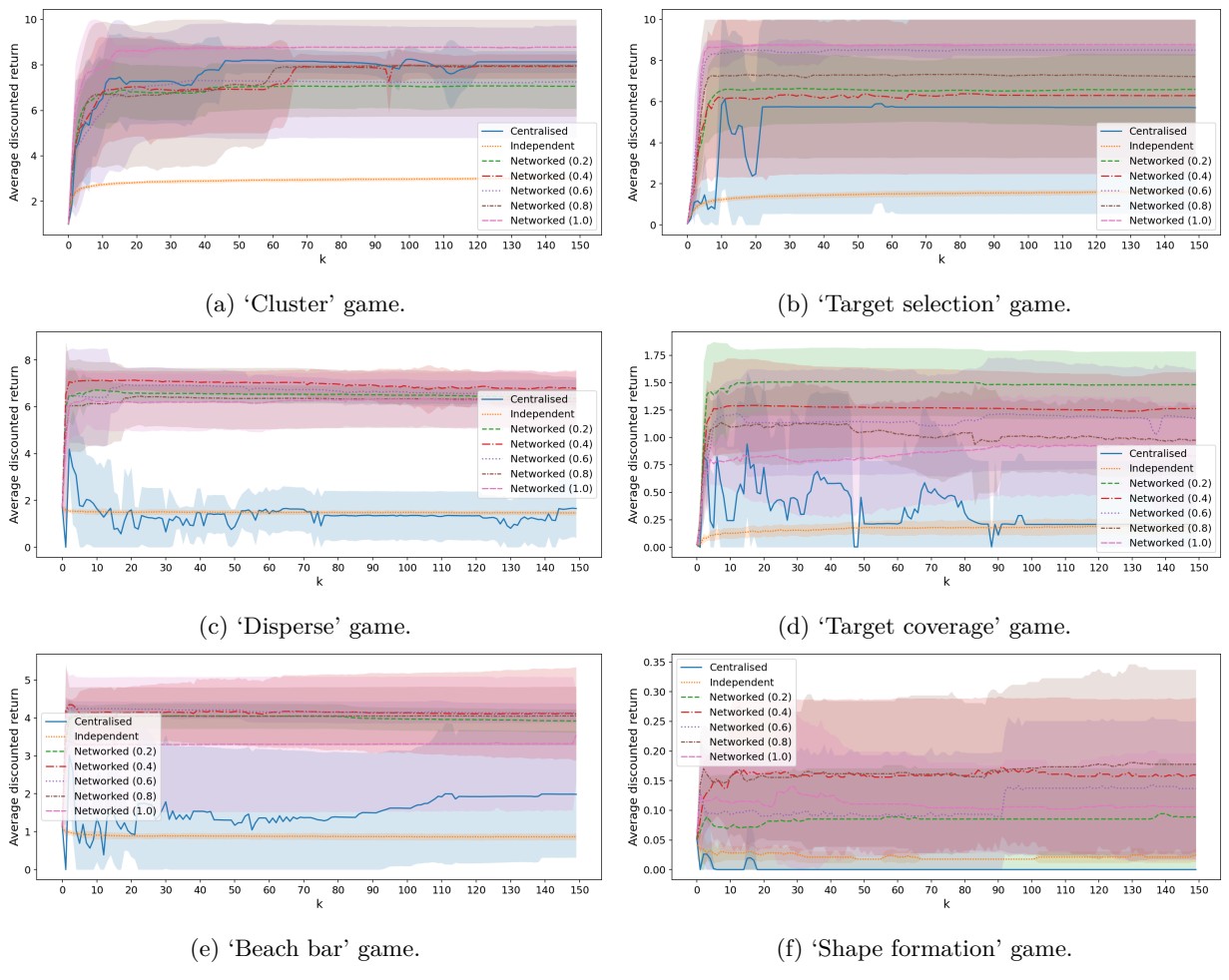

(a) 'Cluster' game.

(b) 'Target selection' game.

(c) 'Disperse' game.

(d) 'Target coverage' game.

(e) 'Beach bar' game.

(f) 'Shape formation' game.

Figure 4: Standard algorithms but $C_e = C_r = C_p = \mathbf{10}$. Discussed in Sec. 6.3.2.

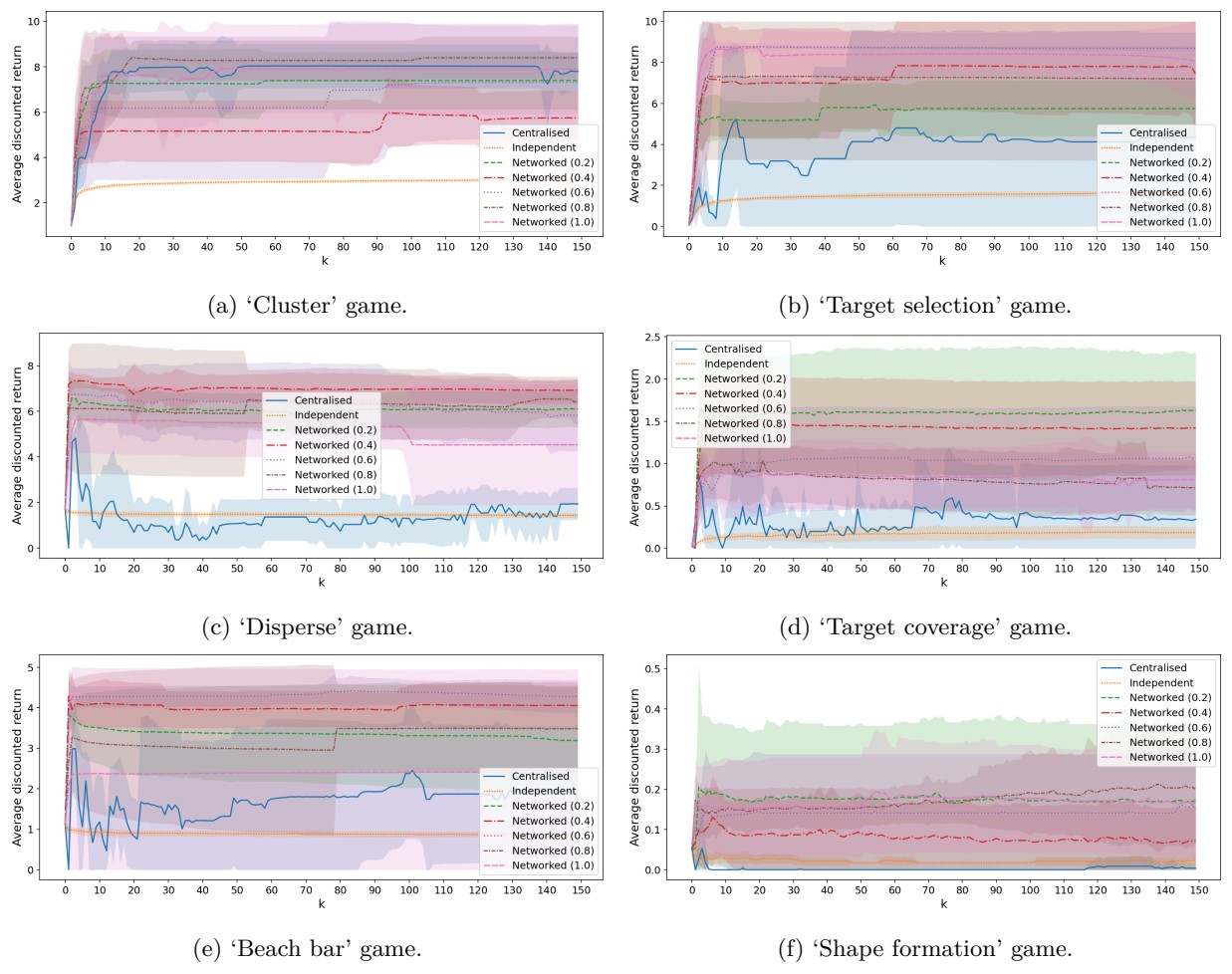

(a) 'Cluster' game.

(b) 'Target selection' game.

(c) 'Disperse' game.

(d) 'Target coverage' game.

(e) 'Beach bar' game.

(f) 'Shape formation' game.

Figure 5: Standard algorithms but $C_e = C_r = C_p = \mathbf{50}$. Discussed in Sec. 6.3.2.

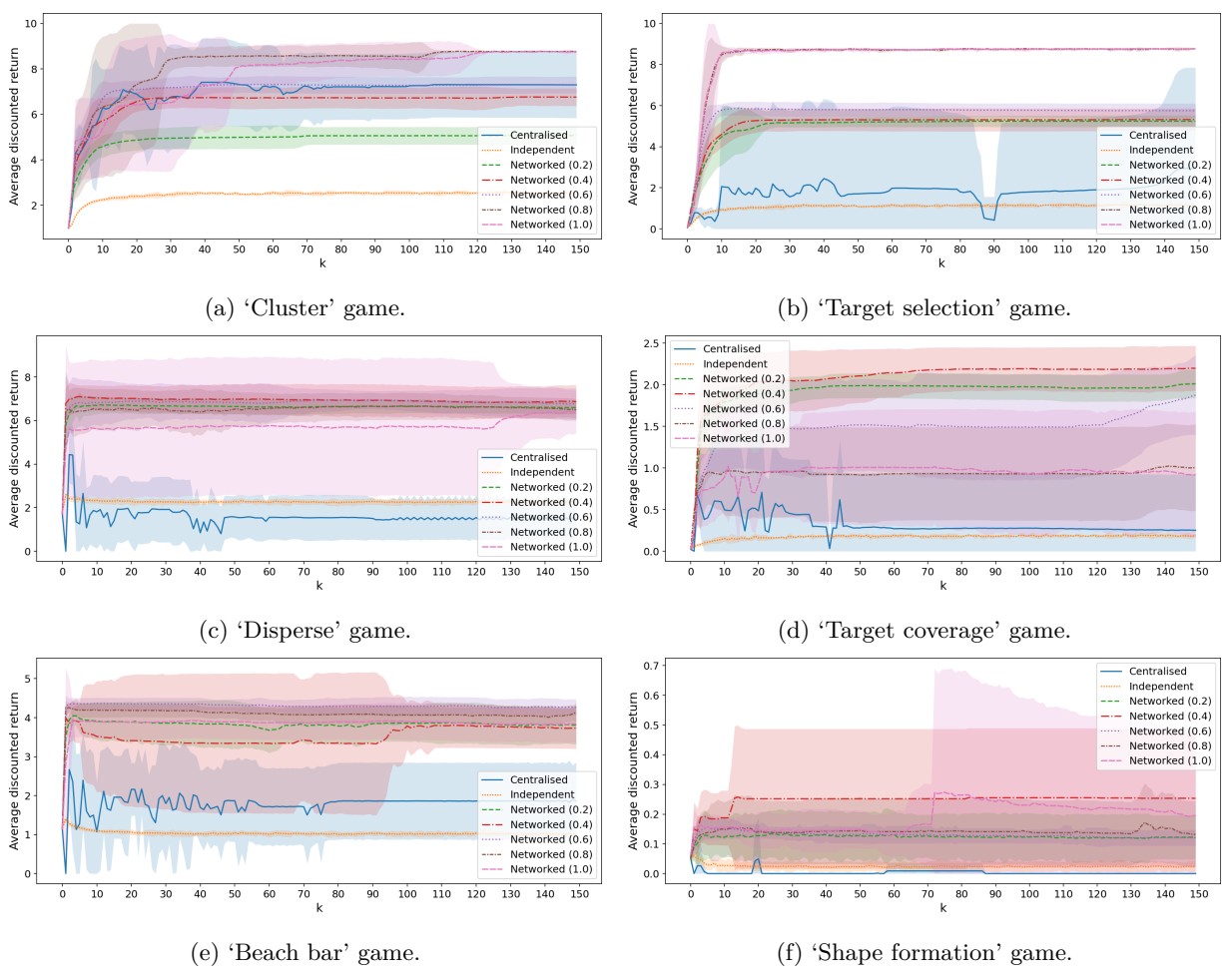

(a) 'Cluster' game.

(b) 'Target selection' game.

(c) 'Disperse' game.

(d) 'Target coverage' game.

(e) 'Beach bar' game.

(f) 'Shape formation' game.

Figure 6: Ablation study of population-*independent* policies. No agents, including centralised and networked ones, observe or estimate the empirical mean field, and all receive a vector of zeros in its place (so as to keep the neural networks the same size as in the standard setting). $C_r = C_p = 1$. Discussed in Sec. 6.3.3.

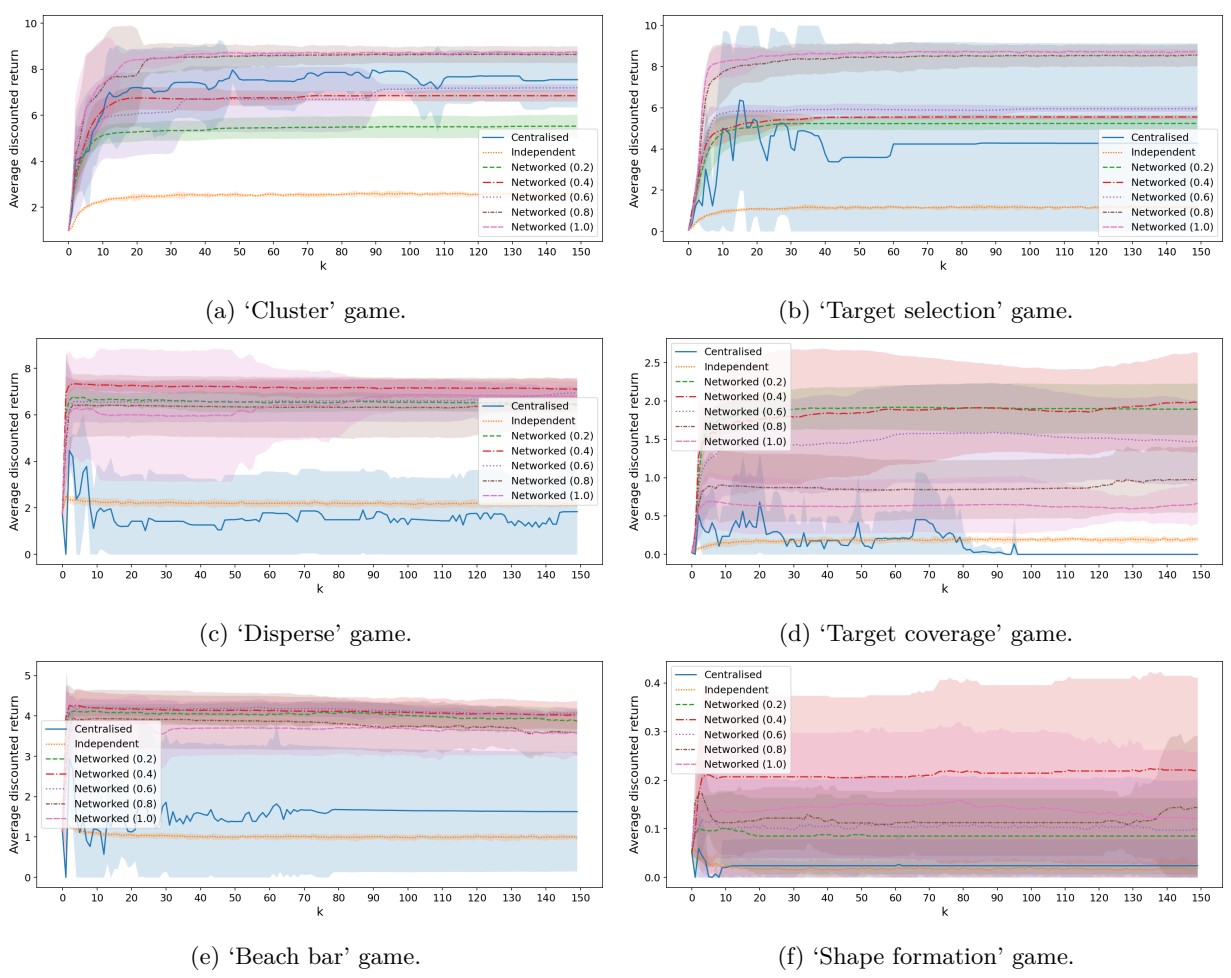

(a) 'Cluster' game.

(b) 'Target selection' game.

(c) 'Disperse' game.

(d) 'Target coverage' game.

(e) 'Beach bar' game.

(f) 'Shape formation' game.

Figure 7: Ablation study of Alg. 4 for estimating the empirical mean field - all agents, including independent ones, directly receive the true global empirical mean field. $C_r = C_p = 1$. Discussed in Sec. 6.3.3.

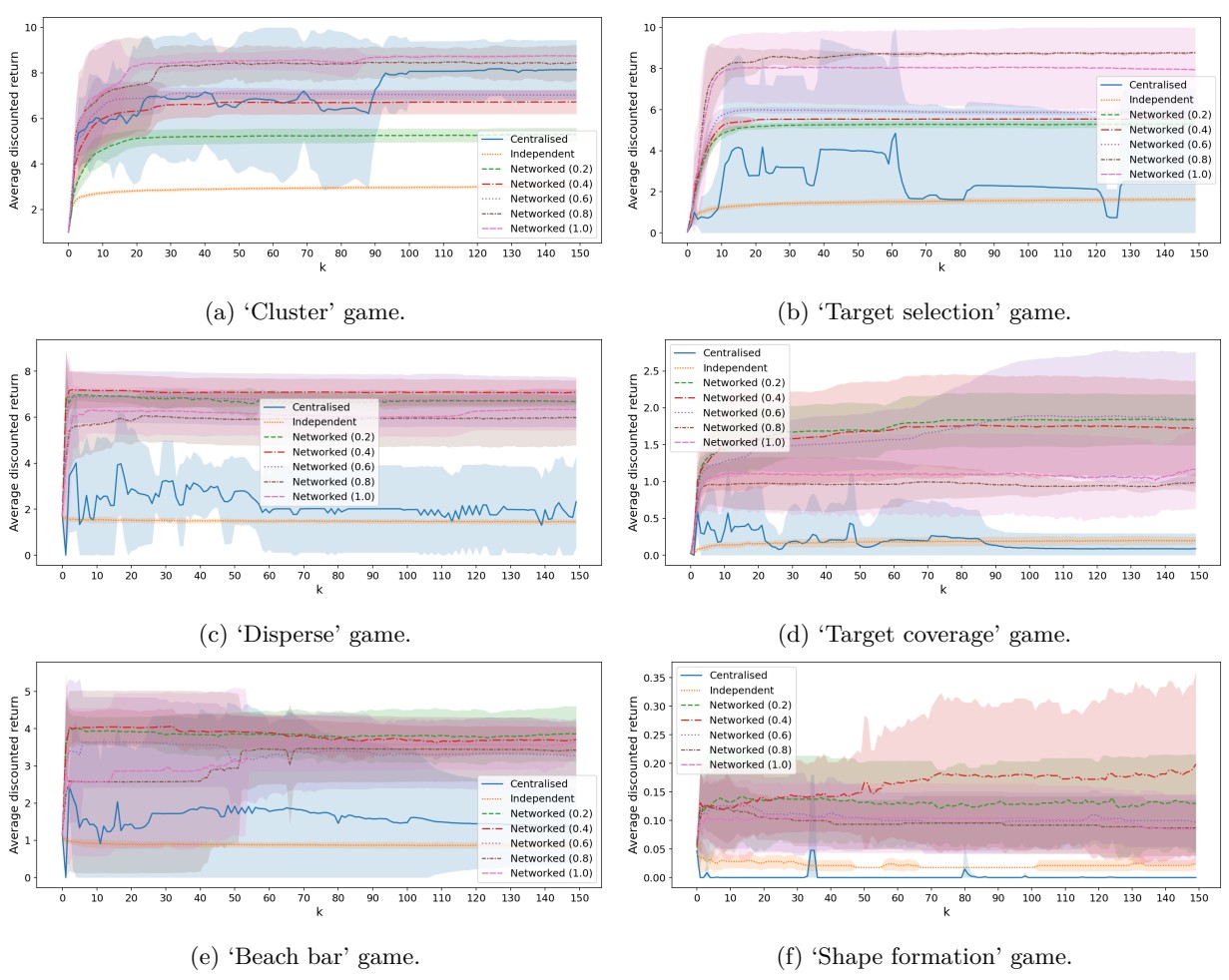

(a) 'Cluster' game.

(b) 'Target selection' game.

(c) 'Disperse' game.

(d) 'Target coverage' game.

(e) 'Beach bar' game.

(f) 'Shape formation' game.

Figure 8: Ablation study for observation of true/estimated global average reward $\hat{r}_t/\tilde{\hat{r}}_t^i$, where all agents, including centralised ones, only have access to $r_t^i$, where in the central-agent case $i = 1$. $C_e = C_p = 1$. Discussed in Sec. 6.3.3.

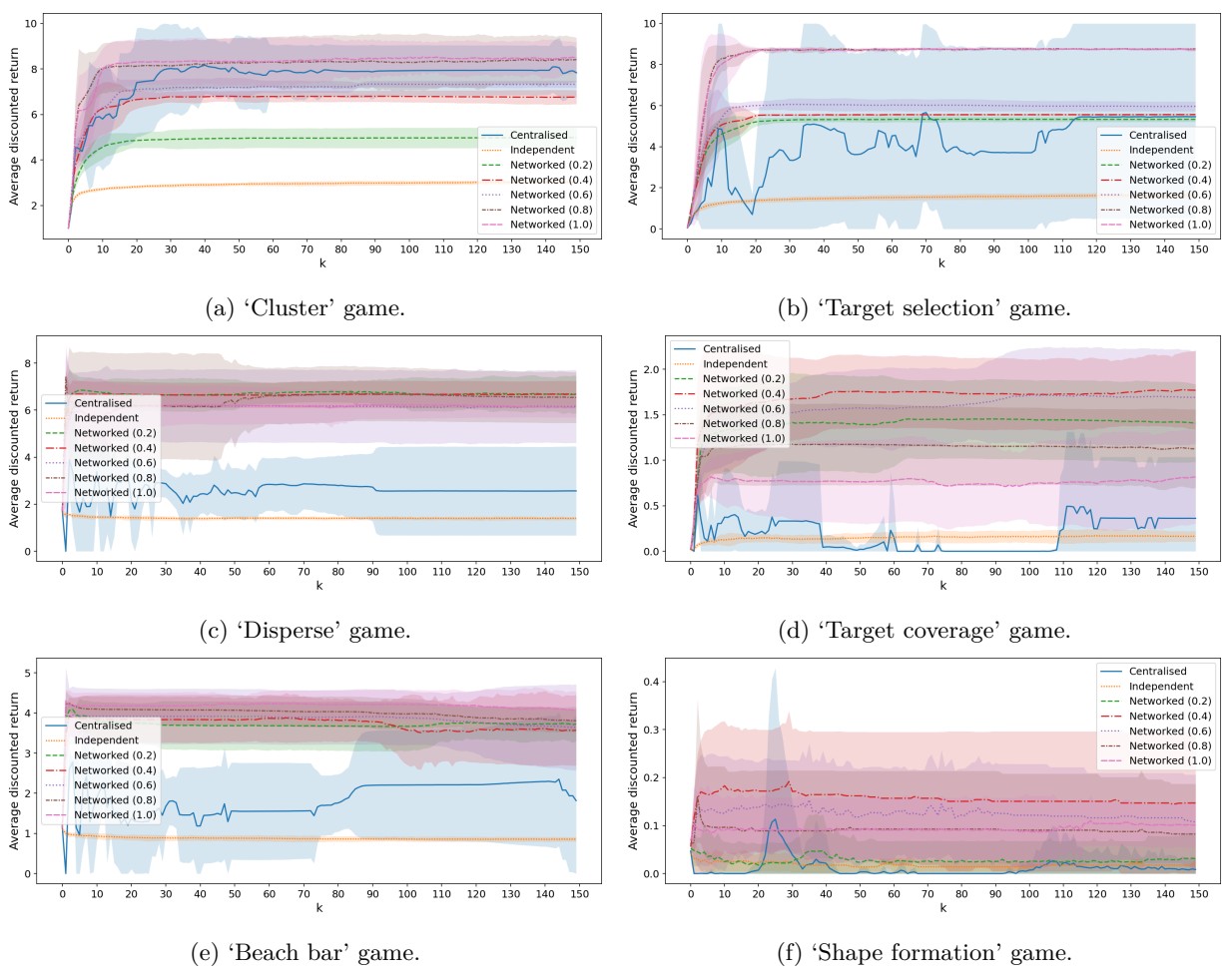

(a) 'Cluster' game.

(b) 'Target selection' game.

(c) 'Disperse' game.

(d) 'Target coverage' game.

(e) 'Beach bar' game.

(f) 'Shape formation' game.

Figure 9: Ablation study for Alg. 1 for estimating the global average reward. All agents, including both networked and independent ones, directly receive the true global average reward such that $\tilde{\hat{r}}_t^i = \hat{r}_t$. $C_e = C_p = 1$. Discussed in Sec. 6.3.3.

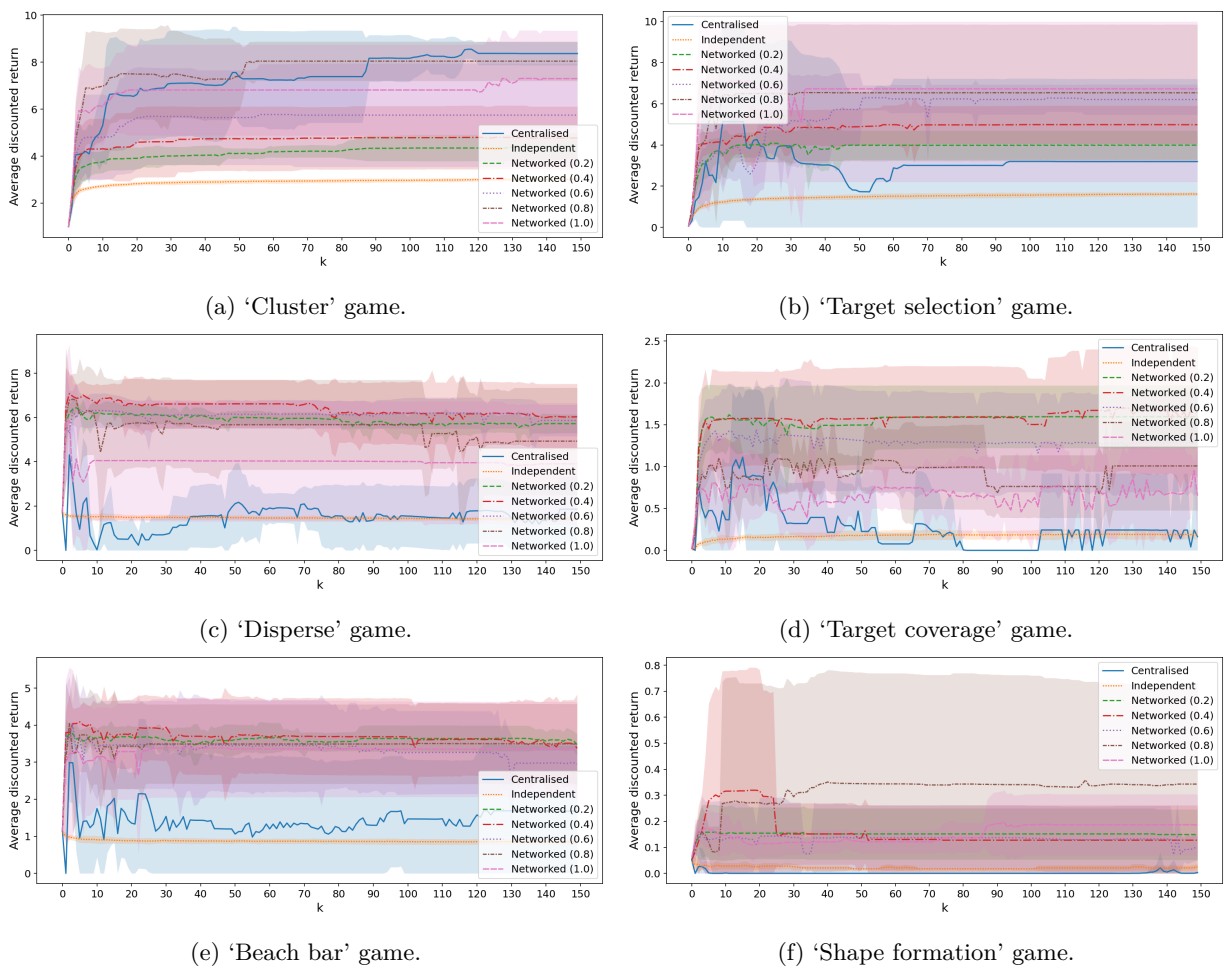

(a) 'Cluster' game.

(b) 'Target selection' game.

(c) 'Disperse' game.

(d) 'Target coverage' game.

(e) 'Beach bar' game.

(f) 'Shape formation' game.

Figure 10: Ablation study of the choice of $\tau_k^{comm}$. Here $\forall k\ \tau_k^{comm} = $ 1e-18 (i.e. $\tau_k^{comm}$ is close to 0, turning the softmax into a max function), rather than linearly increasing from 0.001 to 1 across the $K$ iterations as in all other experiments (see Table 20). $C_e = C_r = C_p = 1$. Discussed in Sec. 6.3.3.

**Methodology for the AUC tables (Tables 1-9).** For each run we report the length-normalised AUC of the per-iteration mean return over $K = 150$ outer iterations. Marginal 95%-CIs use Student's $t$ with four degrees (dof) of freedom across 5 seeds. Each $\Delta_{\text{vs Central}}$ is the mean AUC difference between the named architecture and the central-agent baseline, computed via Welch's (unpaired) two-sample $t$-test; CIs use Welch-Satterthwaite dof, and $^*$ indicates $p < 0.05$. A positive $\Delta$ therefore means the named architecture outperforms the central-agent baseline in mean AUC, and a negative $\Delta$ means the reverse; rows where our networked architecture significantly outperforms the central-agent baseline are shown in **bold**. Aggregated across the 270 networked-vs-central-agent comparisons in these tables, 172 are significant at $p < 0.05$ with positive $\Delta$, 12 with negative $\Delta$ (almost all on the 'cluster' game at broadcast radii $\leq 0.6$), and 86 are non-significant; at broadcast radii $\geq 0.8$ there are zero significant losses to the central-agent baseline across 108 comparisons. Networked-vs-independent differences are significantly positive in 251/270 comparisons with zero significant negatives (the remaining 19/270 are non-significant; aggregate mean $\Delta = +3.03$ on the $[0, 10]$ return scale) and, for ease of reference, are summarised in prose rather than tabulated.

Table 1: AUC and Welch (unpaired) mean differences vs central-agent, under the standard settings ($C_e = C_r = C_p = 1$; Fig. 2). Positive $\Delta$ indicates the named architecture outperforms the central-agent baseline. $^*$ indicates $p < 0.05$; **bold** marks networked rows where both $\Delta > 0$ and $p < 0.05$.

| Game | Architecture | Mean AUC $\pm$ CI | $\Delta_{\text{vs Central}} \pm$ CI | $p$ |
|---|---|---|---|---|
| Cluster | Central | $8.17 \pm 0.50$ | – | – |
| | Independent | $2.91 \pm 0.07$ | $-5.26 \pm 0.50^*$ | 0.000 |
| | Networked (0.2) | $5.18 \pm 0.44$ | $-2.99 \pm 0.56^*$ | 0.000 |
| | Networked (0.4) | $6.63 \pm 0.08$ | $-1.53 \pm 0.50^*$ | 0.001 |
| | Networked (0.6) | $6.85 \pm 0.55$ | $-1.32 \pm 0.62^*$ | 0.001 |
| | Networked (0.8) | $8.29 \pm 0.47$ | $+0.13 \pm 0.57$ | 0.620 |
| | Networked (1.0) | $8.42 \pm 0.32$ | $+0.25 \pm 0.51$ | 0.282 |
| Target selection | Central | $5.94 \pm 3.89$ | – | – |
| | Independent | $1.49 \pm 0.12$ | $-4.45 \pm 3.89^*$ | 0.034 |
| | Networked (0.2) | $5.17 \pm 0.10$ | $-0.77 \pm 3.89$ | 0.611 |
| | Networked (0.4) | $5.23 \pm 0.36$ | $-0.71 \pm 3.88$ | 0.640 |
| | Networked (0.6) | $5.78 \pm 0.26$ | $-0.16 \pm 3.89$ | 0.912 |
| | Networked (0.8) | $8.22 \pm 0.55$ | $+2.28 \pm 3.87$ | 0.180 |
| | Networked (1.0) | $8.01 \pm 1.43$ | $+2.07 \pm 3.83$ | 0.224 |
| Disperse | Central | $1.54 \pm 1.15$ | – | – |
| | Independent | $1.49 \pm 0.09$ | $-0.04 \pm 1.15$ | 0.923 |
| | **Networked (0.2)** | $\mathbf{6.66 \pm 0.48}$ | $\mathbf{+5.12 \pm 1.13^*}$ | **0.000** |
| | **Networked (0.4)** | $\mathbf{7.21 \pm 0.39}$ | $\mathbf{+5.67 \pm 1.13^*}$ | **0.000** |
| | **Networked (0.6)** | $\mathbf{6.66 \pm 1.02}$ | $\mathbf{+5.12 \pm 1.28^*}$ | **0.000** |
| | **Networked (0.8)** | $\mathbf{6.39 \pm 1.29}$ | $\mathbf{+4.85 \pm 1.44^*}$ | **0.000** |
| | **Networked (1.0)** | $\mathbf{6.61 \pm 0.84}$ | $\mathbf{+5.07 \pm 1.20^*}$ | **0.000** |
| Target coverage | Central | $0.24 \pm 0.22$ | – | – |
| | Independent | $0.17 \pm 0.07$ | $-0.07 \pm 0.22$ | 0.445 |
| | **Networked (0.2)** | $\mathbf{1.89 \pm 0.13}$ | $\mathbf{+1.65 \pm 0.22^*}$ | **0.000** |
| | **Networked (0.4)** | $\mathbf{1.87 \pm 0.65}$ | $\mathbf{+1.62 \pm 0.64^*}$ | **0.001** |
| | **Networked (0.6)** | $\mathbf{1.49 \pm 0.71}$ | $\mathbf{+1.24 \pm 0.70^*}$ | **0.006** |
| | **Networked (0.8)** | $\mathbf{1.02 \pm 0.68}$ | $\mathbf{+0.78 \pm 0.67^*}$ | **0.030** |
| | **Networked (1.0)** | $\mathbf{0.94 \pm 0.42}$ | $\mathbf{+0.70 \pm 0.42^*}$ | **0.007** |
| Beach bar | Central | $1.86 \pm 1.48$ | – | – |
| | Independent | $0.89 \pm 0.07$ | $-0.97 \pm 1.48$ | 0.142 |
| | **Networked (0.2)** | $\mathbf{4.07 \pm 0.34}$ | $\mathbf{+2.21 \pm 1.46^*}$ | **0.013** |
| | **Networked (0.4)** | $\mathbf{3.84 \pm 0.82}$ | $\mathbf{+1.99 \pm 1.47^*}$ | **0.016** |
| | **Networked (0.6)** | $\mathbf{4.18 \pm 0.17}$ | $\mathbf{+2.32 \pm 1.47^*}$ | **0.012** |
| | **Networked (0.8)** | $\mathbf{4.02 \pm 0.62}$ | $\mathbf{+2.16 \pm 1.45^*}$ | **0.012** |
| | **Networked (1.0)** | $\mathbf{3.64 \pm 0.87}$ | $\mathbf{+1.79 \pm 1.48^*}$ | **0.025** |
| Shape formation | Central | $0.00 \pm 0.00$ | – | – |
| | Independent | $0.02 \pm 0.00$ | $+0.02 \pm 0.00^*$ | 0.000 |
| | **Networked (0.2)** | $\mathbf{0.17 \pm 0.07}$ | $\mathbf{+0.17 \pm 0.07^*}$ | **0.003** |
| | **Networked (0.4)** | $\mathbf{0.16 \pm 0.08}$ | $\mathbf{+0.16 \pm 0.08^*}$ | **0.005** |
| | **Networked (0.6)** | $\mathbf{0.14 \pm 0.08}$ | $\mathbf{+0.14 \pm 0.08^*}$ | **0.008** |
| | **Networked (0.8)** | $\mathbf{0.15 \pm 0.04}$ | $\mathbf{+0.15 \pm 0.04^*}$ | **0.001** |
| | **Networked (1.0)** | $\mathbf{0.16 \pm 0.11}$ | $\mathbf{+0.16 \pm 0.11^*}$ | **0.015** |

Table 2: AUC and Welch (unpaired) mean differences vs central-agent, under the 90% communication-link failure setting (Fig. 3). Positive $\Delta$ indicates the named architecture outperforms the central-agent baseline. $^*$ indicates $p < 0.05$; **bold** marks networked rows where both $\Delta > 0$ and $p < 0.05$.

| Game | Architecture | Mean AUC $\pm$ CI | $\Delta_{\text{vs Central}} \pm$ CI | $p$ |
|---|---|---|---|---|
| Cluster | Central | $7.20 \pm 1.07$ | – | – |
| | Independent | $2.90 \pm 0.07$ | $-4.30 \pm 1.07^*$ | 0.000 |
| | Networked (0.2) | $4.45 \pm 0.19$ | $-2.75 \pm 1.07^*$ | 0.002 |
| | Networked (0.4) | $6.72 \pm 0.05$ | $-0.48 \pm 1.07$ | 0.283 |
| | Networked (0.6) | $6.79 \pm 0.07$ | $-0.41 \pm 1.07$ | 0.349 |
| | Networked (0.8) | $8.21 \pm 0.38$ | $+1.01 \pm 1.06$ | 0.058 |
| | **Networked (1.0)** | $\mathbf{8.50 \pm 0.03}$ | $\mathbf{+1.30 \pm 1.07^*}$ | **0.028** |
| Target selection | Central | $6.46 \pm 2.65$ | – | – |
| | Independent | $1.49 \pm 0.10$ | $-4.97 \pm 2.64^*$ | 0.006 |
| | Networked (0.2) | $3.71 \pm 0.28$ | $-2.75 \pm 2.64^*$ | 0.044 |
| | Networked (0.4) | $5.41 \pm 0.02$ | $-1.05 \pm 2.65$ | 0.331 |
| | Networked (0.6) | $5.56 \pm 0.14$ | $-0.90 \pm 2.64$ | 0.398 |
| | Networked (0.8) | $7.78 \pm 0.97$ | $+1.32 \pm 2.60$ | 0.250 |
| | Networked (1.0) | $8.47 \pm 0.08$ | $+2.01 \pm 2.64$ | 0.102 |
| Disperse | Central | $1.45 \pm 0.50$ | – | – |
| | Independent | $1.49 \pm 0.11$ | $+0.04 \pm 0.49$ | 0.831 |
| | **Networked (0.2)** | $\mathbf{7.23 \pm 0.14}$ | $\mathbf{+5.78 \pm 0.49^*}$ | **0.000** |
| | **Networked (0.4)** | $\mathbf{6.88 \pm 0.36}$ | $\mathbf{+5.44 \pm 0.52^*}$ | **0.000** |
| | **Networked (0.6)** | $\mathbf{6.77 \pm 0.20}$ | $\mathbf{+5.32 \pm 0.49^*}$ | **0.000** |
| | **Networked (0.8)** | $\mathbf{6.76 \pm 0.15}$ | $\mathbf{+5.32 \pm 0.49^*}$ | **0.000** |
| | **Networked (1.0)** | $\mathbf{6.88 \pm 0.23}$ | $\mathbf{+5.44 \pm 0.49^*}$ | **0.000** |
| Target coverage | Central | $0.72 \pm 0.35$ | – | – |
| | Independent | $0.18 \pm 0.04$ | $-0.54 \pm 0.35^*$ | 0.012 |
| | **Networked (0.2)** | $\mathbf{1.71 \pm 0.25}$ | $\mathbf{+0.99 \pm 0.37^*}$ | **0.000** |
| | **Networked (0.4)** | $\mathbf{2.07 \pm 0.23}$ | $\mathbf{+1.35 \pm 0.36^*}$ | **0.000** |
| | **Networked (0.6)** | $\mathbf{1.85 \pm 0.44}$ | $\mathbf{+1.13 \pm 0.47^*}$ | **0.001** |
| | **Networked (0.8)** | $\mathbf{1.33 \pm 0.28}$ | $\mathbf{+0.61 \pm 0.38^*}$ | **0.006** |
| | **Networked (1.0)** | $\mathbf{1.28 \pm 0.39}$ | $\mathbf{+0.56 \pm 0.44^*}$ | **0.018** |
| Beach bar | Central | $0.54 \pm 0.29$ | – | – |
| | Independent | $0.87 \pm 0.09$ | $+0.33 \pm 0.29^*$ | 0.031 |
| | **Networked (0.2)** | $\mathbf{4.05 \pm 0.08}$ | $\mathbf{+3.51 \pm 0.29^*}$ | **0.000** |
| | **Networked (0.4)** | $\mathbf{3.77 \pm 0.27}$ | $\mathbf{+3.23 \pm 0.33^*}$ | **0.000** |
| | **Networked (0.6)** | $\mathbf{3.30 \pm 1.42}$ | $\mathbf{+2.76 \pm 1.41^*}$ | **0.005** |
| | **Networked (0.8)** | $\mathbf{3.91 \pm 0.11}$ | $\mathbf{+3.37 \pm 0.29^*}$ | **0.000** |
| | **Networked (1.0)** | $\mathbf{3.90 \pm 0.29}$ | $\mathbf{+3.36 \pm 0.34^*}$ | **0.000** |
| Shape formation | Central | $0.00 \pm 0.00$ | – | – |
| | Independent | $0.02 \pm 0.00$ | $+0.02 \pm 0.00^*$ | 0.000 |
| | Networked (0.2) | $0.02 \pm 0.03$ | $+0.02 \pm 0.03$ | 0.133 |
| | **Networked (0.4)** | $\mathbf{0.23 \pm 0.11}$ | $\mathbf{+0.23 \pm 0.11^*}$ | **0.004** |
| | **Networked (0.6)** | $\mathbf{0.20 \pm 0.16}$ | $\mathbf{+0.20 \pm 0.16^*}$ | **0.025** |
| | Networked (0.8) | $0.18 \pm 0.24$ | $+0.18 \pm 0.24$ | 0.116 |
| | Networked (1.0) | $0.13 \pm 0.19$ | $+0.13 \pm 0.19$ | 0.134 |

Table 3: AUC and Welch (unpaired) mean differences vs central-agent, under the $C_e = C_r = C_p = 10$ setting (Fig. 4). Positive $\Delta$ indicates the named architecture outperforms the central-agent baseline. $^*$ indicates $p < 0.05$; **bold** marks networked rows where both $\Delta > 0$ and $p < 0.05$.

| Game | Architecture | Mean AUC $\pm$ CI | $\Delta_{\text{vs Central}} \pm$ CI | $p$ |
|------|-------------|-------------------|-------------------------------------|-----|
| Cluster | Central | $7.70 \pm 0.67$ | – | – |
| | Independent | $2.91 \pm 0.07$ | $-4.79 \pm 0.67^*$ | 0.000 |
| | Networked (0.2) | $6.90 \pm 0.98$ | $-0.80 \pm 1.01$ | 0.103 |
| | Networked (0.4) | $7.38 \pm 0.92$ | $-0.32 \pm 0.96$ | 0.458 |
| | Networked (0.6) | $7.03 \pm 2.42$ | $-0.67 \pm 2.38$ | 0.494 |
| | Networked (0.8) | $7.42 \pm 2.15$ | $-0.28 \pm 2.12$ | 0.747 |
| | **Networked (1.0)** | $\mathbf{8.60 \pm 0.16}$ | $\mathbf{+0.90 \pm 0.66^*}$ | **0.019** |
| Target selection | Central | $5.31 \pm 4.53$ | – | – |
| | Independent | $1.48 \pm 0.12$ | $-3.84 \pm 4.53$ | 0.078 |
| | Networked (0.2) | $6.42 \pm 1.59$ | $+1.11 \pm 4.45$ | 0.551 |
| | Networked (0.4) | $6.14 \pm 3.71$ | $+0.83 \pm 4.90$ | 0.705 |
| | Networked (0.6) | $8.33 \pm 0.27$ | $+3.02 \pm 4.53$ | 0.138 |
| | Networked (0.8) | $7.14 \pm 3.92$ | $+1.83 \pm 4.99$ | 0.423 |
| | Networked (1.0) | $8.60 \pm 0.05$ | $+3.28 \pm 4.53$ | 0.115 |
| Disperse | Central | $1.37 \pm 0.79$ | – | – |
| | Independent | $1.49 \pm 0.09$ | $+0.12 \pm 0.79$ | 0.693 |
| | **Networked (0.2)** | $\mathbf{6.50 \pm 0.38}$ | $\mathbf{+5.13 \pm 0.78^*}$ | **0.000** |
| | **Networked (0.4)** | $\mathbf{6.94 \pm 0.57}$ | $\mathbf{+5.57 \pm 0.83^*}$ | **0.000** |
| | **Networked (0.6)** | $\mathbf{6.66 \pm 0.67}$ | $\mathbf{+5.29 \pm 0.86^*}$ | **0.000** |
| | **Networked (0.8)** | $\mathbf{6.29 \pm 1.28}$ | $\mathbf{+4.92 \pm 1.29^*}$ | **0.000** |
| | **Networked (1.0)** | $\mathbf{6.23 \pm 1.20}$ | $\mathbf{+4.86 \pm 1.22^*}$ | **0.000** |
| Target coverage | Central | $0.32 \pm 0.39$ | – | – |
| | Independent | $0.17 \pm 0.06$ | $-0.16 \pm 0.39$ | 0.335 |
| | **Networked (0.2)** | $\mathbf{1.47 \pm 0.31}$ | $\mathbf{+1.15 \pm 0.42^*}$ | **0.000** |
| | **Networked (0.4)** | $\mathbf{1.25 \pm 0.35}$ | $\mathbf{+0.93 \pm 0.44^*}$ | **0.001** |
| | **Networked (0.6)** | $\mathbf{1.14 \pm 0.45}$ | $\mathbf{+0.82 \pm 0.49^*}$ | **0.005** |
| | **Networked (0.8)** | $\mathbf{1.03 \pm 0.32}$ | $\mathbf{+0.70 \pm 0.42^*}$ | **0.005** |
| | **Networked (1.0)** | $\mathbf{0.85 \pm 0.42}$ | $\mathbf{+0.53 \pm 0.48^*}$ | **0.035** |
| Beach bar | Central | $1.55 \pm 1.40$ | – | – |
| | Independent | $0.88 \pm 0.08$ | $-0.67 \pm 1.40$ | 0.258 |
| | **Networked (0.2)** | $\mathbf{4.00 \pm 0.31}$ | $\mathbf{+2.45 \pm 1.39^*}$ | **0.007** |
| | **Networked (0.4)** | $\mathbf{4.13 \pm 0.92}$ | $\mathbf{+2.59 \pm 1.43^*}$ | **0.004** |
| | **Networked (0.6)** | $\mathbf{4.17 \pm 0.24}$ | $\mathbf{+2.62 \pm 1.39^*}$ | **0.006** |
| | **Networked (0.8)** | $\mathbf{4.04 \pm 0.77}$ | $\mathbf{+2.49 \pm 1.40^*}$ | **0.005** |
| | Networked (1.0) | $3.30 \pm 1.77$ | $+1.76 \pm 1.89$ | 0.065 |
| Shape formation | Central | $0.00 \pm 0.00$ | – | – |
| | Independent | $0.02 \pm 0.00$ | $+0.02 \pm 0.00^*$ | 0.000 |
| | Networked (0.2) | $0.08 \pm 0.08$ | $+0.08 \pm 0.08$ | 0.053 |
| | **Networked (0.4)** | $\mathbf{0.16 \pm 0.12}$ | $\mathbf{+0.16 \pm 0.12^*}$ | **0.023** |
| | **Networked (0.6)** | $\mathbf{0.11 \pm 0.09}$ | $\mathbf{+0.11 \pm 0.09^*}$ | **0.027** |
| | **Networked (0.8)** | $\mathbf{0.17 \pm 0.13}$ | $\mathbf{+0.16 \pm 0.13^*}$ | **0.025** |
| | **Networked (1.0)** | $\mathbf{0.11 \pm 0.09}$ | $\mathbf{+0.11 \pm 0.09^*}$ | **0.025** |

Table 4: AUC and Welch (unpaired) mean differences vs central-agent, under the $C_e = C_r = C_p = 50$ setting (Fig. 5). Positive $\Delta$ indicates the named architecture outperforms the central-agent baseline. $^*$ indicates $p < 0.05$; **bold** marks networked rows where both $\Delta > 0$ and $p < 0.05$.

| Game | Architecture | Mean AUC $\pm$ CI | $\Delta_{\text{vs Central}} \pm$ CI | $p$ |
|---|---|---|---|---|
| Cluster | Central | $7.72 \pm 0.88$ | – | – |
| | Independent | $2.91 \pm 0.06$ | $-4.81 \pm 0.88^*$ | 0.000 |
| | Networked (0.2) | $7.26 \pm 1.33$ | $-0.46 \pm 1.36$ | 0.449 |
| | Networked (0.4) | $5.38 \pm 0.82$ | $-2.34 \pm 1.00^*$ | 0.001 |
| | Networked (0.6) | $6.63 \pm 2.66$ | $-1.10 \pm 2.62$ | 0.328 |
| | Networked (0.8) | $8.16 \pm 0.73$ | $+0.44 \pm 0.95$ | 0.318 |
| | Networked (1.0) | $7.72 \pm 1.94$ | $-0.00 \pm 1.91$ | 0.996 |
| Target selection | Central | $3.83 \pm 4.30$ | – | – |
| | Independent | $1.49 \pm 0.11$ | $-2.34 \pm 4.30$ | 0.205 |
| | Networked (0.2) | $5.56 \pm 0.93$ | $+1.72 \pm 4.26$ | 0.334 |
| | Networked (0.4) | $7.41 \pm 2.06$ | $+3.57 \pm 4.25$ | 0.085 |
| | **Networked (0.6)** | $\mathbf{8.54 \pm 0.13}$ | $\mathbf{+4.71 \pm 4.30^*}$ | **0.038** |
| | Networked (0.8) | $7.15 \pm 3.96$ | $+3.32 \pm 4.86$ | 0.154 |
| | **Networked (1.0)** | $\mathbf{8.29 \pm 1.06}$ | $\mathbf{+4.46 \pm 4.25^*}$ | **0.043** |
| Disperse | Central | $1.31 \pm 0.81$ | – | – |
| | Independent | $1.48 \pm 0.10$ | $+0.17 \pm 0.80$ | 0.601 |
| | **Networked (0.2)** | $\mathbf{6.08 \pm 0.75}$ | $\mathbf{+4.77 \pm 0.92^*}$ | **0.000** |
| | **Networked (0.4)** | $\mathbf{6.99 \pm 0.61}$ | $\mathbf{+5.68 \pm 0.85^*}$ | **0.000** |
| | **Networked (0.6)** | $\mathbf{6.29 \pm 1.37}$ | $\mathbf{+4.98 \pm 1.37^*}$ | **0.000** |
| | **Networked (0.8)** | $\mathbf{6.22 \pm 1.82}$ | $\mathbf{+4.91 \pm 1.79^*}$ | **0.001** |
| | **Networked (1.0)** | $\mathbf{5.17 \pm 1.60}$ | $\mathbf{+3.86 \pm 1.58^*}$ | **0.001** |
| Target coverage | Central | $0.32 \pm 0.32$ | – | – |
| | Independent | $0.17 \pm 0.07$ | $-0.15 \pm 0.32$ | 0.257 |
| | **Networked (0.2)** | $\mathbf{1.58 \pm 0.73}$ | $\mathbf{+1.26 \pm 0.72^*}$ | **0.006** |
| | **Networked (0.4)** | $\mathbf{1.42 \pm 0.55}$ | $\mathbf{+1.10 \pm 0.55^*}$ | **0.003** |
| | **Networked (0.6)** | $\mathbf{1.02 \pm 0.58}$ | $\mathbf{+0.70 \pm 0.58^*}$ | **0.024** |
| | **Networked (0.8)** | $\mathbf{0.81 \pm 0.34}$ | $\mathbf{+0.49 \pm 0.39^*}$ | **0.020** |
| | **Networked (1.0)** | $\mathbf{0.84 \pm 0.37}$ | $\mathbf{+0.51 \pm 0.41^*}$ | **0.020** |
| Beach bar | Central | $1.70 \pm 1.43$ | – | – |
| | Independent | $0.89 \pm 0.07$ | $-0.81 \pm 1.43$ | 0.191 |
| | **Networked (0.2)** | $\mathbf{3.35 \pm 1.17}$ | $\mathbf{+1.65 \pm 1.55^*}$ | **0.040** |
| | **Networked (0.4)** | $\mathbf{4.01 \pm 0.47}$ | $\mathbf{+2.31 \pm 1.41^*}$ | **0.009** |
| | **Networked (0.6)** | $\mathbf{4.31 \pm 0.27}$ | $\mathbf{+2.61 \pm 1.42^*}$ | **0.006** |
| | Networked (0.8) | $3.24 \pm 1.44$ | $+1.54 \pm 1.69$ | 0.069 |
| | Networked (1.0) | $2.39 \pm 2.56$ | $+0.69 \pm 2.55$ | 0.539 |
| Shape formation | Central | $0.00 \pm 0.00$ | – | – |
| | Independent | $0.02 \pm 0.00$ | $+0.02 \pm 0.00^*$ | 0.000 |
| | Networked (0.2) | $0.17 \pm 0.19$ | $+0.17 \pm 0.19$ | 0.064 |
| | Networked (0.4) | $0.08 \pm 0.08$ | $+0.08 \pm 0.08$ | 0.060 |
| | Networked (0.6) | $0.14 \pm 0.14$ | $+0.14 \pm 0.14$ | 0.054 |
| | **Networked (0.8)** | $\mathbf{0.17 \pm 0.07}$ | $\mathbf{+0.17 \pm 0.07^*}$ | **0.002** |
| | **Networked (1.0)** | $\mathbf{0.17 \pm 0.13}$ | $\mathbf{+0.17 \pm 0.13^*}$ | **0.022** |

Table 5: AUC and Welch (unpaired) mean differences vs central-agent, under the population-independent-policies ablation (Fig. 6). Positive $\Delta$ indicates the named architecture outperforms the central-agent baseline. $^*$ indicates $p < 0.05$; **bold** marks networked rows where both $\Delta > 0$ and $p < 0.05$.

| Game | Architecture | Mean AUC $\pm$ CI | $\Delta_{\text{vs Central}} \pm$ CI | $p$ |
|---|---|---|---|---|
| Cluster | Central | $6.97 \pm 1.52$ | – | – |
| | Independent | $2.47 \pm 0.06$ | $-4.50 \pm 1.51^*$ | 0.001 |
| | Networked (0.2) | $4.92 \pm 0.42$ | $-2.05 \pm 1.49^*$ | 0.018 |
| | Networked (0.4) | $6.54 \pm 0.46$ | $-0.42 \pm 1.49$ | 0.492 |
| | Networked (0.6) | $7.10 \pm 0.36$ | $+0.13 \pm 1.50$ | 0.827 |
| | Networked (0.8) | $8.19 \pm 0.72$ | $+1.22 \pm 1.50$ | 0.092 |
| | Networked (1.0) | $7.75 \pm 0.73$ | $+0.79 \pm 1.50$ | 0.243 |
| Target selection | Central | $1.80 \pm 3.30$ | – | – |
| | Independent | $1.09 \pm 0.06$ | $-0.71 \pm 3.30$ | 0.580 |
| | Networked (0.2) | $5.03 \pm 0.35$ | $+3.22 \pm 3.29$ | 0.053 |
| | **Networked (0.4)** | $\mathbf{5.14 \pm 0.51}$ | $\mathbf{+3.34 \pm 3.28^*}$ | **0.047** |
| | **Networked (0.6)** | $\mathbf{5.66 \pm 0.34}$ | $\mathbf{+3.86 \pm 3.29^*}$ | **0.031** |
| | **Networked (0.8)** | $\mathbf{8.47 \pm 0.12}$ | $\mathbf{+6.67 \pm 3.30^*}$ | **0.005** |
| | **Networked (1.0)** | $\mathbf{8.47 \pm 0.16}$ | $\mathbf{+6.67 \pm 3.29^*}$ | **0.005** |
| Disperse | Central | $1.58 \pm 0.79$ | – | – |
| | Independent | $2.27 \pm 0.13$ | $+0.69 \pm 0.79$ | 0.072 |
| | **Networked (0.2)** | $\mathbf{6.62 \pm 0.33}$ | $\mathbf{+5.04 \pm 0.78^*}$ | **0.000** |
| | **Networked (0.4)** | $\mathbf{6.92 \pm 0.68}$ | $\mathbf{+5.34 \pm 0.87^*}$ | **0.000** |
| | **Networked (0.6)** | $\mathbf{6.84 \pm 0.68}$ | $\mathbf{+5.26 \pm 0.87^*}$ | **0.000** |
| | **Networked (0.8)** | $\mathbf{6.55 \pm 0.66}$ | $\mathbf{+4.97 \pm 0.86^*}$ | **0.000** |
| | **Networked (1.0)** | $\mathbf{5.76 \pm 2.84}$ | $\mathbf{+4.18 \pm 2.80^*}$ | **0.013** |
| Target coverage | Central | $0.32 \pm 0.59$ | – | – |
| | Independent | $0.17 \pm 0.02$ | $-0.14 \pm 0.59$ | 0.534 |
| | **Networked (0.2)** | $\mathbf{1.92 \pm 0.14}$ | $\mathbf{+1.60 \pm 0.58^*}$ | **0.001** |
| | **Networked (0.4)** | $\mathbf{2.09 \pm 0.29}$ | $\mathbf{+1.77 \pm 0.58^*}$ | **0.000** |
| | **Networked (0.6)** | $\mathbf{1.49 \pm 0.52}$ | $\mathbf{+1.17 \pm 0.66^*}$ | **0.003** |
| | Networked (0.8) | $0.93 \pm 0.53$ | $+0.61 \pm 0.66$ | 0.065 |
| | Networked (1.0) | $0.94 \pm 0.63$ | $+0.62 \pm 0.72$ | 0.080 |
| Beach bar | Central | $1.80 \pm 0.82$ | – | – |
| | Independent | $1.04 \pm 0.06$ | $-0.76 \pm 0.82$ | 0.062 |
| | **Networked (0.2)** | $\mathbf{3.83 \pm 0.43}$ | $\mathbf{+2.03 \pm 0.82^*}$ | **0.001** |
| | **Networked (0.4)** | $\mathbf{3.53 \pm 1.24}$ | $\mathbf{+1.73 \pm 1.27^*}$ | **0.015** |
| | **Networked (0.6)** | $\mathbf{4.30 \pm 0.18}$ | $\mathbf{+2.50 \pm 0.82^*}$ | **0.001** |
| | **Networked (0.8)** | $\mathbf{4.10 \pm 0.17}$ | $\mathbf{+2.30 \pm 0.82^*}$ | **0.001** |
| | **Networked (1.0)** | $\mathbf{3.86 \pm 0.44}$ | $\mathbf{+2.06 \pm 0.82^*}$ | **0.001** |
| Shape formation | Central | $0.00 \pm 0.01$ | – | – |
| | Independent | $0.02 \pm 0.02$ | $+0.02 \pm 0.02^*$ | 0.020 |
| | **Networked (0.2)** | $\mathbf{0.12 \pm 0.08}$ | $\mathbf{+0.12 \pm 0.08^*}$ | **0.012** |
| | **Networked (0.4)** | $\mathbf{0.25 \pm 0.22}$ | $\mathbf{+0.24 \pm 0.22^*}$ | **0.037** |
| | **Networked (0.6)** | $\mathbf{0.13 \pm 0.02}$ | $\mathbf{+0.13 \pm 0.02^*}$ | **0.000** |
| | **Networked (0.8)** | $\mathbf{0.14 \pm 0.11}$ | $\mathbf{+0.14 \pm 0.11^*}$ | **0.022** |
| | Networked (1.0) | $0.19 \pm 0.20$ | $+0.19 \pm 0.20$ | 0.059 |

Table 6: AUC and Welch (unpaired) mean differences vs central-agent, under the true-global-mean-field ablation (Fig. 7). Positive $\Delta$ indicates the named architecture outperforms the central-agent baseline. $^*$ indicates $p < 0.05$; **bold** marks networked rows where both $\Delta > 0$ and $p < 0.05$.

| Game | Architecture | Mean AUC $\pm$ CI | $\Delta_{\text{vs Central}} \pm$ CI | $p$ |
|---|---|---|---|---|
| Cluster | Central | $7.31 \pm 1.11$ | – | – |
| | Independent | $2.50 \pm 0.07$ | $-4.81 \pm 1.11^*$ | 0.000 |
| | Networked (0.2) | $5.35 \pm 0.43$ | $-1.96 \pm 1.09^*$ | 0.005 |
| | Networked (0.4) | $6.64 \pm 0.25$ | $-0.67 \pm 1.10$ | 0.170 |
| | Networked (0.6) | $6.64 \pm 0.81$ | $-0.67 \pm 1.16$ | 0.216 |
| | Networked (0.8) | $8.32 \pm 0.44$ | $+1.00 \pm 1.09$ | 0.064 |
| | **Networked (1.0)** | $\mathbf{8.43 \pm 0.19}$ | $\mathbf{+1.12 \pm 1.10^*}$ | **0.048** |
| Target selection | Central | $4.13 \pm 4.33$ | – | – |
| | Independent | $1.12 \pm 0.07$ | $-3.01 \pm 4.33$ | 0.126 |
| | Networked (0.2) | $5.08 \pm 0.37$ | $+0.95 \pm 4.32$ | 0.576 |
| | Networked (0.4) | $5.37 \pm 0.16$ | $+1.25 \pm 4.33$ | 0.469 |
| | Networked (0.6) | $5.77 \pm 0.20$ | $+1.65 \pm 4.33$ | 0.351 |
| | Networked (0.8) | $8.21 \pm 0.64$ | $+4.09 \pm 4.30$ | 0.058 |
| | **Networked (1.0)** | $\mathbf{8.46 \pm 0.14}$ | $\mathbf{+4.33 \pm 4.33^*}$ | **0.050** |
| Disperse | Central | $1.59 \pm 1.71$ | – | – |
| | Independent | $2.20 \pm 0.15$ | $+0.61 \pm 1.71$ | 0.381 |
| | **Networked (0.2)** | $\mathbf{6.54 \pm 0.34}$ | $\mathbf{+4.94 \pm 1.70^*}$ | **0.001** |
| | **Networked (0.4)** | $\mathbf{7.16 \pm 0.35}$ | $\mathbf{+5.57 \pm 1.70^*}$ | **0.001** |
| | **Networked (0.6)** | $\mathbf{6.60 \pm 0.77}$ | $\mathbf{+5.01 \pm 1.69^*}$ | **0.000** |
| | **Networked (0.8)** | $\mathbf{6.34 \pm 1.27}$ | $\mathbf{+4.74 \pm 1.80^*}$ | **0.000** |
| | **Networked (1.0)** | $\mathbf{6.43 \pm 1.43}$ | $\mathbf{+4.84 \pm 1.87^*}$ | **0.000** |
| Target coverage | Central | $0.14 \pm 0.14$ | – | – |
| | Independent | $0.18 \pm 0.03$ | $+0.04 \pm 0.14$ | 0.459 |
| | **Networked (0.2)** | $\mathbf{1.86 \pm 0.28}$ | $\mathbf{+1.72 \pm 0.28^*}$ | **0.000** |
| | **Networked (0.4)** | $\mathbf{1.83 \pm 0.66}$ | $\mathbf{+1.69 \pm 0.65^*}$ | **0.002** |
| | **Networked (0.6)** | $\mathbf{1.48 \pm 0.60}$ | $\mathbf{+1.34 \pm 0.59^*}$ | **0.003** |
| | **Networked (0.8)** | $\mathbf{0.87 \pm 0.39}$ | $\mathbf{+0.73 \pm 0.39^*}$ | **0.005** |
| | **Networked (1.0)** | $\mathbf{0.63 \pm 0.30}$ | $\mathbf{+0.49 \pm 0.30^*}$ | **0.008** |
| Beach bar | Central | $1.56 \pm 1.38$ | – | – |
| | Independent | $1.02 \pm 0.06$ | $-0.54 \pm 1.38$ | 0.338 |
| | **Networked (0.2)** | $\mathbf{4.01 \pm 0.23}$ | $\mathbf{+2.45 \pm 1.37^*}$ | **0.007** |
| | **Networked (0.4)** | $\mathbf{4.10 \pm 0.35}$ | $\mathbf{+2.54 \pm 1.37^*}$ | **0.006** |
| | **Networked (0.6)** | $\mathbf{4.13 \pm 0.14}$ | $\mathbf{+2.57 \pm 1.38^*}$ | **0.007** |
| | **Networked (0.8)** | $\mathbf{3.78 \pm 0.61}$ | $\mathbf{+2.22 \pm 1.36^*}$ | **0.008** |
| | **Networked (1.0)** | $\mathbf{3.66 \pm 0.64}$ | $\mathbf{+2.10 \pm 1.37^*}$ | **0.010** |
| Shape formation | Central | $0.02 \pm 0.05$ | – | – |
| | Independent | $0.02 \pm 0.02$ | $-0.00 \pm 0.05$ | 0.814 |
| | Networked (0.2) | $0.09 \pm 0.08$ | $+0.06 \pm 0.08$ | 0.104 |
| | **Networked (0.4)** | $\mathbf{0.21 \pm 0.18}$ | $\mathbf{+0.19 \pm 0.17^*}$ | **0.039** |
| | Networked (0.6) | $0.10 \pm 0.10$ | $+0.08 \pm 0.10$ | 0.096 |
| | **Networked (0.8)** | $\mathbf{0.12 \pm 0.07}$ | $\mathbf{+0.09 \pm 0.07^*}$ | **0.018** |
| | Networked (1.0) | $0.14 \pm 0.15$ | $+0.12 \pm 0.15$ | 0.101 |

Table 7: AUC and Welch (unpaired) mean differences vs central-agent, under the no-global-average-reward ablation (Fig. 8). Positive $\Delta$ indicates the named architecture outperforms the central-agent baseline. $^*$ indicates $p < 0.05$; **bold** marks networked rows where both $\Delta > 0$ and $p < 0.05$.

| Game | Architecture | Mean AUC $\pm$ CI | $\Delta_{\text{vs Central}} \pm$ CI | $p$ |
|---|---|---|---|---|
| Cluster | Central | $7.09 \pm 1.65$ | $-$ | $-$ |
| | Independent | $2.90 \pm 0.06$ | $-4.19 \pm 1.65^*$ | $0.002$ |
| | Networked (0.2) | $5.11 \pm 0.33$ | $-1.98 \pm 1.63^*$ | $0.027$ |
| | Networked (0.4) | $6.52 \pm 0.62$ | $-0.57 \pm 1.62$ | $0.406$ |
| | Networked (0.6) | $6.91 \pm 0.30$ | $-0.18 \pm 1.64$ | $0.778$ |
| | Networked (0.8) | $8.13 \pm 0.74$ | $+1.03 \pm 1.62$ | $0.167$ |
| | Networked (1.0) | $8.36 \pm 0.26$ | $+1.27 \pm 1.64$ | $0.099$ |
| Target selection | Central | $2.53 \pm 3.11$ | $-$ | $-$ |
| | Independent | $1.49 \pm 0.12$ | $-1.04 \pm 3.11$ | $0.406$ |
| | Networked (0.2) | $5.12 \pm 0.17$ | $+2.59 \pm 3.11$ | $0.082$ |
| | Networked (0.4) | $5.38 \pm 0.05$ | $+2.85 \pm 3.11$ | $0.064$ |
| | **Networked (0.6)** | $\mathbf{5.74 \pm 0.12}$ | $\mathbf{+3.21 \pm 3.11^*}$ | $\mathbf{0.046}$ |
| | **Networked (0.8)** | $\mathbf{8.42 \pm 0.19}$ | $\mathbf{+5.89 \pm 3.11^*}$ | $\mathbf{0.006}$ |
| | **Networked (1.0)** | $\mathbf{7.81 \pm 1.77}$ | $\mathbf{+5.28 \pm 3.12^*}$ | $\mathbf{0.006}$ |
| Disperse | Central | $2.21 \pm 1.15$ | $-$ | $-$ |
| | Independent | $1.49 \pm 0.10$ | $-0.72 \pm 1.14$ | $0.155$ |
| | **Networked (0.2)** | $\mathbf{6.72 \pm 0.31}$ | $\mathbf{+4.51 \pm 1.13^*}$ | $\mathbf{0.000}$ |
| | **Networked (0.4)** | $\mathbf{7.07 \pm 0.58}$ | $\mathbf{+4.86 \pm 1.14^*}$ | $\mathbf{0.000}$ |
| | **Networked (0.6)** | $\mathbf{6.74 \pm 1.12}$ | $\mathbf{+4.53 \pm 1.33^*}$ | $\mathbf{0.000}$ |
| | **Networked (0.8)** | $\mathbf{5.89 \pm 1.14}$ | $\mathbf{+3.68 \pm 1.34^*}$ | $\mathbf{0.000}$ |
| | **Networked (1.0)** | $\mathbf{6.13 \pm 0.86}$ | $\mathbf{+3.92 \pm 1.21^*}$ | $\mathbf{0.000}$ |
| Target coverage | Central | $0.18 \pm 0.22$ | $-$ | $-$ |
| | Independent | $0.18 \pm 0.05$ | $+0.00 \pm 0.22$ | $0.994$ |
| | **Networked (0.2)** | $\mathbf{1.72 \pm 0.33}$ | $\mathbf{+1.54 \pm 0.34^*}$ | $\mathbf{0.000}$ |
| | **Networked (0.4)** | $\mathbf{1.66 \pm 0.64}$ | $\mathbf{+1.48 \pm 0.62^*}$ | $\mathbf{0.002}$ |
| | **Networked (0.6)** | $\mathbf{1.65 \pm 0.52}$ | $\mathbf{+1.47 \pm 0.51^*}$ | $\mathbf{0.001}$ |
| | **Networked (0.8)** | $\mathbf{0.95 \pm 0.19}$ | $\mathbf{+0.77 \pm 0.24^*}$ | $\mathbf{0.000}$ |
| | **Networked (1.0)** | $\mathbf{1.07 \pm 0.52}$ | $\mathbf{+0.90 \pm 0.51^*}$ | $\mathbf{0.006}$ |
| Beach bar | Central | $1.58 \pm 1.89$ | $-$ | $-$ |
| | Independent | $0.88 \pm 0.08$ | $-0.70 \pm 1.88$ | $0.360$ |
| | **Networked (0.2)** | $\mathbf{3.83 \pm 0.60}$ | $\mathbf{+2.25 \pm 1.85^*}$ | $\mathbf{0.027}$ |
| | **Networked (0.4)** | $\mathbf{3.81 \pm 0.41}$ | $\mathbf{+2.22 \pm 1.87^*}$ | $\mathbf{0.029}$ |
| | Networked (0.6) | $3.39 \pm 1.03$ | $+1.81 \pm 1.88$ | $0.057$ |
| | Networked (0.8) | $3.16 \pm 1.32$ | $+1.57 \pm 1.95$ | $0.099$ |
| | Networked (1.0) | $3.31 \pm 1.14$ | $+1.73 \pm 1.90$ | $0.068$ |
| Shape formation | Central | $0.00 \pm 0.00$ | $-$ | $-$ |
| | Independent | $0.02 \pm 0.00$ | $+0.02 \pm 0.00^*$ | $0.000$ |
| | **Networked (0.2)** | $\mathbf{0.13 \pm 0.08}$ | $\mathbf{+0.13 \pm 0.08^*}$ | $\mathbf{0.009}$ |
| | **Networked (0.4)** | $\mathbf{0.16 \pm 0.09}$ | $\mathbf{+0.16 \pm 0.09^*}$ | $\mathbf{0.009}$ |
| | **Networked (0.6)** | $\mathbf{0.10 \pm 0.05}$ | $\mathbf{+0.10 \pm 0.05^*}$ | $\mathbf{0.006}$ |
| | **Networked (0.8)** | $\mathbf{0.10 \pm 0.04}$ | $\mathbf{+0.10 \pm 0.04^*}$ | $\mathbf{0.003}$ |
| | **Networked (1.0)** | $\mathbf{0.10 \pm 0.06}$ | $\mathbf{+0.10 \pm 0.06^*}$ | $\mathbf{0.010}$ |

Table 8: AUC and Welch (unpaired) mean differences vs central-agent, under the true-global-average-reward ablation (Fig. 9). Positive $\Delta$ indicates the named architecture outperforms the central-agent baseline. $^*$ indicates $p < 0.05$; **bold** marks networked rows where both $\Delta > 0$ and $p < 0.05$.

| Game | Architecture | Mean AUC $\pm$ CI | $\Delta_{\text{vs Central}} \pm$ CI | $p$ |
|---|---|---|---|---|
| Cluster | Central | $7.59 \pm 1.20$ | – | – |
| | Independent | $2.92 \pm 0.08$ | $-4.67 \pm 1.20^*$ | 0.000 |
| | Networked (0.2) | $4.85 \pm 0.42$ | $-2.73 \pm 1.18^*$ | 0.002 |
| | Networked (0.4) | $6.62 \pm 0.26$ | $-0.97 \pm 1.19$ | 0.088 |
| | Networked (0.6) | $7.07 \pm 0.26$ | $-0.51 \pm 1.19$ | 0.304 |
| | Networked (0.8) | $8.15 \pm 1.04$ | $+0.56 \pm 1.32$ | 0.358 |
| | Networked (1.0) | $8.23 \pm 0.82$ | $+0.64 \pm 1.23$ | 0.262 |
| Target selection | Central | $4.15 \pm 2.19$ | – | – |
| | Independent | $1.50 \pm 0.12$ | $-2.65 \pm 2.19^*$ | 0.028 |
| | Networked (0.2) | $5.15 \pm 0.13$ | $+1.00 \pm 2.19$ | 0.272 |
| | Networked (0.4) | $5.39 \pm 0.10$ | $+1.24 \pm 2.19$ | 0.191 |
| | Networked (0.6) | $5.83 \pm 0.22$ | $+1.68 \pm 2.18$ | 0.100 |
| | **Networked (0.8)** | $\mathbf{8.48 \pm 0.11}$ | $\mathbf{+4.33 \pm 2.19^*}$ | **0.005** |
| | **Networked (1.0)** | $\mathbf{8.47 \pm 0.15}$ | $\mathbf{+4.32 \pm 2.18^*}$ | **0.005** |
| Disperse | Central | $2.58 \pm 1.03$ | – | – |
| | Independent | $1.41 \pm 0.07$ | $-1.17 \pm 1.03^*$ | 0.034 |
| | **Networked (0.2)** | $\mathbf{6.69 \pm 0.60}$ | $\mathbf{+4.10 \pm 1.04^*}$ | **0.000** |
| | **Networked (0.4)** | $\mathbf{6.64 \pm 0.56}$ | $\mathbf{+4.05 \pm 1.03^*}$ | **0.000** |
| | **Networked (0.6)** | $\mathbf{6.15 \pm 1.54}$ | $\mathbf{+3.57 \pm 1.58^*}$ | **0.001** |
| | **Networked (0.8)** | $\mathbf{6.46 \pm 1.45}$ | $\mathbf{+3.87 \pm 1.50^*}$ | **0.000** |
| | **Networked (1.0)** | $\mathbf{6.12 \pm 0.71}$ | $\mathbf{+3.54 \pm 1.07^*}$ | **0.000** |
| Target coverage | Central | $0.18 \pm 0.22$ | – | – |
| | Independent | $0.15 \pm 0.06$ | $-0.03 \pm 0.22$ | 0.729 |
| | **Networked (0.2)** | $\mathbf{1.40 \pm 0.46}$ | $\mathbf{+1.22 \pm 0.45^*}$ | **0.001** |
| | **Networked (0.4)** | $\mathbf{1.69 \pm 0.40}$ | $\mathbf{+1.51 \pm 0.40^*}$ | **0.000** |
| | **Networked (0.6)** | $\mathbf{1.57 \pm 0.43}$ | $\mathbf{+1.39 \pm 0.42^*}$ | **0.000** |
| | **Networked (0.8)** | $\mathbf{1.14 \pm 0.42}$ | $\mathbf{+0.96 \pm 0.42^*}$ | **0.001** |
| | **Networked (1.0)** | $\mathbf{0.76 \pm 0.43}$ | $\mathbf{+0.58 \pm 0.43^*}$ | **0.016** |
| Beach bar | Central | $1.85 \pm 0.68$ | – | – |
| | Independent | $0.88 \pm 0.07$ | $-0.97 \pm 0.68^*$ | 0.016 |
| | **Networked (0.2)** | $\mathbf{3.71 \pm 0.51}$ | $\mathbf{+1.86 \pm 0.72^*}$ | **0.000** |
| | **Networked (0.4)** | $\mathbf{3.73 \pm 0.63}$ | $\mathbf{+1.88 \pm 0.77^*}$ | **0.001** |
| | **Networked (0.6)** | $\mathbf{3.85 \pm 0.74}$ | $\mathbf{+2.00 \pm 0.84^*}$ | **0.001** |
| | **Networked (0.8)** | $\mathbf{4.00 \pm 0.32}$ | $\mathbf{+2.15 \pm 0.67^*}$ | **0.000** |
| | **Networked (1.0)** | $\mathbf{4.18 \pm 0.35}$ | $\mathbf{+2.33 \pm 0.68^*}$ | **0.000** |
| Shape formation | Central | $0.01 \pm 0.03$ | – | – |
| | Independent | $0.02 \pm 0.01$ | $+0.01 \pm 0.03$ | 0.576 |
| | Networked (0.2) | $0.03 \pm 0.04$ | $+0.02 \pm 0.05$ | 0.425 |
| | **Networked (0.4)** | $\mathbf{0.16 \pm 0.14}$ | $\mathbf{+0.15 \pm 0.13^*}$ | **0.039** |
| | **Networked (0.6)** | $\mathbf{0.13 \pm 0.10}$ | $\mathbf{+0.11 \pm 0.10^*}$ | **0.029** |
| | Networked (0.8) | $0.09 \pm 0.12$ | $+0.08 \pm 0.12$ | 0.138 |
| | **Networked (1.0)** | $\mathbf{0.09 \pm 0.06}$ | $\mathbf{+0.08 \pm 0.06^*}$ | **0.017** |

Table 9: AUC and Welch (unpaired) mean differences vs central-agent, under the $\tau_k^{comm} \approx 0$ ablation (Fig. 10). Positive $\Delta$ indicates the named architecture outperforms the central-agent baseline. $^*$ indicates $p < 0.05$; **bold** marks networked rows where both $\Delta > 0$ and $p < 0.05$.

| Game | Architecture | Mean AUC $\pm$ CI | $\Delta_{\text{vs Central}} \pm$ CI | $p$ |
|---|---|---|---|---|
| Cluster | Central | $7.40 \pm 1.11$ | – | – |
| | Independent | $2.91 \pm 0.08$ | $-4.49 \pm 1.11^*$ | 0.000 |
| | Networked (0.2) | $4.15 \pm 0.38$ | $-3.25 \pm 1.10^*$ | 0.001 |
| | Networked (0.4) | $4.67 \pm 1.34$ | $-2.73 \pm 1.46^*$ | 0.003 |
| | Networked (0.6) | $5.59 \pm 2.07$ | $-1.81 \pm 2.06$ | 0.076 |
| | Networked (0.8) | $7.70 \pm 0.62$ | $+0.30 \pm 1.11$ | 0.531 |
| | Networked (1.0) | $6.78 \pm 1.80$ | $-0.62 \pm 1.83$ | 0.444 |
| Target selection | Central | $3.10 \pm 3.70$ | – | – |
| | Independent | $1.48 \pm 0.11$ | $-1.62 \pm 3.70$ | 0.291 |
| | Networked (0.2) | $3.89 \pm 0.67$ | $+0.79 \pm 3.68$ | 0.591 |
| | Networked (0.4) | $4.80 \pm 0.85$ | $+1.69 \pm 3.66$ | 0.278 |
| | Networked (0.6) | $5.60 \pm 0.74$ | $+2.50 \pm 3.67$ | 0.135 |
| | Networked (0.8) | $6.36 \pm 3.19$ | $+3.25 \pm 4.07$ | 0.103 |
| | Networked (1.0) | $6.59 \pm 4.31$ | $+3.49 \pm 4.74$ | 0.128 |
| Disperse | Central | $1.47 \pm 0.81$ | – | – |
| | Independent | $1.46 \pm 0.10$ | $-0.00 \pm 0.81$ | 0.995 |
| | **Networked (0.2)** | **$5.87 \pm 0.29$** | **$+4.41 \pm 0.80^*$** | **0.000** |
| | **Networked (0.4)** | **$6.38 \pm 1.09$** | **$+4.91 \pm 1.14^*$** | **0.000** |
| | **Networked (0.6)** | **$6.10 \pm 0.60$** | **$+4.64 \pm 0.85^*$** | **0.000** |
| | **Networked (0.8)** | **$5.47 \pm 1.97$** | **$+4.00 \pm 1.94^*$** | **0.003** |
| | Networked (1.0) | $4.00 \pm 2.69$ | $+2.53 \pm 2.64$ | 0.057 |
| Target coverage | Central | $0.27 \pm 0.21$ | – | – |
| | Independent | $0.17 \pm 0.05$ | $-0.09 \pm 0.20$ | 0.283 |
| | **Networked (0.2)** | **$1.55 \pm 0.38$** | **$+1.28 \pm 0.38^*$** | **0.000** |
| | **Networked (0.4)** | **$1.58 \pm 0.63$** | **$+1.31 \pm 0.62^*$** | **0.003** |
| | **Networked (0.6)** | **$1.29 \pm 0.58$** | **$+1.02 \pm 0.57^*$** | **0.006** |
| | **Networked (0.8)** | **$0.92 \pm 0.26$** | **$+0.65 \pm 0.28^*$** | **0.001** |
| | **Networked (1.0)** | **$0.66 \pm 0.24$** | **$+0.39 \pm 0.26^*$** | **0.009** |
| Beach bar | Central | $1.45 \pm 1.45$ | – | – |
| | Independent | $0.87 \pm 0.07$ | $-0.58 \pm 1.44$ | 0.330 |
| | **Networked (0.2)** | **$3.61 \pm 0.30$** | **$+2.16 \pm 1.43^*$** | **0.013** |
| | **Networked (0.4)** | **$3.68 \pm 0.92$** | **$+2.23 \pm 1.47^*$** | **0.009** |
| | **Networked (0.6)** | **$3.32 \pm 1.04$** | **$+1.88 \pm 1.50^*$** | **0.021** |
| | **Networked (0.8)** | **$3.48 \pm 1.08$** | **$+2.03 \pm 1.52^*$** | **0.015** |
| | **Networked (1.0)** | **$3.32 \pm 1.26$** | **$+1.87 \pm 1.60^*$** | **0.027** |
| Shape formation | Central | $0.00 \pm 0.00$ | – | – |
| | Independent | $0.02 \pm 0.00$ | $+0.02 \pm 0.00^*$ | 0.000 |
| | **Networked (0.2)** | **$0.15 \pm 0.10$** | **$+0.15 \pm 0.10^*$** | **0.013** |
| | **Networked (0.4)** | **$0.16 \pm 0.14$** | **$+0.16 \pm 0.14^*$** | **0.035** |
| | **Networked (0.6)** | **$0.13 \pm 0.12$** | **$+0.13 \pm 0.12^*$** | **0.037** |
| | Networked (0.8) | $0.32 \pm 0.39$ | $+0.32 \pm 0.39$ | 0.085 |
| | **Networked (1.0)** | **$0.15 \pm 0.05$** | **$+0.15 \pm 0.05^*$** | **0.001** |

Table 10: Per-run CPU time, in seconds rounded to the nearest integer (mean over 5 seeds), under the standard settings ($C_e = C_r = C_p = 1$; Fig. 2).

| Game | Central | Indep | 0.2 | 0.4 | 0.6 | 0.8 | 1.0 |
|---|---|---|---|---|---|---|---|
| Cluster | 211 | 210 | 276 | 275 | 276 | 276 | 276 |
| Target selection | 211 | 210 | 276 | 277 | 277 | 277 | 276 |
| Disperse | 211 | 210 | 276 | 276 | 276 | 277 | 276 |
| Target coverage | 210 | 210 | 275 | 276 | 275 | 275 | 276 |
| Beach bar | 211 | 210 | 277 | 277 | 277 | 277 | 276 |
| Shape formation | 212 | 210 | 277 | 277 | 277 | 276 | 277 |
| **All games** | **1266** | **1261** | **1657** | **1657** | **1658** | **1657** | **1657** |

Table 11: Per-run CPU time, in seconds rounded to the nearest integer (mean over 5 seeds), under the 90% communication-link failure setting (Fig. 3).

| Game | Central | Indep | 0.2 | 0.4 | 0.6 | 0.8 | 1.0 |
|---|---|---|---|---|---|---|---|
| Cluster | 211 | 211 | 276 | 275 | 276 | 275 | 275 |
| Target selection | 211 | 210 | 276 | 277 | 276 | 276 | 276 |
| Disperse | 211 | 210 | 277 | 276 | 276 | 277 | 276 |
| Target coverage | 211 | 210 | 275 | 275 | 275 | 276 | 275 |
| Beach bar | 211 | 210 | 277 | 277 | 276 | 276 | 276 |
| Shape formation | 211 | 210 | 278 | 277 | 278 | 276 | 278 |
| **All games** | **1267** | **1261** | **1659** | **1658** | **1658** | **1657** | **1657** |

Table 12: Per-run CPU time, in seconds rounded to the nearest integer (mean over 5 seeds), under the $C_e = C_r = C_p = 10$ setting (Fig. 4).

| Game | Central | Indep | 0.2 | 0.4 | 0.6 | 0.8 | 1.0 |
|---|---|---|---|---|---|---|---|
| Cluster | 211 | 211 | 504 | 504 | 505 | 504 | 504 |
| Target selection | 211 | 210 | 503 | 504 | 503 | 503 | 503 |
| Disperse | 211 | 210 | 506 | 505 | 505 | 505 | 505 |
| Target coverage | 211 | 210 | 503 | 503 | 503 | 504 | 503 |
| Beach bar | 211 | 210 | 505 | 506 | 505 | 505 | 506 |
| Shape formation | 211 | 210 | 505 | 505 | 506 | 505 | 506 |
| **All games** | **1267** | **1261** | **3027** | **3028** | **3028** | **3027** | **3028** |

Table 13: Per-run CPU time, in seconds rounded to the nearest integer (mean over 5 seeds), under the $C_e = C_r = C_p = 50$ setting (Fig. 5).

| Game | Central | Indep | 0.2 | 0.4 | 0.6 | 0.8 | 1.0 |
|---|---|---|---|---|---|---|---|
| Cluster | 211 | 210 | 2421 | 2422 | 2421 | 2421 | 2420 |
| Target selection | 211 | 210 | 2418 | 2417 | 2418 | 2418 | 2418 |
| Disperse | 211 | 210 | 2421 | 2421 | 2421 | 2421 | 2422 |
| Target coverage | 210 | 210 | 2419 | 2419 | 2419 | 2419 | 2419 |
| Beach bar | 211 | 210 | 2421 | 2421 | 2421 | 2422 | 2421 |
| Shape formation | 211 | 210 | 2421 | 2421 | 2421 | 2421 | 2421 |
| **All games** | **1265** | **1260** | **14521** | **14521** | **14521** | **14521** | **14521** |

Table 14: Per-run CPU time, in seconds rounded to the nearest integer (mean over 5 seeds), under the population-independent-policies ablation (Fig. 6).

| Game | Central | Indep | 0.2 | 0.4 | 0.6 | 0.8 | 1.0 |
|---|---|---|---|---|---|---|---|
| Cluster | 210 | 210 | 215 | 216 | 215 | 215 | 215 |
| Target selection | 211 | 210 | 215 | 215 | 215 | 216 | 215 |
| Disperse | 210 | 211 | 215 | 215 | 215 | 216 | 215 |
| Target coverage | 210 | 210 | 217 | 216 | 216 | 216 | 216 |
| Beach bar | 210 | 210 | 216 | 215 | 215 | 215 | 215 |
| Shape formation | 210 | 210 | 215 | 215 | 216 | 215 | 216 |
| **All games** | **1262** | **1261** | **1293** | **1291** | **1291** | **1292** | **1291** |

Table 15: Per-run CPU time, in seconds rounded to the nearest integer (mean over 5 seeds), under the true-global-mean-field ablation (Fig. 7).

| Game | Central | Indep | 0.2 | 0.4 | 0.6 | 0.8 | 1.0 |
|---|---|---|---|---|---|---|---|
| Cluster | 211 | 210 | 215 | 215 | 216 | 215 | 216 |
| Target selection | 211 | 209 | 216 | 215 | 215 | 215 | 215 |
| Disperse | 211 | 210 | 217 | 216 | 216 | 216 | 216 |
| Target coverage | 210 | 209 | 216 | 217 | 216 | 216 | 216 |
| Beach bar | 211 | 210 | 215 | 215 | 216 | 215 | 216 |
| Shape formation | 212 | 210 | 215 | 215 | 215 | 216 | 215 |
| **All games** | **1267** | **1258** | **1294** | **1293** | **1294** | **1292** | **1293** |

Table 16: Per-run CPU time, in seconds rounded to the nearest integer (mean over 5 seeds), under the no-global-average-reward ablation (Fig. 8).

| Game | Central | Indep | 0.2 | 0.4 | 0.6 | 0.8 | 1.0 |
|---|---|---|---|---|---|---|---|
| Cluster | 211 | 210 | 276 | 276 | 276 | 275 | 276 |
| Target selection | 211 | 210 | 274 | 274 | 274 | 274 | 275 |
| Disperse | 211 | 210 | 277 | 277 | 277 | 276 | 277 |
| Target coverage | 211 | 210 | 277 | 276 | 276 | 277 | 276 |
| Beach bar | 211 | 210 | 277 | 276 | 277 | 276 | 277 |
| Shape formation | 211 | 210 | 276 | 277 | 276 | 276 | 276 |
| **All games** | **1267** | **1260** | **1657** | **1656** | **1656** | **1655** | **1656** |

Table 17: Per-run CPU time, in seconds rounded to the nearest integer (mean over 5 seeds), under the true-global-average-reward ablation (Fig. 9).

| Game | Central | Indep | 0.2 | 0.4 | 0.6 | 0.8 | 1.0 |
|---|---|---|---|---|---|---|---|
| Cluster | 211 | 211 | 275 | 274 | 275 | 274 | 274 |
| Target selection | 210 | 210 | 276 | 277 | 276 | 275 | 276 |
| Disperse | 211 | 210 | 278 | 277 | 277 | 277 | 276 |
| Target coverage | 210 | 210 | 275 | 276 | 275 | 275 | 275 |
| Beach bar | 211 | 210 | 277 | 277 | 277 | 277 | 278 |
| Shape formation | 211 | 210 | 277 | 276 | 276 | 277 | 276 |
| **All games** | **1265** | **1261** | **1656** | **1656** | **1656** | **1655** | **1655** |

Table 18: Per-run CPU time, in seconds rounded to the nearest integer (mean over 5 seeds), under the $\tau_k^{comm} \approx 0$ ablation (Fig. 10).

| Game | Central | Indep | 0.2 | 0.4 | 0.6 | 0.8 | 1.0 |
|---|---|---|---|---|---|---|---|
| Cluster | 211 | 210 | 275 | 275 | 275 | 275 | 275 |
| Target selection | 211 | 210 | 276 | 276 | 276 | 276 | 276 |
| Disperse | 211 | 210 | 276 | 276 | 275 | 276 | 275 |
| Target coverage | 210 | 210 | 275 | 275 | 274 | 274 | 275 |
| Beach bar | 211 | 210 | 276 | 276 | 276 | 276 | 276 |
| Shape formation | 212 | 210 | 276 | 276 | 277 | 276 | 276 |
| **All games** | **1266** | **1260** | **1655** | **1654** | **1652** | **1652** | **1653** |

Table 19: Principal notation used throughout the paper.

| Symbol | Description |
|---|---|
| *Spaces and sets* | |
| $\mathcal{S}$, $\mathcal{A}$ | Finite state and common action spaces. |
| $\Delta_{\mathcal{X}}$ | Simplex of probability measures over finite set $\mathcal{X}$. |
| $\mathbf{e}_x$ | One-hot indicator vector for $x$. |
| $\mathcal{N}$ | Finite set of $N$ agents. |
| $\Pi$, $\mathcal{Q}$ | Sets of policies and of $Q$-functions. |
| *Populations and mean field* | |
| $N$ | Number of agents. |
| $\hat{\mu}_t$ | Empirical distribution of agent states at time $t$. |
| $\mu_t$, $\boldsymbol{\mu}$ | Mean-field snapshot at $t$; mean-field flow $(\mu_t)_{t \geq 0}$. |
| $\mu_{\bar{t}}$ | Initial mean-field distribution. |
| $\tilde{\mu}_t^i$ | Agent $i$'s local estimate of the mean field. |
| $I(\pi)_t$, $I(\pi)$ | Mean-field flow induced by policy $\pi$ at time $t$ (or whole flow). |
| *Policies and value functions* | |
| $\pi^i$, $\boldsymbol{\pi}$ | Individual agent $i$'s policy; joint policy $(\pi^1, \ldots, \pi^N)$. |
| $\pi_{k+1}^{\mathrm{net}}$, $\pi_{k+1}^{\mathrm{cent}}$ | Networked and centralised policy updates after iteration $k$. |
| $\pi^*$ | Social optimum (maximiser of social welfare). |
| $V^i(\boldsymbol{\pi}, \mu_{\bar{t}})$ | Agent $i$'s individual discounted return. |
| $V^{\mathrm{pop}}(\boldsymbol{\pi}, \mu_{\bar{t}})$ | Individual expected discounted returns averaged across the $N$-agent population. |
| $W(\pi; I(\pi))$ | Social welfare function. |
| $Q_{k+1}(o, a)$ | Q-function of $\pi_k$. |
| $\check{Q}_{\theta_k^i}(o, a)$ | Agent $i$'s Munchausen Q-network at iteration $k$ (with parameters $\theta_k^i$). |
| $\sigma_{k+1}^i$, $\sigma_{k+1}^{\mathrm{net}}$, $\sigma_{k+1}^{\mathrm{cent}}$ | Per-iteration return estimates (agent, networked, centralised). |
| *Algorithm parameters* | |
| $K$ | Number of outer policy-update iterations (horizon of the learning loop). |
| $M$ | Environment steps per outer iteration. |
| $L$ | Q-network gradient steps per outer iteration. |
| $E$ | Rollout length used during policy evaluation. |
| $C_r$, $C_e$, $C_p$ | Communication rounds for reward estimation, mean-field estimation, policy sharing. |
| $\gamma$ | Discount factor. |
| $\tau_q$, $\tau_k^{\mathrm{comm}}$ | Softmax temperatures (Q-function; policy adoption at iteration $k$). |
| $|B|$, $\nu$ | Replay minibatch size; target-network update period. |
| $cl$ | Munchausen log-policy clipping level. |
| *Network / graph structure* | |
| $\mathcal{G}_t^{\mathrm{comm}} = (\mathcal{N}, \mathcal{E}_t^{\mathrm{comm}})$ | Time-varying communication graph. |
| $\mathcal{G}_t^{\mathrm{vis}} = (\mathcal{N}', \mathcal{E}_t^{\mathrm{vis}})$ | Time-varying state-visibility graph. |
| $d_{\mathcal{G}_t^{\mathrm{comm}}}$ | Diameter of communication graph. |
| $J_t^i$ | Agent $i$'s one-hop communication neighbourhood at time $t$, including $i$ itself. |
| *Coordination/anti-coordination decomposition* | |
| $b(\pi)$ | Base return function. |
| $f_c(\cdot)$, $f_d(\cdot)$ | Coordination (resp. anti-coordination) scaling function. |
| $h(\cdot, \cdot)$ | Function composing $b$ and $f_c$ (or $f_d$); monotonic in both arguments. |
| *Estimators and approximators* | |
| $\tilde{r}_t^i$, $\hat{r}_t$ | Agent $i$'s average-reward estimate; true global average reward. |
| $\hat{\mathcal{R}}_{t,c}^i$ | Agent $i$'s collected rewards after $c$ communication rounds. |
| $\hat{v}_{t,c}^i[s]$ | Agent $i$'s running count of state $s$ after $c$ communication rounds. |
| $\hat{\mathcal{L}}(\theta, \theta')$ | Q-network training loss. |
| $\theta_k^i$, $\theta_{k,l}^{i,'}$ | Agent $i$'s Q-network parameters at iteration $k$; agent $i$'s target-network parameters at iteration $k$, gradient step $l$. |
| *Trajectories and counters* | |
| $s_t^i$, $o_t^i$, $a_t^i$, $r_t^i$ | State, observation, action, reward for agent $i$ at time $t$. |
| $P(\cdot \mid s, a, \mu)$, $R(s, a, \mu)$ | Mean-field-dependent transition and reward functions. |
| $t$, $k$ | Environment time step; outer iteration index. |

Table 20: Hyperparameters

| Hyperparam. | Value | Comment |
|---|---|---|
| Trials | 5 | We run 5 trials with different random seeds for each experiment. We plot the mean and 95% Student's-$t$ confidence intervals across the seeds. |
| Gridsize | 20x20 | - |
| Population | 500 | We chose 500 for our demonstrations to show that our algorithm can handle large populations, indeed often larger than those demonstrated in other mean-field works, especially for grid-world environments, while also being feasible to simulate with respect to time and computation constraints (Yang et al., 2018; Subramanian & Mahajan, 2019; Ganapathi Subramanian et al., 2020; 2021; Cui & Koeppl, 2021; Yongacoglu et al., 2024; Subramanian et al., 2022; Cui et al., 2023a; Guo et al., 2023; Benjamin & Abate, 2023; 2024; Wu et al., 2024). For example, the MFC work in Carmona et al. (2019) uses 10 agents; the work on decentralised execution for MFC by Cui et al. (2023b) uses 200 agents. |
| Number of neurons in input layer | 440 | The agent's position is represented by two concatenated one-hot vectors, indicating the agent's row and column. The mean-field distribution is a flattened vector of the same size as the grid. As such, the input size is $[(2 \times \text{dimension}) + (\text{dimension}^2)]$. |
| Neurons per hidden layer | 256 | We draw inspiration from common rules of thumb when selecting the number of neurons in hidden layers, e.g. it should be between the number of input neurons and output neurons / it should be 2/3 the size of the input layer plus the size of the output layer / it should be a power of 2 for computational efficiency. Using these rules of thumb as rough heuristics, we select the number of neurons per hidden layer by rounding the size of the input layer down to the nearest power of 2. The layers are all fully connected. |
| Hidden layers | 2 | We achieved sufficient learning speed with just 2 hidden layers, but further optimising the number of layers may lead to better results. |
| Activation function | ReLU | This is a common choice in deep RL. |
| $K$ | 150 | $K$ is chosen to be large enough to see convergence in most networked cases. |
| $M$ | 20 | We tested $M$ in $\{20,50,100\}$ and found that the lowest value was sufficient to achieve convergence while minimising training time. It may be possible to converge with even smaller choices of $M$. |
| $L$ | 20 | We tested $L$ in $\{20,50,100\}$ and found that the lowest value was sufficient to achieve convergence while minimising training time. It may be possible to converge with even smaller choices of $L$. |
| $E$ | 20 | We tested $E$ in $\{20,50,100\}$, and choose the lowest value to show the benefit to convergence even from very few evaluation steps. It may be possible to reduce this value further and still achieve similar results. |
| $C_p$ | 1 (10/50) | As in Benjamin & Abate (2023; 2024), we choose a value of 1 for most experiments to show the convergence benefits brought by even a single policy communication round, even in networks that may have limited connectivity. We also conduct additional studies to show the effect of further rounds in Figs. 4 and 5. |
| $C_r$ | 1 (10/50) | Similar to $C_p$, we choose this value to show our algorithm's ability to appropriately estimate the average reward even with only a single communication round, even in networks that may have limited connectivity. We conduct additional studies to show the effect of further rounds in Figs. 4 and 5. |
| $C_e$ | 1 (10/50) | Similar to $C_p$, we choose this value to show the ability of our algorithm to appropriately estimate the mean field even with only a single communication round, even in networks that may have limited connectivity. We also conduct additional studies to show the effect of further rounds in Figs. 4 and 5. |
| $\gamma$ | 0.9 | Standard choice across RL literature. |
| $\tau_q$ | 0.03 | We follow Vieillard et al. (2020) and Benjamin & Abate (2024), which tested a range of values. |
| $|B|$ | 32 | This is a common choice of batch size that trades off noisy updates and computational efficiency. |
| $cl$ | -1 | We use the same value as in Vieillard et al. (2020) and Benjamin & Abate (2024). |
| $\nu$ | $L-1$ | We follow Benjamin & Abate (2024), which is similar to Laurière et al. (2022b). |
| Optimiser | Adam | As in Vieillard et al. (2020), we use the Adam optimiser with initial learning rate 0.01. |
| $\tau_k^{comm}$ | cf. comment | We follow Benjamin & Abate (2024), where $\tau_k^{comm}$ increases linearly from 0.001 to 1 across the $K$ iterations. Further optimising this inverse annealing process may lead to better results; we provide an ablation study in Fig. 10. |

