# OpenReview forum: "Networked Communication for Decentralised Cooperative Agents in Mean-Field Control"
_TMLR — Decision pending for TMLR_

### Review · Reviewer_pDJJ · 2026-01-10

**Summary Of Contributions:**

The authors adapt recent results from the realm of Mean Field Games (MFG) into the realm of Mean Field Control (MFC). In particular, the authors propose the first set of algorithms for decentralized learning in MFC. They adapt a prior algorithm for MFG and propose a novel algorithm for estimating the average population reward via local communication. They also provide theoretical results which show that, under several assumptions, decentralized communication accelerates learning over both independent and central-agent settings.

**Audience:**

Yes

**Audience Explanation:**

This paper would be of interest for individuals interested in RL and MARL.

**Claims And Evidence:**

No

**Claims Explanation:**

The authors make three claims in this paper:

1) That they introduce decentralized (‘networked’) communication into MFC,
2) That they contribute a suitable novel algorithm for estimating the average population reward via local communication, and
3) That the networked communication scheme accelerates learning over both independent and central-agent settings.

In terms of the first claim, the authors adapt an existing algorithm for MFG into MFC that does indeed introduce networked communication into MFC. They provide both theoretical and empirical results which do support the validity of the proposed framework. Overall, I find the first claim to be adequately supported.

In terms of the second claim, I am not convinced that it is adequately supported. In particular, the authors propose a method for estimating the global average reward, but do not provide any sort of justification on its validity? How good of an estimate does the proposed algorithm provide? Can we bound this estimate somehow? How do we know that it is even possible to compute the global average reward with this algorithm? All in all, there is an entire theoretical section missing that supports the suitability of the proposed algorithm.

In terms of the third claim, the authors provide a rigorous theoretical analysis that supports the claim, however they fail to mention (in the abstract/introduction) that the results are only valid under several simplifying assumptions.

**Requested Changes:**

From a writing perspective, the paper is quite bloated. In particular, there is much repetition in the first 10 pages that could easily be trimmed. The theoretical section in the middle of the paper is a bit wordy but acceptable. The excessive number of figures in the experiments section also makes it harder to read; most of the figures could go in an appendix (with so many figures that all look very similar to one another it makes it difficult for the reader to take anything away from them).

Non-formatting/grammar requests are as follows:
- As mentioned above, the authors propose a method for estimating the global average reward, but do not provide any sort of justification on its validity? How good of an estimate does the proposed algorithm provide? Can we bound this estimate somehow? How do we know that it is even possible to compute the global average reward with this algorithm? All in all, there is an entire theoretical section missing that supports the suitability of the proposed algorithm.
- Similarly, the authors never explain why this novel algorithm (for estimating the global average reward) is needed in the first place. What does it add to the prior algorithm? Is it optional or is it necessary? I advise the authors to better contextualize this novel algorithm in the context of prior work.
- I am confused about Definition 3.3 vs 3.5 (population return vs social welfare). In some parts of the paper, the authors say that the goal is to optimize the population return, and in other sections the authors say that the goal is to optimize social welfare. Are these two objectives equivalent? If not, then I advise the authors to rectify such conflicting statements in the text and to clarify which of the two is the actual objective of their proposed approach.

---

> ### Author Response · Authors · 2026-05-09
> **Response 1 of 2**
>
> Thank you very much for your time and effort in reviewing our paper, and for your constructive feedback. We respond to your questions and comments slightly out of sequence below.
>
> **”I am confused about Definition 3.3 … objective of their proposed approach.”** The cooperative objective of the finite-population problem is the population-average return (Def. 3.3), which we have renamed ‘individual expected discounted return averaged across the $N$-agent population’ for clarity. In the infinite limit, called a mean-field control problem, an optimal solution can be obtained by the whole infinite population following a single policy, where the expected return of a policy that is being followed by the whole infinite population is called the social welfare (Def. 3.5).  The optimal solution to the MFC problem is the single policy that has the highest social welfare. Intuitively, the expected return of the single policy followed by the whole infinite population (the social welfare) is equivalent to the individual expected discounted return averaged across the $N$-agent population all following the same policy, when $N$ tends to infinity. When $N$ does not tend to infinity or the population does not all follow a single policy, then these two values are not necessarily equivalent (see the paragraph before Assumption 5.8), but this is the best proxy we have, when using a finite population of agents that may not be following the same policy to simulate an infinite population, for the purpose of solving the MFC problem. Therefore when using a finite population of agents that may not be following the same policy to simulate an infinite population, we maximise the individual returns averaged across the $N$-agent population as a proxy for social welfare. Moreover we might be using the solution to the MFC problem as an approximate solution to the $N$-agent problem when $N$ is large (see Rem. 3.8).
>
> Given the definition of social welfare as the expected return of the single policy followed by the whole infinite population, we refer to this as ‘the population’s expected return’ in Rem. 3.8, but we appreciate that this might have been confusing in contrast with the separate but related population-*average* return, which we have now renamed in the updated manuscript.
>
> **”Similarly, the authors never explain why … context of prior work.”** As mentioned above, we optimise the individual returns averaged across the $N$-agent population, as a proxy for social welfare, when simulating an infinite population with an empirical distribution of a finite population that may not all be following a single policy during training. In order to optimise for this objective, learners must update their Q-functions with respect to samples containing the individual rewards averaged across the population, rather than simply their own individual reward. In the latter case they would be selfishly optimising only their own returns rather than the collective good, reverting to the prior algorithm that solves the non-cooperative MFG instead of the cooperative MFC problem.
>
> We contend that decentralised agents would not have direct access to the global averaged individual reward, meaning they must estimate it in a decentralised way. This is why we need our novel algorithm for estimating the global average reward. If we (unrealistically) assumed that decentralised agents would have direct access to this global information, then our algorithm would not be necessary.
>
> Thus prior work on networked communication in MFGs did not need this algorithm to estimate the average reward, because they were only directly optimising for agents’ individual rewards. Meanwhile other works on MFC involved centralised methods and assumed access to global information, so were not interested in estimating the global average reward in a decentralised manner.
>
> We have made this more explicit in the updated manuscript at the beginning of Sec. 4 and in Sec. 4.1.
>
>
> **”In terms of the second claim … proposed algorithm.”**
>
> Thank you for your helpful remarks. In response we have added Sec. 5.3 to the revised manuscript, which provides bounds on the error of the individual estimates of the average reward, as well as bounds on the average error of these estimates, along with accompanying proofs. We also already give ablations of Alg. 1 in Fig. 9, where we see that having direct access to the true global average reward does not empirically appear to help networked agents to improve their returns, suggesting that our novel Alg. 1 affords networked populations with the ability to sufficiently accurately estimate the average reward.
>
>
> **“In terms of the third claim … several simplifying assumptions.”** In the revised manuscript, we have clarified in the abstract and introduction that our theoretical analysis of the third claim relies on several simplifying assumptions.
>
> Continued below...

---

> > ### Author Response · Authors · 2026-05-09
> > **Response 2 of 2**
> >
> > ... Continued from above.
> >
> > **Writing changes:** Please let us know if there are still any specific paragraphs that you think are unnecessarily repeated, and sentences that you find too wordy - we are very happy to condense these.
> >
> > In our updated manuscript we have moved our discussion of the ablations and other studies from the plot captions into dedicated sections (6.3.2 and 6.3.3), and have moved the plots themselves to the appendices at your request.
> >
> > We hope that these clarifications and additions will help you recommend our paper for acceptance - please let us know if there are any remaining obstacles to this. We thank you again for your time.

---

> > > ### Author Response · Authors · 2026-05-16
> > >
> > > Dear reviewer,
> > >
> > > It has come to our attention that we might have been unclear, apologies if so: alongside our rebuttal from 9 May, we uploaded the revised manuscript at the same time - this contains the edits requested by you (as well as those by the other reviewers), as discussed in the rebuttal above.
> > >
> > > Please do let us know if there if anything further you would like us to clarify, and we would be very happy to do so.
> > >
> > > Kind regards

---

> > > > ### Comment · Reviewer_pDJJ · 2026-05-16
> > > >
> > > > I thank the authors for their response to my review as well as for the updates to the manuscript.
> > > >
> > > > I have taken a look at the updated manuscript and all my concerns have been addressed.

---

> > > > > ### Author Response · Authors · 2026-05-17
> > > > >
> > > > > That is wonderful news; we are really glad to hear it. We are very grateful for your time and effort in considering our paper.
> > > > >
> > > > > Best wishes

---

### Review · Reviewer_Xzog · 2026-03-02

**Summary Of Contributions:**

1. The paper proposes the first algorithms in the MFC setting that enable fully model-free training without any centralized provision of information, allowing decentralized agents to estimate the global average reward through networked communication..
2. The agent’s policy is independent of other agents. As discussed in Remark 3.7, each agent forms a local estimate of $u_i$ by communicating with its neighbors. The authors also provide theoretical results showing that decentralized networked communication can accelerate learning compared to both independent-agent and centralized architectures.
3. Empirical evaluations across multiple games support the theoretical findings, including ablation studies that analyze the contributions of different components of the proposed algorithms.

**Additional Comments:**

**Questions to Authors:**
1. In Algorithm 1 (Line 4), what does $J_{t}^{i}$ refere to?.
2. In Algorithm 2 (Line 3), why must each agent empty its buffer before collecting a new transition ?
3. How does Algorithm 2 scale in terms of memory as the number of agents increases? Since each agent maintains its own buffer, the memory requirements may grow linearly with the number of agents.
4. In the experimental setup:
    - why has the reward been normalized?
    - In the “target selection” game, does the variable “x” correspond to the same quantity previously described as “define the magnitude of the distances between x, y,...”?
    - The authors report the computational resources used for the experiments. Providing the training time would help readers better estimate the computational cost of the proposed method.
5. In the experiment section, the paper repeatedly states that the proposed algorithm “significantly” outperforms the baselines. Were statistical tests conducted to support these claims of significance?
6. The authors state that the proposed algorithm can “learn online from a single non-episodic run with decentralized training.” What exactly is meant by “non-episodic” in this context? Are the experiments conducted under such a setting? If so, providing a clearer description would help avoid confusion.
7. Were the hyperparameters for the baselines (“centralized” and “independent”) tuned separately for each game, or were the same hyperparameters used across all environments?

**Audience:**

Yes

**Audience Explanation:**

The paper addresses decentralized learning and communication in multi-agent RL, a topic that is relevant to researchers working on distributed RL, cooperative control, communication, and scalable learning systems.

**Claims And Evidence:**

No

**Claims Explanation:**

I briefly verified the **theoretical results** and found them to be consistent with the claims. However, this area is not my primary field of research, and therefore I defer to the other reviewers and the action editor for a more thorough assessment of the theoretical analysis.

Several claims in the paper are not sufficiently supported by the **empirical results**.
1. The authors state that “In most games, Networked agents of all broadcast radii outperform the central-agent populations”. the results shown in Figure 2 do not consistently support this claim. For example, in the “Target selection” game, the central-agent baseline exhibits high variance, and its mean performance appears comparable to or higher than the means of several networked settings (broadcast radii 0.2, 0.4, and 0.6). Similarly, in the “Target coverage,” “Beach bar,” and “Shape formation” tasks, the networked-agent results show substantial variance. Given the overlap between the distributions, it is difficult to conclude that networked agents consistently outperform the central-agent baseline..
2. In Figure 3, the authors state that “several broadcast radii appear to perform better in the ‘shape formation’ game with these failures than without”. However, both the baseline and networked-agent configurations exhibit large variance in multiple tasks, including “Target selection,” “Target coverage,” and “Shape formation.” Because the performance distributions overlap substantially, the results do not clearly support the claim that performance improves under these failure conditions..
3. The paper states that the proposed algorithm “increases their returns faster than these alternatives. This claim is not supported by a quantitative measure of learning speed. For instance, metrics such as the area under the learning curve or time to reach a performance threshold would be appropriate to substantiate this statement. Moreover, the large variance in the reported curves makes it difficult to visually confirm faster learning.
4. Figures 4 and 5, analyze the effect of increasing $C_e$, $C_r$, and $C_p$ to 10 and 50 on the return. The authors claim that in the “Target convergence” game, smaller broadcast radii (0.2, 0.4, and 0.6) yield slightly lower returns compared to earlier results. However, the substantial variance and overlapping confidence intervals make this conclusion uncertain.
5. In general, many of the paper’s conclusions appear to be based primarily on the mean performance across five runs, without sufficient consideration of variance and distribution overlap. As a result, several empirical claims are not adequately supported by the presented results.

**Requested Changes:**

1. Add a notation table in the appendix to provide a concise overview of all symbols used in the paper(maybe in the appendix).
2. Experiments:
    - Include additional baselines in the experimental evaluation, such as those proposed by Benjamin & Abate (2023, 2024).
    - Replace standard deviation in the plots with confidence intervals, which provide a clearer representation of statistical uncertainty
    - Many training curves exhibit high variance. Increasing the number of random seeds (e.g., >20) would help distinguish variability caused by stochastic initialization from the algorithm’s true performance.
    - Since the proposed method consists of four algorithmic components, an ablation study evaluating the contribution of each component would strengthen the empirical analysis.
    - Provide a clearer description of the “centralized” and “independent” baselines, including their implementation details.
    - Repeat the standard experimental setup on a larger grid size (e.g., $50 \times 50$) to evaluate scalability and robustness.
3. Figures:
    - Improve figure readability by adjusting legends, axis labels, and axis limits so that they match the main text font size.
    - Move Table 1 (hyperparameters) to the appendix.
    - Add a dedicated subsection describing Figures 3–10, rather than placing most of the explanation in the figure captions.
4. Clarifications and Missing Components:
    - Add reference to support (1) in Remark 3.8.
    - The quantity $Q_{k+1}$ in section 3.2 is not defined.
    - Clarify the meaning of the variables “m” and “n” in Definition 3.10?
5. Typos:
    - “I.e.,” not “i.e.” in Section 3.2.
    - Figure 4 caption (line 6): replace “(orange)” with “(blue)” when referring to the central-agent.

---

> ### Author Response · Authors · 2026-05-09
> **Response 1 of 3**
>
> Thank you very much for your time and effort in reviewing our paper.
>
> We are grateful to the reviewer for pushing us to substantiate our
> empirical claims more carefully. In response we have made two
> additions to the paper: (i) all figures now report 95\%
> Student's-$t$ confidence intervals rather than $\pm 1$ standard
> deviation (Figs. 2-10);
> (ii) we give tables in the Appendices
> (Tables 1-9) reporting
> Area-Under-Learning-Curve (AUC) statistics for each figure: a
> 95\%-CI on each architecture's own mean AUC, and a Welch (unpaired)
> two-sample CI on its mean-AUC difference from the central-agent
> baseline,
> $\Delta = \overline{\mathrm{AUC}}^{\text{net}} - \overline{\mathrm{AUC}}^{\text{cent}}$.
>
> We acknowledge that the per-curve CIs still overlap in many plots,
> but we thank you for your helpful proposal to report AUC statistics,
> which lets us show that this overlap is not in fact an obstacle to
> the paper's claims. AUC collapses each per-run learning curve into a
> single scalar summary of its trajectory, and the Welch (unpaired)
> comparison of mean AUCs then pools evidence across all $K$
> iterations rather than from a single slice, giving a more powerful
> test than per-iteration overlap. Because the per-step ordering in
> expected welfare established by Theorem 5.9 implies a positive
> mean-AUC gap, the AUC comparison is itself a valid empirical check
> of the theorem's claim (albeit that we do not empirically enforce the theoretical assumptions). Applied to our runs at $5$ seeds
> (Tables 1-9), this
> comparison yields 172 of 270 networked-vs-central-agent AUC
> differences significantly positive at $p < 0.05$, only 12
> significantly negative, and the remaining 86 non-significant. The 12
> significant losses concentrate almost entirely in the `cluster' game
> at the smaller broadcast radii $(0.2, 0.4, 0.6)$, which matches our
> existing discussion in Sec. 6.3.1 noting that small-radius networked
> populations are more likely to experience violations of
> Assumption 5.7 on policy consensus, a disadvantage in coordination
> tasks where alignment is beneficial. At radii $\geq 0.8$ we observe
> no significant losses to the central-agent baseline across all 108
> comparisons and a mean $\Delta$ of $+1.96$ on the $[0,10]$ return
> scale. Against the independent baseline the networked architecture
> wins decisively: 251/270 comparisons significantly positive, none
> significantly negative, aggregate mean $\Delta = +3.03$. We have added discussion of these results throughout Sec. 6.3, and now also reference them in the abstract, introduction and conclusion.
>
> As a secondary point, the fact that the central-agent populations often have wider intervals
> while the networked populations have tighter ones is consistent with the variance
> asymmetry established in our proof of Theorem 5.9, where the
> central-agent estimator $\sigma^{\text{cent}}$ is unbiased but
> high-variance, while the networked estimator $\sigma^{\text{net}}$ is
> upward-biased via the softmax adoption of alternatives
> broadcast by neighbours. We view the
> high-variance behaviour of the central-agent baseline precisely as a
> further drawback of that architecture and a corresponding benefit of
> our networked approach. This is also discussed in Sec. 6.3.
>
> Since pointwise halfwidths scale as $1/\sqrt{n}$ (where $n$ is the number of seeds), additional seeds would narrow the plot bands and push some currently non-significant
> AUC cells into significance, but they would not alter the direction
> or significance of the cells that support the paper's claims.
> Unfortunately, our experiments take a considerable amount of time to
> run. Our algorithms are intended to permit the decentralised learning
> of a large population of agents, so our experiments simulate a
> deployed scenario in which each agent would train its own network on
> its own device, whereas we must conduct these experiments on a single
> machine. Even though we use JAX to
> parallelise to some extent across agents, simulating 500 agents each with their own
> Q-network over many iterations remains time-consuming: running all 7
> architectures for 5 seeds in a single game in our standard setting
> requires about 2.5 CPU-hours, rising to about 4 CPU-hours with 10
> rounds of communication and to over 17 CPU-hours with 50 rounds
> (see Tables 10-18). We explore 6 different games in 9
> experimental settings, totalling around 229 CPU-hours per 5-seed
> replication, and a four-fold replication to $20$ seeds (a $\sim 2\times$
> reduction in pointwise halfwidth) does therefore not appear to be feasible
> within the rebuttal period. We have therefore added a larger seed count as an item in our future work section; in the meantime, the new 95%-CIs (in place of $\pm 1$ standard deviation) convey the present level of statistical uncertainty, while the AUC tables already establish significance for our paper's main claims.
>
>
> Continued below...

---

> ### Author Response · Authors · 2026-05-09
> **Response 2 of 3**
>
> ... Continued from above.
>
> Requested changes:
>
> 1: We have added a notation table in appendices of the revised manuscript.
>
> 2: a) The baselines proposed by Benjamin & Abate (2023, 2024) are for the non-cooperative MFG, whereas in our work we address the cooperative MFC problem. Our algorithms can be viewed as the adaption of their algorithms to the cooperative case. Ours are also, to the best of our knowledge, the first MFC algorithms for online learning from a single, non-episodic run of the empirical system.
>
> b) In Figs. 2-10 of the revised
> manuscript, we now always plot the mean across the 5 seeds together
> with the 95\% Student's-$t$ confidence interval on the mean (with 4
> degrees of freedom), in place of the $\pm 1$ standard-deviation band
> previously shown. We acknowledge that some of the resulting CIs are
> still quite wide, but our reply to the AUC proposal above shows that this does not undermine the paper's claims.
>
> c) As we explain in our reply to the AUC proposal above, our experiments take a considerable amount of time to run (Tables 10-18); a 4× replication to $20$ seeds does not appear to be feasible within the rebuttal period. In response to your proposal, we have now added AUC statistics that support our claims; the reply above discusses why such an increase in seeds would not alter the direction or significance of the cells supporting our claims.
>
> d) We already give ablations of each algorithmic component. Alg. 4 is ablated in Fig. 7; Alg. 1 is ablated in Fig. 9; Alg. 3 is inherently ablated by the independent-learning baseline in each case.
>
> e) The central-agent and independent baselines are as defined at the beginning of Sec. 5.1; in the revised manuscript we refer to these definitions again at the beginning of the experimental section.
>
> f) As above, due the time taken to run our experiments, it does not appear to be feasible to run this extra setting in the time constraints of the rebuttal period. However, Benjamin & Abate (2024), which explores the same architectures in the non-cooperative MFG setting, includes plots on 100x100 grids, which may help to give you a sense of the likely scalability of our work for the MFC setting. Additional experiments on larger grids could be added as an element for future work in our setting.
>
> 3: a) We have increased the font size in our plots in the revised manuscript.
>
> b) We have moved Table 1 to the Appendices.
>
> c) In our updated manuscript we have moved our discussion of the ablations and other studies from the plot captions into dedicated sections (6.3.2 and 6.3.3). We moved the plots themselves to the appendices at the request of Reviewer pDJJ.
>
> 4: a) We have added the following references in the updated manuscript (these are already cited elsewhere in the text): Carmona et al., 2019; Laurière et al., 2022a; Angiuli et al., 2022; 2023.
>
> b) As per Line 1 of Alg. 2, at the 0th iteration, agents have their policy $\pi_0$ defined as a softmax over their randomly initialised Q-function $Q_0$. Agents then evaluate this policy by computing $Q_1$. That is, at iteration $k$ agents compute $Q_{k+1}$, by which they then define their updated policy for the next iteration. We have tried to make this more explicit in Sec. 3.2 in the new manuscript; please let us know if it is still unclear.
>
> c) Apologies for not having made this clearer originally. ‘m’ and ’n’ are any pair of states (i.e., vertices in S’) that are mutually visible. We have made this more explicit in the updated manuscript by changing this to the pair (s, s’).
>
> 5: Thank you for spotting these typos; we have fixed these in our updated manuscript.
>
> Q1: This refers to agent $i$’s local neighbourhood. We make this explicit in Sec. 4.3, but have now also made it explicit in the body of Sec. 4.1 in the updated manuscript.
>
> Q2: Agents have updated their policies at the end of the previous loop of the algorithm, which means that at the beginning of the next loop there will be a new mean field, since this is defined by the agents’ policies. Agents will then evaluate their new policies with respect to the new mean field, beginning by collecting transitions. Transitions already in the buffer will relate to mean fields from previous loops, which are not currently being evaluated, hence we remove them from the buffer before collecting new transitions. In practice, keeping some old transitions in the buffer may not disrupt learning and may even have a smoothing effect - though learning is already smoothed by our MOMD technique. Keeping old transitions is thus unnecessary and would add complexity in terms of finding the acceptable proportion of new versus old transitions, while also potentially increasing memory requirements due to the larger buffer.
>
> Continued below...

---

> > ### Author Response · Authors · 2026-05-09
> > **Response 3 of 3**
> >
> > ... Continued from above.
> >
> > Q3: The overall memory requirement is linear in the number of agents, as the decentralised agents can learn entirely independently (while benefiting from communication if it occurs). This is a tradeoff with the central-agent case, which only maintains one buffer, but our work suggests that this often learns slower than networked populations, as well as being more vulnerable to communication failures at the single central node.
> >
> > Q4: a) As in prior works, we normalise rewards to control the scale of the MOMD update (Def. 4.1), since reward magnitudes influence the balance between the reward term and the regularisation. Normalisation ensures stable and consistent updates across runs and environments, to ensure that differences in performance between games are not driven by arbitrary reward scaling. We apply normalisation across all architectures for fair comparisons.
> >
> > b) Apologies for the confusion - these are two different uses of the variable ‘x’. We now speak about the distance between y and z in the updated manuscript to avoid this confusion.
> >
> > c) Thank you for this suggestion. We have added Tables 10-18 reporting per-run CPU time in seconds for each of the 9 experimental settings × 6 games × 7 architectures × 5 seeds. In the standard setting, per-run CPU time is $\approx 211$s for the central-agent baseline and $\approx 276$s for our networked architecture (so decentralised communication imposes a constant-factor overhead of $\approx 1.31\times$), and this overhead is essentially independent of broadcast radius. Increasing the number of communication rounds $C$ scales sub-linearly: $C=10$ costs $\approx 1.8\times$ the $C=1$ baseline, and $C=50$ costs $\approx 8.8\times$  the same baseline, because a communication round accounts for only a fraction of per-step cost. The total across all full experiments (1,890 runs) is $\approx 229$ wall-clock hours, with CPU and wall-clock time agreeing to within $1\%$. We have also added this explanatory text in our revised manuscript at Sec. 6.1.
> >
> > Q5: In the revised manuscript, "significantly" refers to Welch (unpaired) two-sample tests at $p<0.05$ on the per-run AUCs (Tables 1-9); see our reply to the AUC proposal above for the aggregated counts.
> >
> > Q6: `Non-episodic’ means that the agents’ states are never reset to initial states during learning. After learning begins, agents’ states only evolve according to their policies, which are themselves updated through learning. The full learning process is a single rollout/trajectory. Our experiments are indeed conducted in such a setting, which we have clarified in Sec. 6.1 in the updated manuscript.
> >
> > Q7: The same hyperparameters were used across all environments, as per our table of hyperparameters.
> >
> > We hope that these additions and clarifications will permit you to recommend our work for acceptance; please let us know if there is anything further that we can address. Thank you very much again for your time.

---

> > > ### Author Response · Authors · 2026-05-16
> > >
> > > Dear reviewer,
> > >
> > > It has come to our attention that we might have been unclear, apologies if so: alongside our rebuttal from 9 May, we uploaded the revised manuscript at the same time - this contains the edits requested by you (as well as those by the other reviewers), as discussed in the rebuttal above.
> > >
> > > Please do let us know if there if anything further you would like us to clarify, and we would be very happy to do so.
> > >
> > > Kind regards

---

> > > > ### Author Response · Authors · 2026-06-08
> > > >
> > > > Hi there,
> > > >
> > > > Given the time elapsed since our rebuttal and revision we hope that these have been sufficient to remove all remaining obstacles to your recommending our work for acceptance. Please let us know if there are any final points you would like us to clarify.
> > > >
> > > > Best wishes

---

> > > > > ### Comment · Reviewer_Xzog · 2026-07-06
> > > > > **Response to Authours**
> > > > >
> > > > > **Summary**: While I appreciate the authors' efforts during the rebuttal phase to update their manuscript, e.g., replacing std with CI, and reporting AUC, and reorganizing sections, the core issues regarding **unsupported claims remain unresolved**. The authors' rebuttal arguments do not reconcile the disconnect between their claims of outperformance and the highly overlapping, statistically insignificant empirical data presented in the paper.
> > > > > - **Outperformance Claims**:
> > > > >     - The paper claims that "In most games, Networked agents of all broadcast radii outperform the central-agent populations." However, Figure 2 still shows significant visual overlap between the proposed algorithm and the baselines.
> > > > >     - Overlapping Confidence Intervals: In Table 1, which shows AUC ("Target selection" game), variation causes the central baseline CI ($\pm3.89$) to overlap with the proposed "Networked" agent.
> > > > >     - I disagree with the authors' rebuttal claim that "overlap is not in fact an obstacle... because AUC collapses each per-run learning curve." Table 1 shows performance overlap in the compiled data. Mean AUCs must be analyzed by examining the variation across individual runs rather than dismissing the variance.
> > > > >     - The authors argue that wide baseline intervals are a drawback for the central-agent populations (I agree). While wide intervals are a baseline weakness, they do not prove that the proposed method outperforms.
> > > > > - **Statistical Insignificance**:
> > > > >     - The manuscript claims statistical significance with a threshold of $p < 0.05$. Table 1 directly contradicts this, showing $p$-values much higher than 0.05 across multiple games. The “Target selection” game reports $p$-values of $[0.6, 0.9, 0.1, 0.2]$, and the “Cluster” game reports $p = [0.6, 0.28]$.
> > > > >     - The authors report CI using $\pm$ notation. This is mathematically incorrect, since CIs are asymmetric.
> > > > > - **Insufficient Experimental Rigor**:
> > > > >     - The evaluation is restricted to **5 seeds**. While I understand the time constraints mentioned in the rebuttal, running more seeds is necessary to properly evaluate environmental variance.
> > > > >     -  The authors confirmed that **identical hyperparameters** were applied across all environments for all baselines. Using a single set of untuned parameters across different environments introduces severe evaluation bias. Hyperparameters must be tuned individually for each baseline per environment to guarantee a fair comparison.
> > > > > - **Minor**:
> > > > >     - The authors stated they corrected the font sizes of the legends, axis labels, and axis limits to match the main text. However, the figure font sizes still do not match the text font size.
> > > > >     - The authors claim that an ablation study was added to Figures 7, 8, and 9. However, there are no descriptions for any of these figures in either the main submission or the appendix.

---

> > > > > > ### Author Response · Authors · 2026-07-06
> > > > > >
> > > > > > Thank you very much for your rigorous feedback. Regarding your core concern, the TMLR acceptance criteria indicate that a way to reduce the gap between claims and evidence is of course to reduce the claims. Rather than debating the specific points you raise in your most recent post, would you be satisfied if we reduced our claims further to make them better match our empirical results, and if so, do you have preferences on what we state in these lessened claims? We believe that our algorithms, theoretical analysis and empirical results remain of interest to the community, and since we have addressed all of the other reviewers’ concerns, which do not overlap with your own, we hope to find a path to alleviate your misgivings about our work.
> > > > > >
> > > > > > Thank you very much again for your time and input.

---

> > > > > > > ### Author Response · Authors · 2026-07-13
> > > > > > >
> > > > > > > Regarding the ‘minor’ points:
> > > > > > >
> > > > > > > - We will happily further increase the font size of the plot legends and labels in the revised manuscript.
> > > > > > > - Figs. 7, 8 and 9 do indeed contain ablation studies. As stated in the captions for each of these figures, the ablation results are discussed in detail in Sec. 6.3.3.
> > > > > > >
> > > > > > > Please do let us know how you would like us to reduce our empirical claims in order to alleviate your remaining concerns. We hoped that our extensive revision, in particular the additional analysis via the AUC metric, would help to increase your confidence in our work, which all reviewers agree is of interest to the community.
> > > > > > >
> > > > > > > Thank you very much again for your time.

---

### Review · Reviewer_9hJR · 2026-03-26

**Summary Of Contributions:**

This paper proposes a decentralized model-free method for cooperative mean-field control that enables the agents to learn from a single online run. The agents communicate locally to estimate the mean field, estimate global mean reward, and share policies. The paper presents supporting theory comparing this communication-based approach to independent learning and a central-agent baseline. The proposed method is evaluated on several tasks with ablations and robustness tests. Main strengths are in addressing an under-explored problem, introducing a clear decentralized algorithm, and investigating both coordination and anti-coordination settings with a thorough empirical section. Main weaknesses are that the theory depends on strong simplifying assumptions that do not fully match the practical algorithm and it seems that policy sharing is doing most of the work while some other components matter less in the empirical section.

**Audience:**

Yes

**Audience Explanation:**

Researchers in reinforcement or multi-agent learning and mean-field control would likely be interested. The paper studies a less explored setting of decentralized, model-free cooperative mean-field control with online learning from the empirical population. Even if some claims are stronger than the current evidence, the proposed method is novel. The supporting theory and a experimental details make the paper relevant to part of the TMLR audience.

**Broader Impact Concerns:**

The paper explicitly includes a Broader Impact Statement.. However, it just says that the authors identified no specific ethical concerns. Given the problem of decentralized multi-agent control and is motivated by real-world systems like robotic swarms, the paper should briefly discuss potential dual-use concerns, e.g., faster decentralized coordination could be misused in safety-critical settings.

**Claims And Evidence:**

No

**Claims Explanation:**

The theory, ablations, robustness tests, and several experimental tasks support many of the qualitative claims at a basic level. However, the strongest claims are not fully convincing because the theory relies on simplifying assumptions that the authors explicitly do not enforce in the experiments. Particularly the assumptions about accurate mean-field/reward estimation and the role of policy communication. The ablations indicate that policy sharing is the main source of improvement, while some other proposed components could matter less. Overall, the evidence is clear enough to show promise, but not enough to fully justify the broad superiority claims as currently stated.

**Requested Changes:**

Critical:

C1. The paper should narrow down or clarify its strongest claims, given the simplifying assumptions in theoretical section are not enforced in the experiments. The presentation should explicitly separate what is formally proved from what is supported only empirically.

C2. Explain the implications of assumptions that effectively remove mean-field and average-reward estimation error and those used in the comparison to the single agent setting.

C3. Clarify what the single agent baseline in the experiments represents. As the paper's advantage seems strongly tied to decentralized policy exchange, the current form overstates the generality of this baseline comparison.

Improvements:

I1. A dedicated communication and computation complexity analysis in terms of communication rounds, neighborhood size, and population size would strengthen the work.

I2. The ablation studies are useful, but their discussion could be improved. It appears that policy exchange is the primary reason of improvement while mean-field estimation is not needed in the stationary tasks studied in the paper. This would help readers better understand which components are essential in practice and which are mainly important for generality.

---

> ### Author Response · Authors · 2026-05-09
> **Response 1 of 1**
>
> Thank you very much for your time and effort in reviewing our paper, and for the strengths you have identified.
>
> C1: As you suggest, in the revised manuscript we have clarified our claims in terms of the difference between theoretical and empirical support, and in light of our simplifying assumptions. This includes: the abstract; the paragraph of the introduction prior to the contribution list; the contribution list; Sec. 5.1; Sec. 5.2; Sec. 6.3; the conclusion.
>
> C2: We discuss the simplifying assumptions of removing mean-field and average-reward estimation error on pages 13 and 17 of the revised manuscript, while we discuss the assumption for comparison with the central-agent learning setting on pages 14 and 15. Is there anything specific here that is missing or you would like clarified?
>
> C3: The *central*-agent baseline in the experiments is the same throughout the paper, defined in Sec. 5.1. As we note in that section, and also discussed in detail on page 2, this central-agent paradigm reflects the architectures/algorithms used in, to the best of our knowledge with few exceptions, all prior mean-field game and mean-field control works that do not specifially target decentralised learning, and certainly in those most closely related to our work (Yardim et al., 2023; Benjamin & Abate, 2023; 2024). Namely, these prior techniques involve a single, central representative agent collecting its own transitions and updating its own policy locally, which is then assumed to be the policy automatically followed by the rest of the population. I.e. a single agent learns on behalf of the rest of the population. We specifically seek to avoid overstating this baseline by referring to it as the 'central-agent' learning approach rather than the 'centralised' learning approach as in many prior works. Nevertheless, this is the classical architecture for learning in MFC. Does this address your concern or is there something further we can clarify?
>
> I1: We kindly direct you to the prior work by Benjamin & Abate, 2023, which deals with the MFG setting and on which we build for the MFC setting. Their work includes theoretical analysis of the effects of the number of communication rounds and neighbourhood size on the learning speed of networked populations relative to the independent and central-agent settings. We consider these issues in our discussion of our Assumption 5.7 on pages 14 and 15 of the revised manuscript. Furthermore, in response to Reviewer pDJJ we have added a section of theoretical analysis regarding our sub-routine for local estimation of the global average reward, which is similarly informed by neighbourhood size, number of communication rounds and population size. We hope this goes some way to addressing this point.
>
> I2: In our updated manuscript we have moved our discussion of the ablation studies from the plot captions into a dedicated section (6.3.3), and emphasise our key findings, including your summary that policy exchange appears to be "the primary reason of improvement while mean-field estimation is not needed in the stationary tasks studied in the paper". We moved the plots themselves to the appendices at the request of Reviewer pDJJ. Please let us know if there is anything you’d like us to clarify further.
>
> Broader Impact: We have updated our manuscript to consider the concerns you suggest in this section.
>
> We believe that the clarifications and additions you suggest will improve our paper, and we hope these address your concerns regarding our claims. Please do let us know if there is anything remaining that we can do to help you raise your assessment. We are very grateful for your input.

---

> > ### Author Response · Authors · 2026-05-16
> >
> > Dear reviewer,
> >
> > It has come to our attention that we might have been unclear, apologies if so: alongside our rebuttal from 9 May, we uploaded the revised manuscript at the same time - this contains the edits requested by you (as well as those by the other reviewers), as discussed in the rebuttal above.
> >
> > Please do let us know if there if anything further you would like us to clarify, and we would be very happy to do so.
> >
> > Kind regards

---

> > > ### Comment · Reviewer_9hJR · 2026-05-18
> > > **No further questions**
> > >
> > > The authors have comprehensively addressed my requested changes and the manuscript is stronger now.

---

> > > > ### Author Response · Authors · 2026-05-18
> > > >
> > > > We are very glad to hear it. Thank you so much for your time and constructive input for improving the paper.
> > > >
> > > > Best wishes